# Future projections of Siberian wildfire and aerosol emissions

Reza Kusuma Nurrohman[1,2], Tomomichi Kato[3], Hideki Ninomiya[4], Lea Végh[3,5], Nicolas Delbart[6], Tatsuya Miyauchi[3], Hisashi Sato[7], Tomohiro Shiraishi[8] and Ryuichi Hirata[5]

[1]Graduate School of Agriculture, Hokkaido University, Sapporo, Hokkaido 060-8589, Japan
[2]Department of Agricultural Engineering, University of Mataram, Mataram, Nusa Tenggara Barat 83126, Indonesia
[3]Research Faculty of Agriculture, Hokkaido University, Sapporo, Hokkaido 060-8589, Japan
[4]Graduate School of Global Food Resources, Hokkaido University, Sapporo, Hokkaido 060-8589, Hokkaido, Japan
[5]National Institute for Environmental Studies, Tsukuba, Ibaraki, Japan
[6]Laboratoire Interdisciplinaire des Energies de Demain, UMR 8236 CNRS – Université de Paris, 75013 Paris, France
[7]Japan Agency for Marine-Earth Science and Technology (JAMSTEC), 3173-25 Showamachi, Kanazawa-ku, Yokohama-City, Kanagawa 236-0001, Japan
[8]Nippon Bunri University, Oita, Oita, 870-0397, Japan

*Correspondence to*: Tomomichi Kato (tkato@agr.hokudai.ac.jp)

**Abstract.** Wildfires are among the most influential disturbances affecting ecosystem structure and biogeochemical cycles in Siberia. Therefore, accurate fire modeling via dynamic global vegetation models is important for predicting greenhouse gas emissions and other burning biomass emissions to understand changes in biogeochemical cycles. In this study, we integrated the widely used SPread and InTensity of FIRE (SPITFIRE) fire module into the dynamic global vegetation model (SEIB-DGVM) to improve the accuracy of fire predictions and then simulated future fire regimes to better understand their impacts. The model is able to reproduce historical data well compared to the benchmark datasets. Based on the spatial validation, the results are as follows: Aboveground biomass ($R^2$=0.847, RMSE=18.3 Mg ha$^{-1}$), burned fraction ($R^2$=0.75, RMSE=0.01), burned area ($R^2$=0.609, RMSE=690 ha), dry matter emission ($R^2$=0.624, RMSE=0.01 kg DM m$^{-2}$), $CO_2$ emissions ($R^2$=0.705, RMSE=6.79 Tg). Overall, the model is able to produce output with spatial distribution patterns similar to the benchmark dataset, with an average similarity of 70.7%. Furthermore, based on the comparison of mean values with the benchmark datasets, the model produces high accuracy, amounting to 99%. We estimated that the $CO_2$, CO, $PM_{2.5}$, total particulate matter (TPM), and total particulate carbon (TPC) emissions in Siberia in 20-year historical period (2000-2020) will increase relatively by 189.66 ± 6.55, 15.18 ± 0.52, 2.47 ± 0.09, 1.87 ± 0.06, 1.30 ± 0.04 Tg species year$^{-1}$, respectively in the 20-year future period (2081-2100) under the Representative Concentration Pathways 8.5. Under the same climate scenario and period comparison, we estimated that the number of trees burnt increased by 100 %, resulting in a 385.19 ± 40.4 g C m$^{-2}$ year$^{-1}$ loss of net primary production (NPP). These findings show that Siberia faces an increasing frequency of extreme fire events due to changing climate conditions. Our study offers insights into future fire regimes and provides helpful information for development strategies for enhancing regional resilience and for mitigating the broader environmental consequences of heightened fire activity in Siberia.

## 1 Introduction

Fires are among the most significant disturbances affecting biogeochemical cycles, atmospheric chemistry, the carbon cycle, and ecosystem structure and function worldwide (Pickett et al., 1999). Wildfires are also the dominant climate-driven disturbance agent in boreal forests (Goldammer and Furyaev, 1996; Shorohova et al., 2011; De Groot et al., 2013), shaping major forest cover in Russia (Krylov et al., 2014) and rapidly increasing burned area and emission intensity in Canada and Alaska (Zheng et al., 2021). Fires influence vegetation dynamics by allowing plants to adapt to fire regimes, influencing vegetation productivity, litter, and fuel load (Cochrane, 2003; Bergeron et al., 2004; Whelan, 2009). The intensity and frequency of large-scale boreal forest fires are expected to increase in the future due to increased global temperatures, drier conditions, and longer fire seasons, which will cause more emissions from biomass burning (Flannigan et al., 2009; Gauthier et al., 2015) and human activity by using fire for land management (e.g. use of fire as a tool in the deforestation process) (Hantson et al., 2016; Archibald et al., 2013; Morton et al., 2008). Globally, from 2000 to 2019, satellites detected a decrease in the burned area of grassland, while there was a slight increase in the area of forest fires in Russia (Zheng et al., 2021). Europe, France, Spain, Portugal, and Greece are already experiencing larger and more devastating fires (Carnicer et al., 2022). Not only large fires but also small fires have a significant impact: Areas burned by small fires contributed 35% to the total burnt area, from 345 Mha year$^{-1}$ to 464 Mha year$^{-1}$, and related carbon emissions increased from 1.9 Pg C year$^{-1}$ to 2.5 Pg C year$^{-1}$ from 2001-2010 (Randerson et al., 2012). This finding is in line with current studies reporting that the global mean $CO_2$ emission intensity has increased by $0.9 \pm 0.9\%$ year$^{-1}$ from 2000 to 2019 (Zheng et al., 2021) and that the Fire Weather Index (FWI) reached levels above 30, corresponding to high, very high, and extreme levels of fire frequency, causing $CO_2$ emissions to increase in Europe since 1980 (Carnicer et al., 2022).

Forest fires are important ecological factors that influence both the establishment and succession of vegetation (Abaimov and Sofronov, 1996). Climate-driven large fires are responsible for rapid changes in vegetation (Cleve and Viereck, 1981), soil properties (Pastor and Post, 1986; Pellegrini et al., 2021), biogeochemical cycling, microclimate, forest ecosystems (Crutzen and Goldammer, 1993), productivity, stability, and many other ecological properties (Melillo et al., 1993). Forest fires also indirectly affect vegetation dynamics by increasing $CO_2$ levels in the atmosphere (Seiler and Crutzen, 1980; Nguyen and Wooster, 2020), as $CO_2$ is one of the primary products of biomass combustion and is emitted in all phases of fire (ignition, flaming, glowing, pyrolysis, and extinction) (Andreae and Merlet, 2001), with the flaming phase leading to emissions (Lobert et al., 1991; Ward and Hardy, 1991). Thus, it is challenging to estimate $CO_2$ emissions because they are generated in large quantities during biomass combustion and because of the different emission timelines produced during each combustion stage. Increasing atmospheric $CO_2$ concentrations alter the global carbon cycle by causing global warming (Van Der Werf et al., 2006, 2010, 2017; Neto et al., 2009; Kaiser et al., 2012; Lin et al., 2013), and the resulting global warming is expected to intensify extreme fire seasons, leading to further surges in carbon emissions that significantly contribute to the global burden of greenhouse gases (fire-climate feedbacks) (Bowman et al., 2009). This event also affect the agricultural sector positively and negatively depending on the region, environment, and crop types (Kimball and Idso, 1983).

Additionally, prolonged exposure to very high $CO_2$ concentrations at ground level has negative impacts on health (Jacobson et al., 2019). Therefore, accurate modeling of future wildfires and their emissions is required to understand the associated risks.

Boreal vegetation store between 17% of the world's carbon, yet encompasses almost 30% of all terrestrial carbon stocks (Kasischke, 2000; Gauthier et al., 2015), and two-thirds are located in Siberia, Russia (Shvidenko and Nilsson, 2003). In Siberia, burned biomass emissions approached 0.4 Gt C year$^{-1}$ in 2021, three times the average value between 1997 and 2020, according to GFED4s (Friedlingstein et al., 2020). Kharuk et al., (2022) also stated that the decadal frequency of wildfires tripled between 2001–2010 and 2011–2020. Catastrophic boreal forest fires are expected to continue to increase in the future due to increased global temperature, drier conditions, and longer fire seasons, and these fires will increase the severity and emissions produced from biomass burning (Flannigan et al., 2009). Burning vegetation is a major source of black carbon (BC), carbon monoxide (CO) (Forster, P. et al., 2018), and particulate matter (PM) (Reddington et al., 2016). According to records from the Copernicus Atmosphere Monitoring Service (CAMS), Russia experienced a drastic increase to 8 megatons (Mt) in $PM_{2.5}$ emissions in 2021, which is 78% higher than the average level between 2004 and 2021 (4.5 Mt) (Romanov et al., 2022). Furthermore, an increase in atmospheric emissions negatively affects the climate by contributing to global warming and climate change (Randerson et al., 2006; Westerling et al., 2006; Bowman et al., 2009) and affect weather systems by modulating solar radiation and cloud properties (Schultz et al., 2008).

Understanding how long-term climate change, fire regimes, and forest vegetation interact under multiple climate scenarios is critical for forecasting forest succession trends (Clark and Richard, 1996). Modeling of fire regimes using dynamic global vegetation models (DGVMs) is a key approach to analyzing these factors. However, including interactive fire disturbances in vegetation models is critical for accurately simulating vegetation dynamics (Thonicke et al., 2001). A well-structured process-based fire module can accurately assess fire activity, consumed biomass due to fire, and biomass burning emissions. The assessment of each fire-related variable is interconnected with another variable, so the module must be well constructed because the amount of consumed biomass during forest fires can vary significantly. Several factors affect burned biomass, such as spatial and temporal variations in burned area based on ignition factors, the quantity and quality of the fuels available, and vegetation or plant functional type (PFT); additionally, every PFT reacts differently to fire disturbance (Cramer et al., 2001; Ito, 2011). Since the first global fire models were integrated into dynamic global vegetation models (DGVMs) two decades ago, the variety and complexity of fire models have expanded (Hantson et al., 2016). The Fire Modeling Intercomparison Project (FireMIP) compared eleven current fire models by structure and simulation protocols, using a benchmarking system to evaluate the models (Rabin et al., 2017). The results indicate that models that explicitly distinguish ignition factors, such as lightning and human-caused "ignition events", as well as physical properties and processes that determine fire spread and intensity by plant functional type (PFT), performed better. One such fire module is SPITFIRE (an upgrade of GlobFIRM) (Thonicke et al., 2010), which has been used in several DGVMs: LPJ-GUESS-SPITFIRE, ORCHIDEE-SPITFIRE, JSBACH-SPITFIRE, and LPJ-LMfire. In this study, we integrated the SPITFIRE fire module into the spatially explicit individual-based dynamic global vegetation model (SEIB-DGVM) to predict fire,

vegetation, and burned biomass emission variables in Siberia in the future. We selected the SEIB-DGVM because of its high-quality biogeochemical model coupled with a three-dimensional representation of forest structure where individual trees compete for light and space (Sato et al., 2007). The SEIB-DGVM processes physical, physiological, and vegetation dynamics and was previously used for reconstructing the geographical distributions of fundamental plant productivity properties (Sato et al., 2020), evaluating the geographic and environmental heterogeneity of larch forests with a special focus on topography (Sato and Kobayashi, 2018), and assessing the impacts of global warming on Siberian larch forests and their interactions with vegetation dynamics and thermohydrology (Sato et al., 2016). The SEIB-DGVM accurately simulates forest ecology after typhoon disturbances (Wu et al., 2019), nonstructural carbohydrate dynamics (Ninomiya et al., 2023), and masting in a temperate forest (Végh and Kato, 2024).

The original fire module of the SEIB-DGVM is Glob-FIRM (Thonicke et al., 2001), which has several limitations; for example, human-changed fire regimes and other land use impacts are not considered (Thonicke et al., 2001). In addition, GlobFIRM derives the burnt fractional area of a grid cell from the simulated length of the fire season and from the minimum annual fuel load; this method does not specify ignition sources and assumes a constant fire-induced mortality rate for each plant functional type (PFT) (Thonicke et al., 2010). To improve the fire simulations with the SEIB-DGVM, we replaced its fire module with the SPITFIRE model (Thonicke et al., 2010) by adding complete ignition equations (human and lightning effects, etc.). The module included a calculation mechanism for trace gas and aerosol emissions (Andreae and Merlet, 2001) and was adjusted to produce monthly outputs for all variables in the SEIB-DGVM. These improvements allowed us to simulate fire activity and aboveground biomass dynamics and spatiotemporally assess the projected burned biomass and its emissions for the 21st century in Siberia under representative concentration pathways (RCPs).

## 2 Methods

### 2.1 Study sites

Boreal forests represent the largest forest biome and one-third of global forest cover (De Groot et al., 2013) and play an important role in the atmosphere–land interactions of the global climate system (Randerson et al., 2006; Bonan, 2008). Geographically, boreal forests are found in Canada, Alaska, and Siberia, of which Siberia has the largest forested area. Siberia is largely covered by deciduous needleleaf conifers (Figure 1), which consist mostly of the larch species *Larix sibirica, L. decidua,* and *L. dahurica* (Abaimov et al., 1998), which are categorized as pyrophytic species, meaning that they require periodic fires to persist on the landscape (Kharuk et al., 2011). The Siberian land cover has changed very little over the last century (Ivanov et al., 2022), and the boreal forest covers approximately >15 million $km^2$, and contain a large amount of carbon that is comparable to the combined carbon storage in tropical and temperate forests (Dixon et al., 1994; Kasischke, 2000).

The main external factors affecting Siberian boreal forests are fires and climate change (Goldammer and Furyaev, 1996; Shorohova et al., 2009). Climate change increased the frequency of forest fires, which in turn amplified the impacts of

climate change locally. In the Arctic, a rapid warming trend has been observed, and the increase in temperature over the last 20 years of the 20th century was 2 to 3 times higher than the global average, while in the first 20 years of the 21st century, it exceeded four times (Chylek et al., 2022). This enormous increase in temperature in Siberia, affecting the duration and speed

of snowmelt and accelerates thawing of carbon-rich permafrost (Natali et al., 2019; Schuur et al., 2015; Nitzbon et al., 2020), which results in drier ground cover, an increased frequency of wildfires, longer fire seasons, and increased ignition sources (Kharuk et al., 2022). These changes may result in a new climate state in which heatwaves as well as the associated the occurrence of wildfires may become routine and more severe (Hantemirov et al., 2022; Landrum and Holland, 2020). Produced emissions from thawing permafrost and from wildfire are likely to feed into the global carbon cycle's feedback on

climate change (Schuur et al., 2015), and triggering further warming trends globally (Schimel et al., 2001; Kharuk et al., 2011; Krylov et al., 2014).

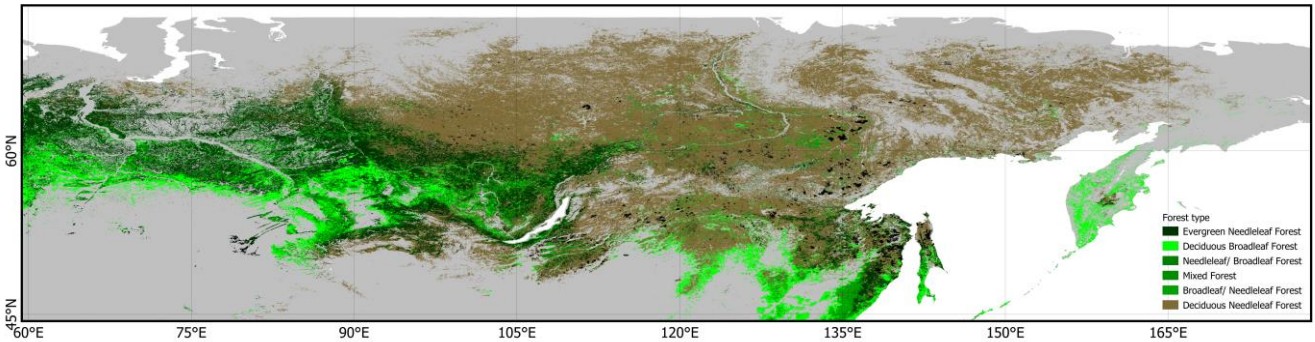

**Figure 1.** Study site (black rectangle: 60°-180°E and 45°-80°N). Green and brown color indicate the forest types in Siberia
are provided from Global Land Cover dataset (GLC 2000): Northern Eurasia (Bartalev et al., 2003). Grey color indicate
other vegetation types in the Siberian area provided by Database of Global Administrative Areas (GADM).

## 2.2 Improved fire module principles

We improved the SEIB-DGVM fire module by replacing the Glob-FIRM (Thonicke et al., 2001) with the SPITFIRE model
(Thonicke et al., 2010). First, we added two new input variables for fire ignition: population and lightning data. Second, we
incorporated the complete SPITFIRE equation (Thonicke et al., 2010), which included new variables, PFT parameters, and
local parameters, and improved the output to be able to be produced on a monthly scale (Figure 2). The variable integration
between the default and improved fire models requires several parameter-specific PFTs (Table 1).

The default SEIB-DGVM model uses annual time steps for vegetation dynamics and disturbance, which we improved to
monthly time step outputs. The fraction of individual trees killed by a fire depends on PFT fire resistance (M3, Table 1). All
grass leaf biomass, all dead and living tree leaf biomass, half of the dead tree trunk biomass, and half of the litter pool are
released into the atmosphere as $CO_2$ during a fire, while the dead tree's residual biomass is converted into litter. In reaction to
fire, all deciduous PFTs convert their phenology phase to dormancy, and if the stock resource of grass PFTs (gmassstock)

does not meet the minimal value (50 g DM $m^{-2}$) following fire, the deficit is supplemented from litter (Sato et al., 2007).
Furthermore, related to the fire-vegetation relationships, for herbaceous PFTs, both below-ground and storage biomass are
preserved after a wildfire and used for the recovery of above-ground biomass. During this recovery period, herbaceous PFTs
work on producing above-ground biomass while reducing their storage biomass, thus increasing the allocation ratio to
above-ground biomass in the post-fire phase. For woody PFTs, fire only gives the option for individual trees to either die or

165 survive. The surviving trees only lose their foliage biomass. As the foliage is lost, fine root biomass becomes unnecessary,
leading to its rapid loss due to its fast turnover rate. In the spring following a fire, surviving trees convert storage resources
into foliage and fine root biomass. The new net primary production (NPP) from the newly formed foliage first prioritizes the

recovery of leaves and fine roots. Therefore, fires increase the allocation ratio to the foliage and fine roots in surviving woody plants.

The basic equation of fire disturbance is the area burned, which we adjusted with the SPITFIRE equation (Thonicke et al., 2010) by including the fire probability and area of the grid cell:

$$A_b = P_b \times A \tag{1}$$

where $A_b$ is the area burned in a grid cell per month (ha month$^{-1}$), $P_b$ is the product of the probability of fire per month at any point inside the grid cell (month$^{-1}$) and $A$ is the area of the grid cell (ha). $P_b$ is the fire probability and is the product of the fuel load (litter + aboveground biomass) and its moisture factor. We used the same $P_b$ mechanism as that of the default fire

module, where if the fuel load satisfies the minimum fuel threshold (200 gC m$^{-2}$), random fires can occur at any point location inside the grid cells. In this improvement, $P_b$ was modified by considering the ignition event $E(n_{ig})$ (ha$^{-1}$) by anthropogenic (human population density) and natural (lightning strikes) ignition possibilities, the fire danger index (FDI), and the mean fire area $\bar{a}f$ (ha). Thus, Equation 2 can be represented as follows:

$$A_b = E(n_{ig}) \times FDI \times \bar{a}f \times A \tag{2}$$

Technically, the SEIB-DGVM simulation of each grid cell is carried out independently among the surrounding grid cell, so

the fire cannot spread to other grid cell without those grid cell meeting the ignition requirements (fuel load and fuel moisture).

After all variables in the SPITFIRE fire module were integrated, we added the trace gas and aerosol emission calculation process to the model. Trace gas and aerosol emissions estimation is referred to the Fire Modeling Intercomparison Project (FireMIP) protocols (Li *et al.*, 2019), the comprehensive study comparison of nine dynamic global vegetation models

(DGVMs) and produced important estimation for long-term and large-scale fire emissions. By using FireMIP protocol reference, SEIB-DGVM SPITFIRE improved to output PFT-level fire emissions.

Trace gas and aerosol emissions are the result of the total amount of burned biomass, the sum of dead and live fuel consumption as the result of surface fire and crown scorch. Trace gas emissions are estimated based on fire carbon emissions, vegetation characteristics from DGVMs, and fire emissions factors. Fire emissions of trace gas and aerosol for

each species $i$ and PFT $j$, $E_{i,j}$ (g species m$^{-2}$) are estimated based on Andreae and Merlet, (2001):

$$E_{i,j} = EF_{i,j} \times \frac{CE_j}{C} \tag{3}$$

Where $EF_{i,j}$ is PFT specific emission factor (g species (kg DM)$^{-1}$), $CE_j$ is the combusted biomass of $PFT_j$ due to the fire (g C m$^{-2}$), and $C$ is the unit conversion factor from carbon to dry matter, $C = 0.5 \times 10^3 gC(kg\,DM)^{-1}$. The Emission factors *(EF)* used in this study are based on Andreae and Merlet, (2001), and the updated pyrogenic emissions species by various types of biomass burning (Andreae, 2019) (Table S2 in the Supplement).

DGVMs generally simulate vegetation as a combination of PFTs in a given grid location to represent plant function at a global scale, instead of land cover types (Li *et al.*, 2019). In this, we classified the PFTs with the land cover types (LCTs) to

integrate the emission factors of each LCTs for trace gas and aerosol emissions estimation process. The BoNE, BoNS, and BoBS PFTs are classified as Boreal Forest LCTs. Other PFTs have been integrated with LCTs but are not listed in Table 1, as this study only covers Boreal Forest. The integration of other PFTs includes TrBE and TrBR, which are classified as Tropical Forest; TeNE, TeBE, and TeBS, which are classified as Temperate Forest; and TeH and TrH, which are classified as Grassland/Savanna. SEIB-DGVM didn't classify crop PFTs, so cropland LCTs will not be used.

Further changes in the input and output of the new SEIB-DGVM SPITFIRE model are shown in Appendix A.1, while Appendix A.2 summarizes the improvement processes represented in this study, which can be classified into two groups: disturbance and biogeochemical dynamics. Appendix A.3 lists the symbols used in the model's equations. Detailed information about the integration of the SPITFIRE module in the SEIB-DGVM, which includes the improvement and adjustment of all the variables and the main important variables, such as ignition events $E(n_{ig})$, fire danger index (FDI), mean fire area $\bar{a}f$, fuel moisture content, rate of spread, fire fraction and intensity, fire damage to plants, and trace gas and aerosol emissions, is provided in the Supplemental File (2.2.1-2.2.7).

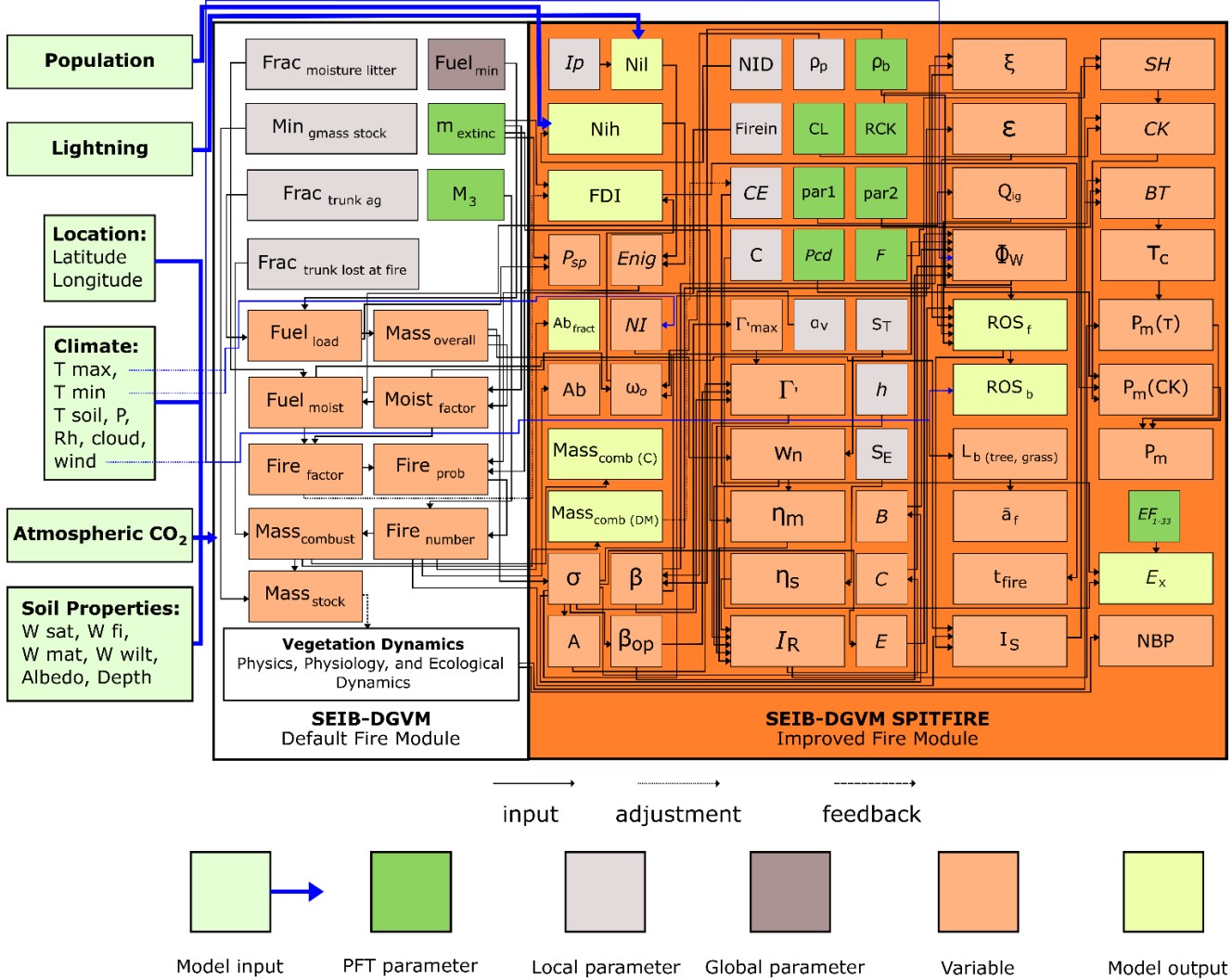

**Figure 2.** SEIB-DGVM SPITFIRE systems diagram. Describing the improvements (SPITFIRE), the interaction between the previous fire module (Glob-FIRM). All original SPITFIRE variables were integrated: ignition factor (lightning and population), PFT parameters, and other fire-related variables. In addition to the default annual output, the improved module had monthly outputs of all variables depending on the user needs. For the meaning of abbreviations, refer to the Appendix 215 A.3.

Table 1. SEIB-DGVM SPITFIRE Plant Functional Type (PFT)-specific model parameter values and their attribution to LCTs. This table was modified from Thonicke et al. (2010).

| PFTs | Land Cover Types (LCTs) | Fuel bulk density (kg m$^{-3}$) | | Scorch height parameter | | Crown length parameter | Bark thickness parameters | | | Crown damage parameter | | Fire resistance |
|------|------|------|------|------|------|------|------|------|------|------|------|------|
| | | $Pb$ | Reference | $F$ | Reference | $CL$ | par1 | par2 | Reference | $R(CK)$ | $p$ | $M3$ |
| BoNE | | 25 | (Miller and Urban, 1999; Hély et al., 2000) | 0.11 | (Hély et al., 2003) | 1/3 | 0.0292 | 0.2632 | (Reinhardt et al., 1997) | 1 | 3 | 0.12 |
| BoNS | Boreal Forest | 22 | (Keane et al., 1990) | 0.094 | (Dickinson and Johnson, 2001) | 1/3 | 0.0347 | 0.1086 | (Reinhardt et al., 1997) | 1 | 3 | 0.12 |
| BoBS | | 22 | (Keane et al., 1990) | 0.094 | (Dickinson and Johnson, 2001) | 1/3 | 0.0347 | 0.1086 | (Reinhardt et al., 1997) | 1 | 3 | 0.12 |

PFTs attributed to land cover types (LCTs) are needed to classify the fire emission factor (*EF*) (Table S2 in the Supplement) to estimate trace gas and aerosol emissions (Andreae and Merlet, 2001).

**2.3 Model calibration**

We calibrate the improved model by using all of the benchmark datasets (Table 3). The calibration process is done sequentially for all of the major variables, from burned fraction, burned area, dry matter, aboveground biomass, burned

biomass emissions, and the forest ecology variables (Figure 3). The process is sequential because one variable is used for the calculation of another variable (such as burned fraction and burned area affecting aboveground biomass, forest structure, dry matter, and emissions). One calibration process is performed with multiple iterations until the output variable has similar numerical values and spatial distribution to the benchmark data, and the process is repeated for other variables once the previous variable has been calibrated.

## 2.4 Model application

The original SEIB-DGVM utilizes three computational time steps: a daily time step for all physical and physiological processes except for soil decomposition and tree growth, a monthly time step for soil decomposition and tree growth, and an annual time step for vegetation dynamics and fire disturbance (Sato et al., 2007). In this study, we improved the fire module to calculate natural and anthropogenic fire ignition factors (based on lightning flashes and population density) and adjusted it to produce monthly outputs using temporal resolution statistical downscaling methods with user-defined weighted monthly parameters (Table 2). The annual average ignition factor variables (population density and lightning flash rate) were used consistently throughout all simulation phases.

We ran the improved model (SEIB-DGVM SPITFIRE) and the default model (SEIB-DGVM GlobFIRM) under the same protocols to equally compare and assess their fire products (Figure S3 in the Supplement)[*]. Simulations were run in three phases (spin-up, historical and future) and the simulation was run with the fire mode on and fire mode off to compare and assess the vegetation products during fire, and also each phase was replicated 5 times to minimize bias due to random variables in the tree morality[1]. The model was run in three phases[2]: 1) a 1000-year spin-up phase to bring the soil and vegetation carbon pools into equilibrium with the climate using daily baseline CRU TS3.22 climate data, 2) a 156-year historical phase also using daily baseline CRU TS3.22 climate data and spin-up simulation results as inputs, and 3) a 95-year future phase using daily MirocAR5 base V3 RCP8.5, RCP6.0, RCP4.5, and RCP2.6 climate data and historical simulation results as inputs (Figure 3). The MirocAR5 Base V3 dataset has been bias-corrected with CRU TS3.22 climate data, so using these two datasets consecutively in spin-up, historical, and future simulations ensures the harmony of the input climate data. Five different types of RCP scenario climate data were used to determine the impact of fire and climate on forest structure and their interactions.

In the previous SEIB-DGVM study, a 2000-year spin-up was needed to obtain the convergence amount of soil organic matter (Sato et al., 2010). However, we have conducted preliminary simulations with the same study area by setting the spin-up years to 1000 years and 2000 years. We confirmed that the outputs of the 1000-year and 2000-year spin-up simulations were very similar; thus, the 1000-year spin-up was enough to reach carbon stock equilibrium. This parameter setting is also in line with the simulation settings in other SEIB-DGVM studies: Sato et al. (2007) performed a 1000-year spin-up and combined it with all of the simulation phases to extract general trends of postfire succession. Another study by Arakida et al. (2021) also confirmed that a spin-up period of 100 years was sufficient for the equilibrium of the LAI, aboveground biomass, and GPP at all the study sites in Siberia.

In addition, we have verification stage[3] to ensures that the new input data can be read, produced, and processed properly (Rabin et al., 2017). Then, we calibrate all of the major emissions individually and sequentially with the benchmark dataset because each variable affects other variables, and we need to ensure the final output is comparable with the benchmark datasets[4]. After verifying that the new module was incorporated seamlessly, we validated the model outputs (fire, vegetation and emissions variables) by using GFED4, GFED4s, ESA Biomass CCI and GBEI benchmark datasets[5].

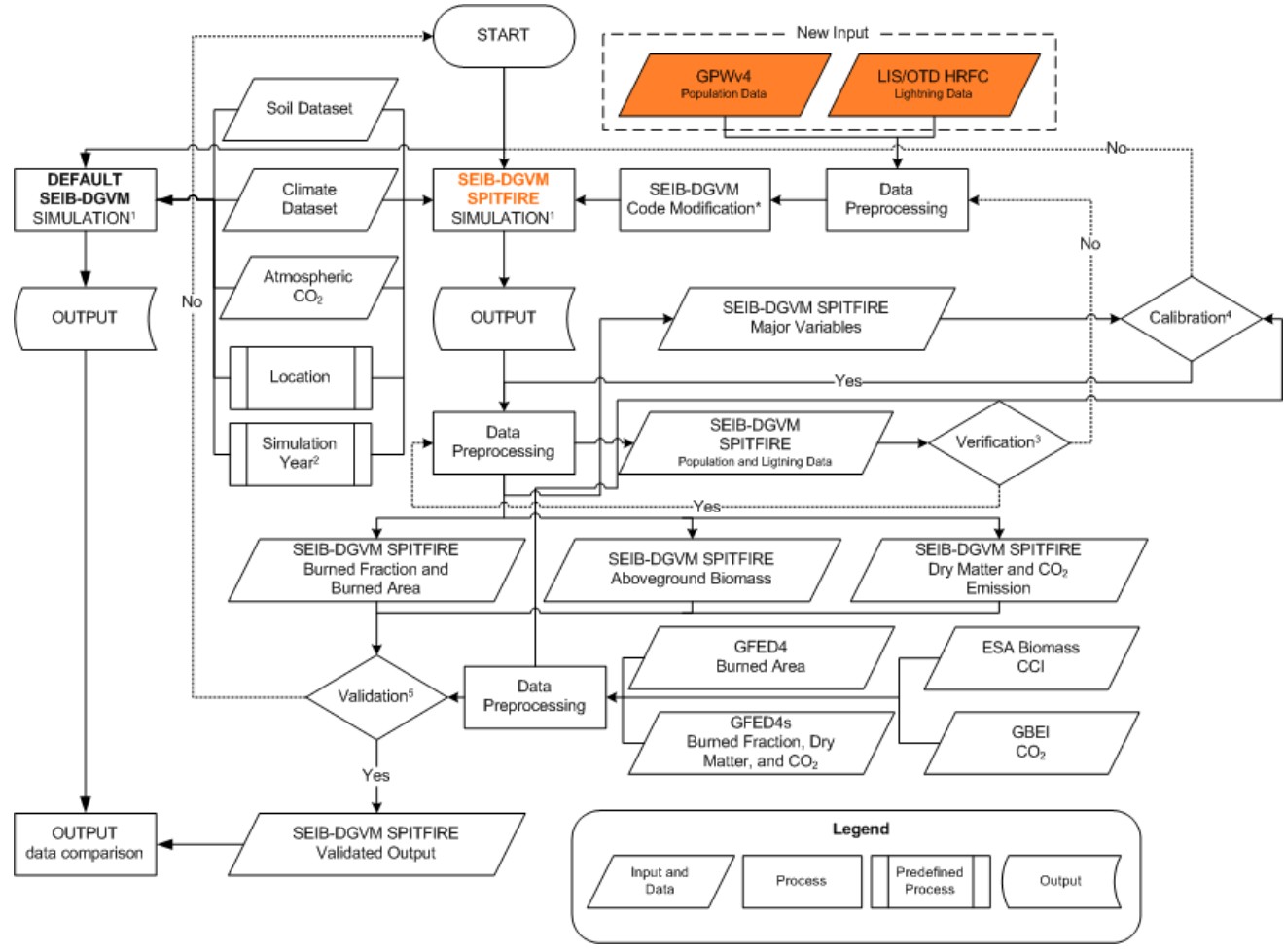

**Figure 3.** Workflow of improving the SEIB-DGVM fire module

Table 2. SEIB-DGVM SPITFIRE input data descriptions

| Model Input | Product | Variable | Spatial Resolution | Temporal Resolution | Temporal Coverage | Reference |
|---|---|---|---|---|---|---|
| **Climatic data** | CRU TS3.22 High-Resolution Gridded Data of Month-by-month Variation in Climate | Cloud cover, diurnal temperature range, frost day frequency, PET, precipitation, daily mean temperature, monthly average daily maximum and minimum | 0.5 degree | monthly | 1901–2013 | (NCAS, 2014) |

| Model Input | Product | Variable | Spatial Resolution | Temporal Resolution | Temporal Coverage | Reference |
|---|---|---|---|---|---|---|
| | MirocAR5 base daily V3 (RCP8.5, RCP6.0, RCP4.5, and RCP2.6) | temperature, vapor pressure, and wet day frequency<br><br>Air temperature, soil temperature, fraction of cloud cover, precipitation, humidity, and wind velocity | 0.5 degree | daily | 1850-2100 | |
| $CO_2$ | - | Global atmospheric Carbon dioxide concentrations ($CO_2$) | - | - | 1850-2100 | - |
| Soil properties | Global Soil Wetness Project 2 | Soil moisture at saturation point, field capacity, matrix potential, wilting point, and albedo | 1 degree (360 x 180) | time-fixed | time-fixed | www.iges.org/gswp |
| Ignition factors | LIS/OTD High-Resolution Full Climatology (HRFC) V2.3.2015 | Lightning flash rate | 2.5 arc-minute | Annual | 2000-2020 | (CIESIN, 2018) |
| | Gridded Population of the World (GPWv4) | Population density | 0.5 degree (720 x 360) | Annual | 2015 | (Cecil and Daniel, 2001) |

## 2.5 Model benchmarks

A common method for validating the outputs of dynamic global vegetation models (DGVMs) is to use satellite-based product datasets. For instance, direct observations of global fire occurrence by satellite-borne sensors can detect active fires, fire radiative power, and burned areas, and these observations have been available since the 1990s (Mouillot et al., 2014). The Fire Modeling Intercomparison Project (FireMIP) also used the satellite-based product database as a benchmark to evaluate the model simulation (Rabin et al., 2017; Li et al., 2019).

In the last few decades, several global biomass burning emission datasets based on burning area and fire radiative energy detection have been developed and used for many purposes, such as global climate and vegetation modeling, together with environmental, health, and security assessments (Ichoku et al., 2008; Mouillot et al., 2014). Although fire-related observation datasets are available and globally accessible, they have relatively large uncertainties and are poorly constrained, especially in models at the global and regional levels (Liousse et al., 2010; Petrenko et al., 2012, 2017; Bond et al., 2013; Zhang et al., 2014; Pan et al., 2015; Pereira et al., 2016).

Pan et al. (2020) reported that this uncertainty could be caused by various measurement and/or analysis processes, including the detection of fire or burned areas, retrieval of fire radiative power, emission factor information, biome type, burning stage, and fuel consumption estimation. The emission factor (EF) is considered an important factor for obtaining specific gaseous or particulate species of smoke emitted from burned dry matter in all major burned biomass (BB) emission datasets. Some EFs originate from laboratory experiments where fuel samples are burned in combustion chambers (Christian et al., 2003; Freeborn et al., 2008), whereas others originate from large-scale, open biomass burning and wildfire experiments. The combustion properties might differ greatly between these two categories; e.g., because of personnel security and other logistical considerations, some EF measurement locations are often not close enough to the biomass-burning source (Aurell et al., 2019). Another factor is the biome type, which affects the scaling factor of the emission coefficient for the FRP-based BB datasets (GFAS, FEER, and QFED). The emission factors of all BB datasets were assigned based on the type of biome, and most of the examined BB datasets had different definitions of major biome types, so uncertainty might be present at certain levels (Pan et al., 2020).

We validated the improved SEIB-DGVM fire module products by using the burned area (GFED4) and burned fraction (GFED4s) datasets, corresponding to the model's output. These datasets have higher resolutions than other burned area-based datasets, and all of the uncertainty probabilities regarding the selected database described by Pan et al. (2020) were adjusted with our model configurations. We used the emission factor (EF) from Andreae and Merlet (2001) with the latest update from Andreae (2019) and integrated the Plant Functional Types (PFTs) model with the land cover types (LCTs) used in the EF (Table 1 and Table S2 in the Supplement).

Furthermore, fire models should be evaluated together with their associated vegetation models because the former might produce burned areas perfectly but incorrectly simulate aboveground biomass (AGB) patterns. Fire products depend on AGB availability, and fire also affects AGB availability and succession after forest fires. Thus, to ensure that the model conducted

correct assessments, we evaluated the aboveground biomass variable using the ESA Biomass Climate Change Initiative dataset (Table 3). The AGB data from the ESA Biomass Climate Change Initiative (CCI) v.3 (2010,2017, and 2018) include high-quality data with a large resolution of 100 m × 100 m obtained from multiple remote sensing observations collected around the year 2010 (Santoro et al., 2021), making them suitable for validating our improved model product.

     Overall, we validated the model spatially and numerically at Siberian level and in smaller regions, to determine the
performance of the model in many points of view (spatial, numeric, wide and small region). We classified Siberia into three regions: west region (60º-90ºE and 45º-80ºN), central region (90º-120ºE and 45º-80ºN), and east region (120º-180ºE and 45º-80ºN) (Figure S12).

Table 3. Description of the observational datasets used for model evaluation

| Type | Variable | Unit | Source | Spatial resolution | Temporal resolution | Temporal coverage | Reference |
|---|---|---|---|---|---|---|---|
| **Fire** | Burned area | Hectares | Global Fire Emissions Database, Version 4.0 (GFED4) | 0.25 degree | Monthly, Annual | 1996-2016 | (Giglio et al., 2013) |
| | Burned fraction | - | Global Fire Emissions Database, Version 4.1 (GFED4s) | 0.25 degree | Monthly, Annual | 1997-2016 | (Giglio et al., 2013) |
| | Dry matter | kg $DM^{-1}$ $m^{-2}$ | | | | | |
| | $CO_2$ emissions | g $CO_2$ $year^{-1}$ | | | | | |
| | $CO_2$ emissions | g $CO_2$ $year^{-1}$ | Global Biomass Burning Emissions Inventory (GBEI) | 1-degree | Annual | 2001-2020 | Shiraishi et al., (2021) |
| **Vegetation** | Above-ground biomass | Mg $hectares^{-1}$ | ESA Biomass Climate Change Initiative (Biomass CCI): Global datasets of forest above-ground biomass for the years 2010, 2017 and 2018, v3 | 100 m | Annual | 2010, 2017-2018 | (Santoro and Cartus, 2021) |

## 3 Results

### 3.1 Improved model validation

#### 3.1.1 Fire products

We compared the annual average distribution patterns of burned fraction variable (1997-2016) in the SEIB-DGVM SPITFIRE and GFED4s data, and most patterns differed only in eastern Siberia (Figure 4, Figure S10). Compared to the burned fraction variable, burned area GFED4 has a smaller distribution pattern because it does not consider small fires (Figure S9.a). Comparison analysis of burned fraction variables between SEIB-DGVM SPITFIRE and GFED4s showed a linear relationship with a correlation coefficient of R=0.87 ($R^2$=0.75) (Figure S11.a). Similar to the comparison with GFED4s, the comparison of SEIB-DGVM SPITFIRE output of burned area variables with GFED4 data (1996-2016) shows a linear relationship with a correlation coefficient of R=0.78 ($R^2$=0.61) (Figure S11.b). Furthermore, in the three regions (west, central and east), the partial comparison of the burned fraction variable with GFED4s showed values of $R^2$=0.68, $R^2$=0.51, and $R^2$=0.58 (Figure S13), while for the burned area variable showed values of $R^2$=0.51, $R^2$=0.54, and $R^2$=0.506 (Figure S14), respectively. The burned fraction correlated better because both the GFED4s and the model's fire module considered small fires; many scattered fire data with values less than 0.1 and approximately 0.1 were found in both the model's output and the GFED4s data.

The fire products (burned fraction and burned area) in the improved model have the same spatial distribution because they are calculated based on one core variable (fire probability) (Eq. 1). However, the spatial distributions of GFED4s (burned fraction) and GFED4 (burned area) differ for two reasons: first, because GFED4 does not consider small fires (Giglio et al., 2013) while GFED4s does, and second, because GFED4s use the modified burned fraction equation, which is able to calculate the exact fire fraction and fuel load (not uniformized) in a grid cell (Van Der Werf et al., 2017).

Although the spatial distributions and patterns of the fire products (burned fraction and burned area) in the model and benchmark datasets (GFED4s and GFED4) data slightly differed, the model was able to produce annual mean value data that were similar to both benchmark datasets. The mean average burned fraction during 1997-2016 was 0.0137 in the simulations, compared to the GFED4s, which recorded the same value of 0.0137 with an RMSE value of 7.2 x $10^{-4}$. Furthermore, the mean average burned area of the model in 1996-2016 was 1428.5 ha grid$^{-1}$ year$^{-1}$, compared to the GFED4 burned area data, which closely recorded value of 1425.1 ha grid$^{-1}$ year$^{-1}$ by an RMSE value of 70.2 ha grid$^{-1}$ year$^{-1}$. In summary, the model was able to produce mean average data that precisely resembled observational data.

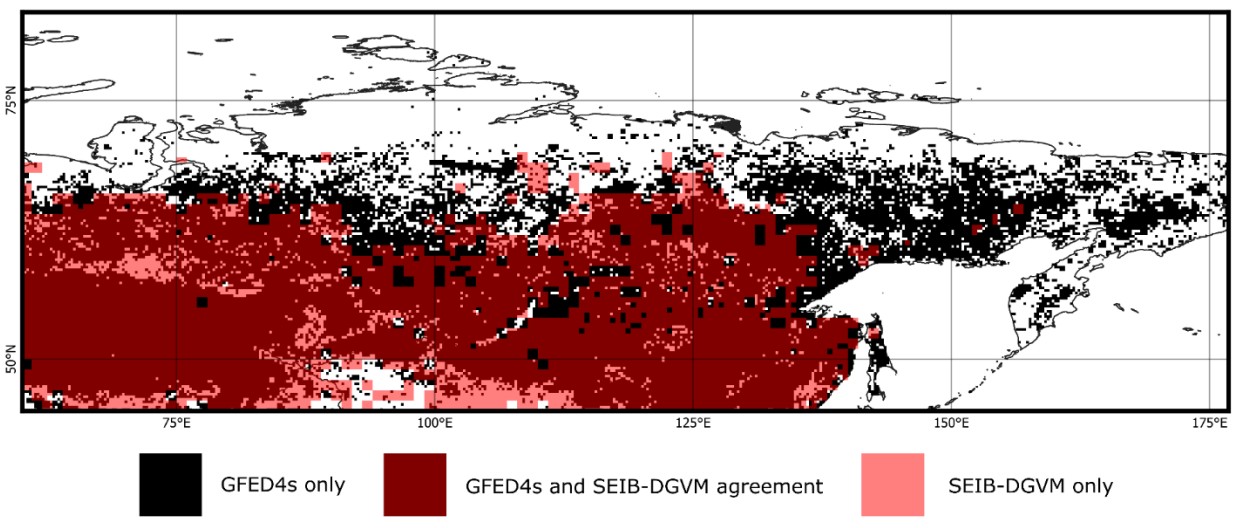

**Figure 4.** Spatial distribution comparison of annual averaged burned fraction variable (1997-2016) of SEIB-DGVM SPITFIRE and GFED4s

### 3.1.2 Aboveground biomass

The improved model simulated similar aboveground biomass values to those of the benchmark data. In 2010, 2017, and 2018, the simulations predicted $63.714 \pm 64.89$ Mg DM ha$^{-1}$ year$^{-1}$, $64.141 \pm 65.54$ Mg DM ha$^{-1}$ year$^{-1}$, and $64.313 \pm 65.61$ Mg DM ha$^{-1}$ year$^{-1}$, respectively, while the ESA Biomass CCI data showed $64.027 \pm 56.95$ Mg DM ha$^{-1}$ year$^{-1}$, $64.548 \pm 54.69$ Mg DM ha$^{-1}$ year$^{-1}$, and $65.05 \pm 55.78$ Mg DM ha$^{-1}$ year$^{-1}$, respectively, for the same years. The annual average AGB of the model in these years also showed the same increasing trend as that of the benchmark data, and the spatial distributions of the AGB model under CRU TS3.22 climate data and ESA Biomass CCI also agreed, with values of 83%, 85%, and 85%, respectively (Figure S15 and Figure S16 in the Supplement). Furthermore, when viewed on a smaller regional scale, the model is able to project better values in the western, central and eastern regions, with average values of $R^2=0.73$, $R^2=0.69$, and $R^2=0,74$, respectively (Figure S17). Although there was an annual average increase in the number of forest fires, there was a high variability trend in the model AGB values, indicating succession after forest fires and respond correctly to climate inputs variables based on each RCP scenario (Figure 9.d).

### 3.1.3 Annual and seasonal fluctuations in burned dry matter

The model's dry matter data have a spatial distribution pattern similar to that of the model's fire products (burned fraction and burned biomass), as calculated from the available fire and fuel load data (fire product derivatives). The annual average dry matter variability from the 1997–2016 model (under the historical climate product [CRU TS.3.22]) and the GFED4s data agreed with 6.24%, similar to the agreement of the fire products (Figure S20). Spatial comparisons at the regional scale in the western, central and eastern regions of Siberia show lower values than the Siberian region as a whole, which has an agreement of 60.2%, 64.4%, and 58.8% (Figure S21).

We also compared seasonal dry matter data to ensure that the monthly outputs of the SEIB-DGVM SPITFIRE model agree with the observations, as this difference influences seasonal aerosol emissions. Between 1997 and 2016, the GFED4s data exhibited high fluctuations/dynamics depending on the month and year, while the SEIB-DGVM SPITFIRE was not able to reproduce these dynamics or accurately predict the occurrence of extreme events (Figure S18a). For example, intense forest fires were recorded in 2003, 2012, and 2016. The monthly burned dry matter data for these years peaked in 2003 in May and in 2012 and 2016 in July (Figure S18.b-d). Severe wildfires in 2003 were due to low precipitation, as total precipitation reached only 36.0 mm in the Buryatia Republic and 45.7 mm in the Chita Oblast between August 2002 and May 2003 (IFFN, 2003). While the 41-year average precipitation between August and May (1981-2022), in the Buryatia Republic was approximately 332.23 mm, and in the Chita Oblast was approximately 119.45 mm. Thus, the low precipitation in 2003 was an anomaly outside of the annual average range.

Furthermore, the improved model's monthly average burned dry matter in 2003, 2012, and 2016 was also lower compared to the GFED4s data. The burned dry matter values of the improved model were $58.64 \pm 5.86$ kg DM m$^{-2}$, $59.41 \pm 5.9$ kg DM m$^{-}$

$^2$, and 59.98 ± 5.99 kg DM m$^{-2}$, while the benchmark data showed values of 122.36 kg DM m$^{-2}$, 101.7 kg DM m$^{-2}$, and 69.95 kg DM m$^{-2}$, respectively.

However, considering the entire period from 1997 to 2016, not only during years with extreme fire events, the model was also able to reproduce similar average values for multiple years and time-series data. When comparing the monthly averages during 1997-2016, the model data yielded a value of 58.94 ± 5.89 kg DM m$^{-2}$, while the GFED4s data yielded 59.12 kg DM m$^{-2}$. The model is not yet able to reproduce the exact value at a specific time of year or month because it runs in a long-term phase and is not yet able to predict sudden natural and anthropogenic conditions (factors). Overall, the spatial distribution comparison of the monthly dry matter variables from GFED4s and SEIB-DGVM SPITFIRE for 20 years (1997-2016) revealed a correlation of 99% (Figure 5); therefore, the model was able to approximate the monthly averages.

### 3.1.4 Carbon dioxide (CO$_2$) and PM$_{2.5}$ emissions

Emissions from biomass burning contribute significantly to the global budget for residual gases and aerosols that affect the climate. It's estimated that biomass burning contributed up to 50% of global CO and NO$_x$ emissions in the troposphere (Galanter et al., 2000), and the most emitted gas during biomass burning is CO$_2$ (Ritchie et al., 2020). Since CO$_2$ emissions are the primary emissions that contribute to climate change, it is critical to assess and monitor them continuously.

In this study, out of 33 projected emissions (Table 4 and Table S6), we validated the CO$_2$ variable that able to represent all projected emissions because all estimated emissions are derived from the same burned dry matter variable, which differs only in the emission factor value of each gaseous emission. The highest annual average value of CO$_2$ emissions from 1997 to 2020 is from GFED4 data, followed by SEIB-DGVM SPITFIRE and then the GBEI product, with values of 105.64 ± 50.69 × 10$^{13}$ g CO$_2$, 76.12 ± 0.87 × 10$^{13}$ g CO$_2$, and 62.4 ± 26.09 × 10$^{13}$ g CO$_2$, respectively (Table S3). The GFED4s and GBEI data have higher standard deviation values than does the SEIB-DGVM SPITFIRE data and appear to have a large difference. Spatially, the annual average CO$_2$ emission model data were 61.3% (Figure 6.a) and 79.8% (Figure 6.b) correlated with the GFED4s and GBEI data, respectively. Furthermore, CO$_2$ emissions of the model compared to the GFED4s in the three regions (west, central, and east) showed lower agreement than Siberia as a whole, at 62.7%, 62.5%, and 61.6%, respectively (Figure S29). Whereas the comparison to GBEI data at the three regions, showed agreements of 74.7%, 77.6%, and 64.3%, respectively (Figure S30). In addition, spatial comparison of annual mean data over 95 years (2006-2100) from SEIB-DGVM SPITFIRE, GFED4s, and GBEI datasets reveals similar values of 141.1 ± 11.5 Gg CO$_2$ year$^{-1}$, 157.2 ± 14.8 Gg CO$_2$ year$^{-1}$, and 148.7 ± 7.12 Gg CO$_2$ year$^{-1}$, respectively

Our study area covers the Boreal Asia (BOAS) area and a small part of Central Asia (CEAS), differing from the GFED4 basis region classification; therefore, we extracted these areas from the GFED4s data for comparison (Figure S22). A comparison of the GFED4s CO$_2$ data between the BOAS area and the Siberian area showed that the two datasets had a similarity of 98.2% (Figure S26), confirming the accuracy of the GFED4s validation data.

As all emission products are derived from fire products (dry matter variables), emission factors displayed spatial and value dynamics similar to those of the fire products (Figure S19, Figure S27, and Figure S43). When comparing the annual average

dry matter emission data and $CO_2$ emissions generated by the model, the results correlated perfectly (100%, Figure S31),

indicating that the model runs well according to Equation (3) and the projected $CO_2$ and other emissions have the same distribution patterns as the dry matter variable, because all of the emissions calculation are based on the dry emission variable. However, they differ in their values because each emission species has a different emission factor.

We also compared the modelled $PM_{2.5}$ emissions and their distribution patterns with the Copernicus Atmosphere Monitoring Service (CAMS) (Romanov et al., 2022) data in seven Russian territories (Amur Region, Buryatia Republic, Irkutsk Region,

Khabarovsk Territory, Krasnoyarsk Territory, Transbaikal Territory, Yakutia (Sakha Republic)) during 2010-2021. The improved model data and CAMS data both exhibited an increasing trend (Figure 7.a and Figure 2 in Romanov et al. (2022)) and a correlation of 85.8% (Figure 7.b).

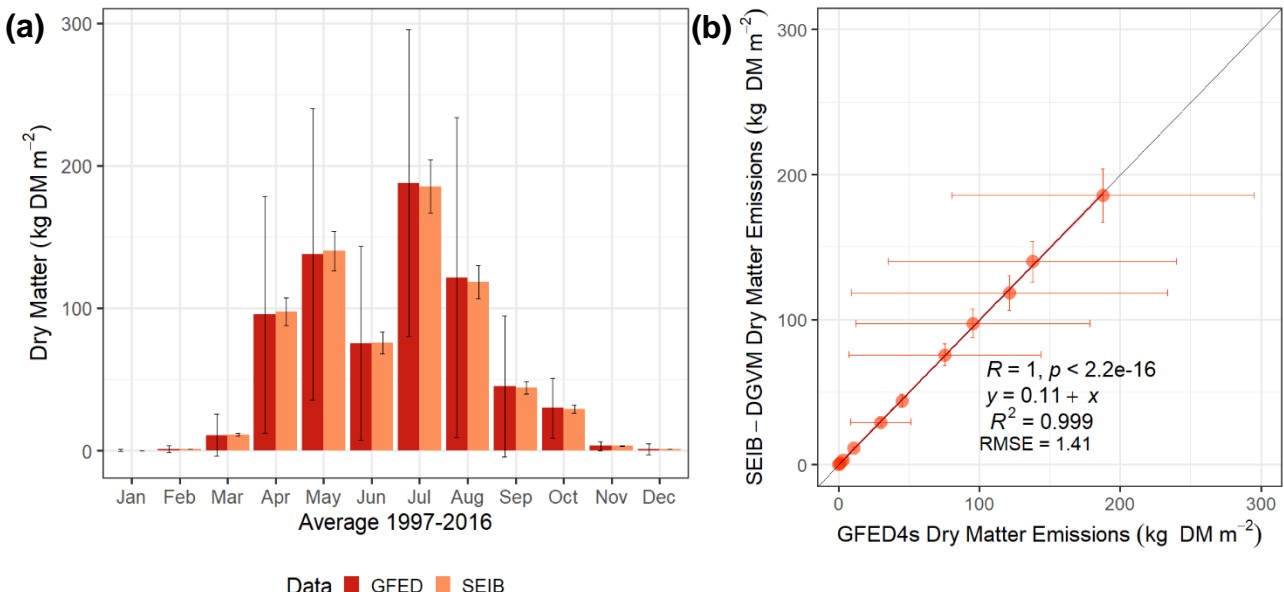

**Figure 5.** (**a**) Monthly temporal variability averaged dry matter emission of GFED4s and SEIB-DGVM from 1997 to 2016. (**b**) Comparison of monthly averaged dry matter emissions of GFED4s and SEIB-DGVM from 1997 to 2016. Standard deviation obtained from each monthly data from 1997 to 2016.

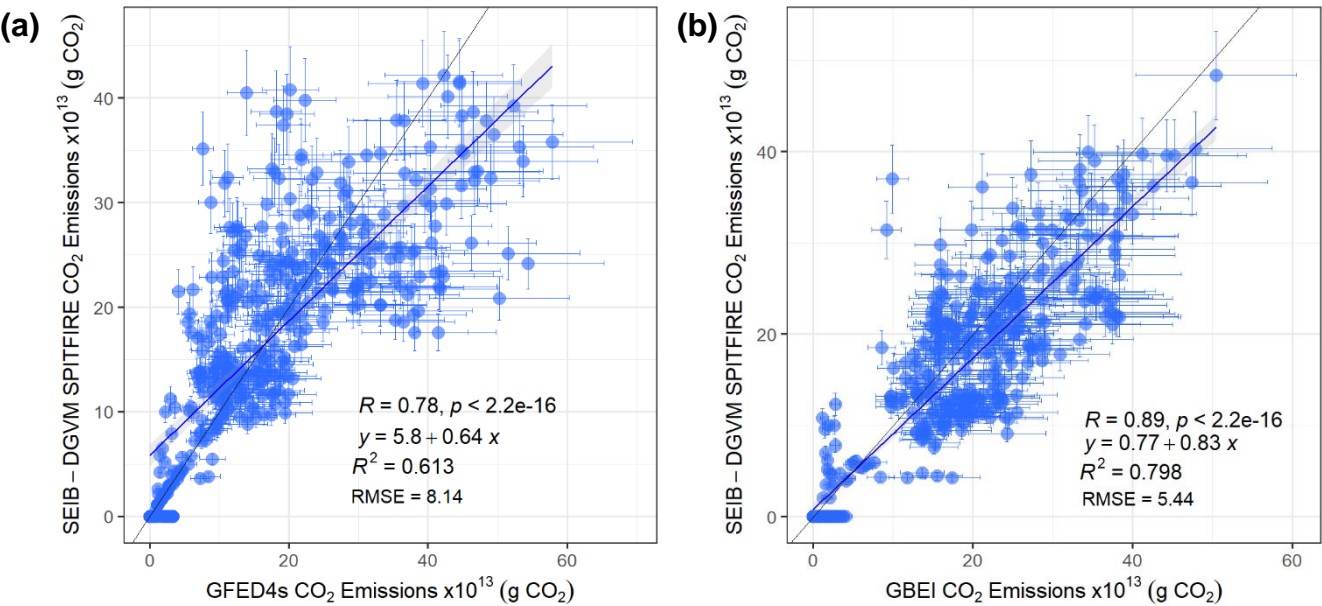

**Figure 6.** Latitude average spatial comparison of simulated $CO_2$ emissions of SEIB-DGVM SPITFIRE with GFED4s from 1997 to 2016 (**a**) and GBEI from 2001 to 2020 (**b**) dataset. Standard deviation obtained from the annual $CO_2$ emission data of each dataset.

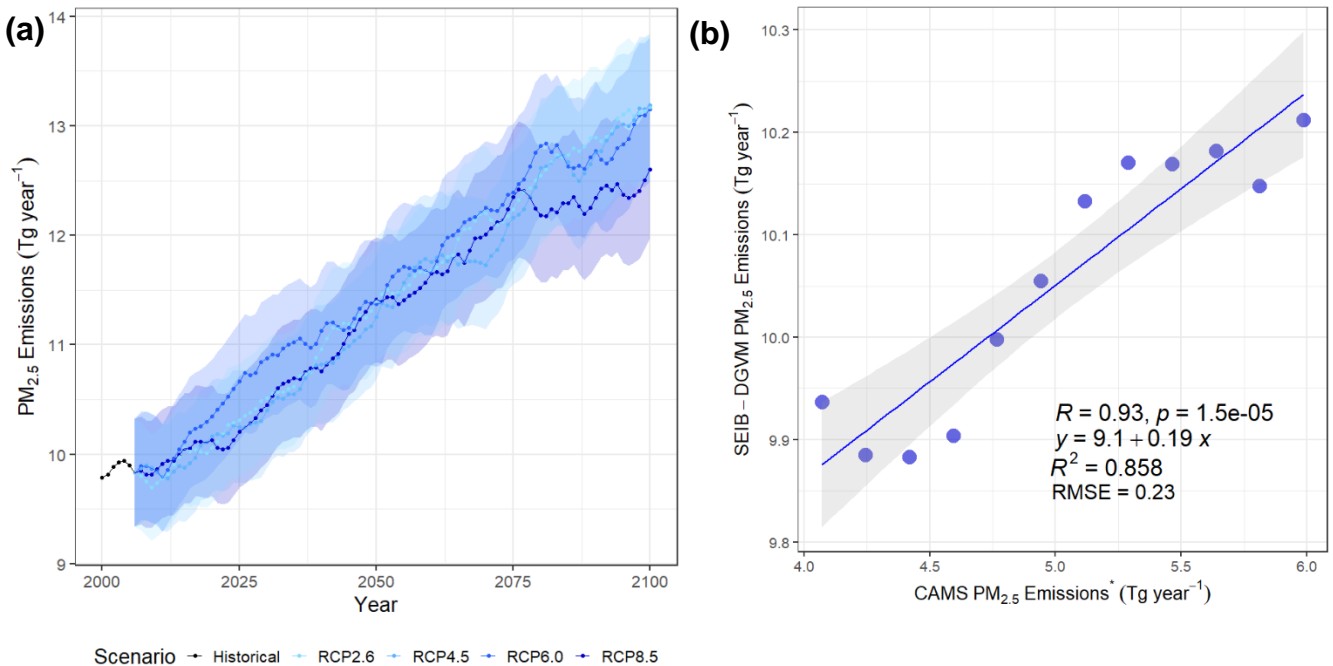

**Figure 7. (a)** Temporal variation of projected PM$_{2.5}$ emissions under several climate scenarios from 2000 to 2100. Standard deviation obtained from the annual PM$_{2.5}$ emission value of each climate scenarios (RCP8.5, RCP6.0, RCP4.5, and RCP2.6). **(b)** Comparison of PM$_{2.5}$ emissions from the SEIB-DGVM SPITFIRE model with the *trendline processed data from the Copernicus Atmosphere Monitoring Service in seven regions in Russia from 2010 to 2021 (Romanov et al., 2022).

### 3.2 Burned fraction

Improved model (SEIB-DGVM SPITFIRE) is able to produce burned fraction variables better than the default model (SEIB-DGVM GlobFIRM). Based on the spatial comparison of the 1997-2016 average burned fraction variables GFED4s, SEIB-DGVM SPITFIRE, and Default SEIB-DGVM, it is known that SEIB-DGVM SPITFIRE is able to produce data with a similarity of 75% with GFED4s data, while the default model is at 68% (Figure S5).

The burned fraction variable in the improved model exhibited a spatial distribution pattern different from that in the default model (Figure S4.a). According to the improved model, the burned fraction data were distributed in the western, central, and southern areas (Figure S4.b). We compared the burned fraction variable with the lightning flash rate and population density data to confirm that the produced variable considered the new ignition factor. The burned fraction showed a 46% correlation with the lightning flash rate and a 6% correlation with population density between 2006 and 2100 (Figure S7.a and b). In general, the burned fraction under all the RCP scenarios exhibited an increasing trend from 2006 to 2100, with the highest value occurring under the RCP4.5 scenario. Under the RCP4.5 scenario, the lowest value was 0.01371, and the highest value was 0.01427, with an average value of 0.01398 (Figure S4.d).

In contrast to the results produced from the improved model, the burned fraction data from the default model were spread throughout most of the area (Figure S4.a). From 2006 to 2100 under all RCP scenarios, the burned fraction in the default model also exhibited an increasing trend. Under the RCP4.5 scenario, the lowest value is 0.002996, and the highest value is 0.003113, with an average value of 0.00306 (Figure S4.c), which is well below the outputs of the improved model.

### 3.3 Burned area

The burned area of the improved model showed a similar spatial distribution pattern under all the RCP scenarios (Figure S6.a). The distribution pattern of the burned area variable was also similar to that of the burned fraction variable, as the burned area and burned fraction calculation processes are both based on fire probability (Eq. 1). Overall, under all the scenarios, the burned area exhibited the same increasing trend, with the RCP4.5 scenario reaching the highest value. Under the RCP4.5 scenario from 2006 to 2100, the burned area has an average value of 1945.9 ha grid$^{-1}$ year$^{-1}$ and is projected to increase with values of 79.7 to 83.8 x 10$^5$ hectares (Figure S6.b). Since the default model does not compute burned area, this variable could not be compared between the improved model and the default model.

### 3.4 Burned biomass

The improved model confirmed uniform spatial distribution patterns for the fire variables: burned fraction (Figure S4.b), burned area (Figure S6.a), and burned biomass (Figure 8.b). All of the improved module fire variables confirmed to be mutually integrated because the calculation process comes from the first fire variable (burned fraction). Compared to the improved model, the spatial distribution pattern of the burned biomass variable from the default model was wider and spread across the entire Siberia region (Figure 8.a). The spatial distribution pattern of burned fraction (S4.a-d) and burned biomass (Figure 8.a) in the default model is different and exhibited a box-like pattern in the center of the map. The internal model calculation flow relationship between the burned fraction and burned biomass variables in both the default and improved models shows a positive linear correlation, indicating harmony between these variables. A higher burned fraction corresponds to a higher burned biomass. The default model (SEIB-DGVM GlobFIRM) has an R² value of 0.83, while the improved model (SEIB-DGVM SPITFIRE) demonstrates better integration, with an R² value of 0.93 (Figure S8.a and d). Under all RCP scenarios from 2006 to 2100, the burned biomass variable in both the default and improved models exhibited an increasing trend (Figure 8.c and d). This indicates correct integration between the burned fraction and burned area variables, and an appropriate response to the climate input data. Furthermore, under the RCP6.0 climate scenario from 2000 to 2100, the burned biomass value in the default model increases from 50.4 to 60.6 kg DM m$^{-2}$ (Figure 8.dc), while in the improved model it increases from 53 to 73.98 kg DM m$^{-2}$ (Figure 8.d). The twenty-year variations and their trends of dry matter emissions up to 2100 in the improved model (SEIB-DGVM SPITFIRE) are 55.90 ± 1.31 (10.5 %), 60.52 ± 1.12 (11.4 %), 64.43 ± 1.36 (12.1 %), 69.23 ± 1.37 (13 %), 71.81 ± 0.94 (13.5 %) (Figure S32).

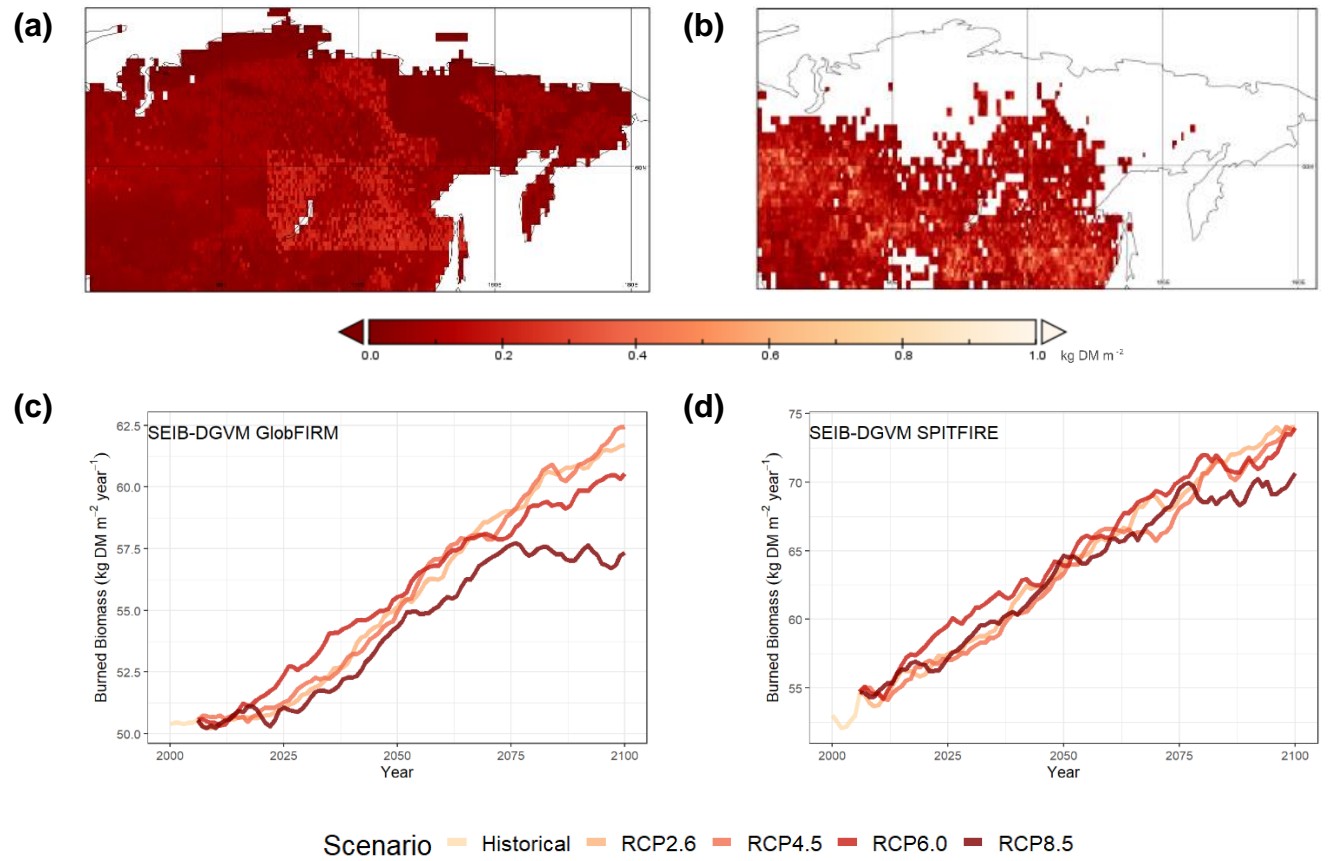

**Figure 8. (a)** Spatial distribution of annual averaged burned biomass of SEIB-DGVM GlobFIRM from 2000 to 2100. **(b)** Spatial distribution of annual averaged burned biomass of SEIB-DGVM SPITFIRE from 2000 to 2100. **(c)** Temporal variation of annual averaged burned biomass of SEIB-DGVM GlobFIRM from 2000 to 2100. **(d)** Temporal variation of annual averaged burned biomass of SEIB-DGVM SPITFIRE from 2000 to 2100.

### 3.5 Aboveground biomass

The aboveground biomass calculations in the default model and improved model used the same estimation process because the trunk biomass in the SEIB-DGVM included coarse root biomass; therefore, only approximately 2/3 of the trunk biomass was classified as aboveground biomass (Sato et al., 2007). However, during the calibration of the aboveground biomass variable with the ESA Biomass CCI benchmark dataset, we adjusted the calculation impact of fire and its distribution pattern (based on natural and anthropogenic ignition factors) on the availability of aboveground biomass.

According to the default model, the AGB distribution pattern appears to be the same as that of the fire variable, a box-like pattern still occurs on the map (Figure 9.a). Under the RCP8.5 scenario, from 2000 to 2100, the AGB increased from 63.72 to 120.1 Mg DM ha$^{-1}$, and the average value was 86.3 Mg DM ha$^{-1}$ (Figure 9.c). The aboveground biomass (AGB) variables in both the default and improved models exhibit an increasing trend and vary across RCP scenarios, with the highest values observed under RCP8.5 and the lowest under RCP2.6. This indicates that the models effectively read and process the RCP input climate data.

Compared to the default model, the improved AGB model has a bit difference in distribution patterns (Figure 9.b). In the central Siberian region, in some locations that have high AGB has been reduced due to the impact of forest fires, so that the box-like pattern is no longer visible (Figure 9.b). The temporal variation of aboveground biomass in the improved model also shows an increasing trend due to the warming scenario of each RCP climate data input. The AGB under the RCP8.5 scenario from 2000 to 2100 increased from 59.08 to 126.7 Mg DM ha$^{-1}$ (Figure 9.d), and the mean was 88.68 Mg DM ha$^{-1}$. The twenty-year variations and their trends of aboveground biomass up to 2100 are $65.45 \pm 1.19$ (10.8 %), $71.69 \pm 2.90$ (11.8 %), $83.38 \pm 3.61$ (13.7 %), $99.17 \pm 5.06$ (16.3 %), $117.92 \pm 5.41$ (19.4 %) (Figure S33).

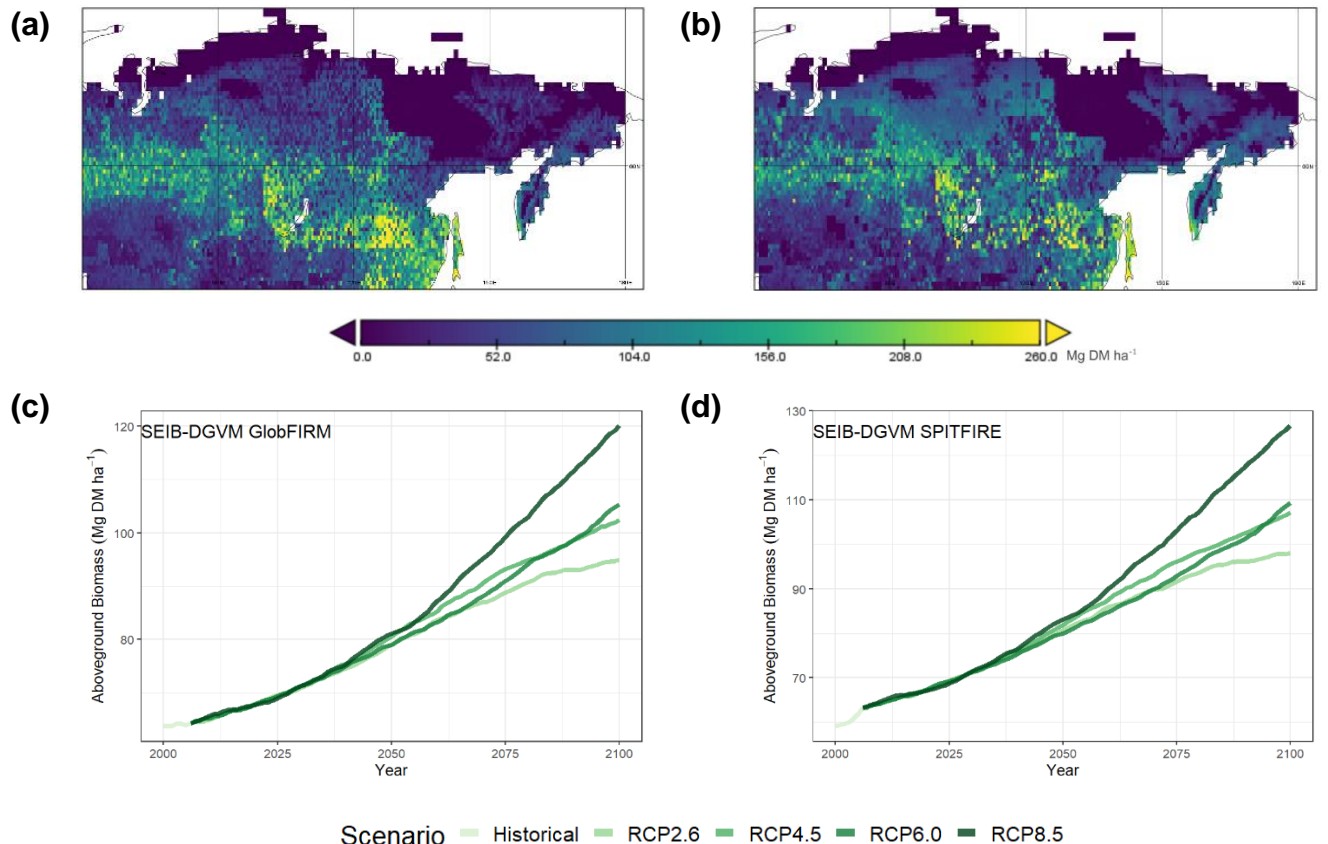

**Figure 9. (a)** Spatial distribution of annual averaged aboveground biomass of SEIB-DGVM GlobFIRM from 2000 to 2100. **(b)** Spatial distribution of annual averaged aboveground biomass of SEIB-DGVM SPITFIRE from 2000 to 2100. **(c)** Temporal variation of aboveground biomass of SEIB-DGVM GlobFIRM from 2000 to 2100. **(d)** Temporal variation of aboveground biomass of SEIB-DGVM SPITFIRE from 2000 to 2100.

## 3.6 Forest ecological variables under fire-on and fire-off simulation

We conducted complete simulations under fire-on and fire-off modes to compare and assess vegetation dynamics during forest fires. Assessing vegetation dynamics can be done by understanding the carbon pools in the certain region or globally, where carbon pools are easier to measure than carbon fluxes. In this study, the net primary production (NPP) is used as a reference variable because it is an important metric of the global carbon cycle (Running, 2022) and measures the rate of global plant growth. We obtained the NPP lost variable due to wildfire from fire-on and fire-off simulations. The NPP lost variable under all RCP scenarios shows a increasing trend. Under the RCP8.5 scenario, an average NPP loss of 385.19 ± 40.4 g C m$^{-2}$ year$^{-1}$ occurred during 2000–2100 (Figure S25.a). In addition to the NPP variable, the improved model (SEIB-DGVM SPITFIRE) can also simulate Net Biome Production (NBP). Under the same RCP8.5 scenario, the annual average NBP from 2000-2100 shows a positive value of 307.7 ± 43 Tg C year$^{-1}$ (Figure S25.b), with a continuous increasing trend.

In relation to wildfires, assessing pre- and postfire tree density variables is critical for measuring the impact of fires. Under the RCP8.5 scenario, in the fire-on simulation from 1997 to 2100, it is projected that the tree density in Siberia was 2,181 tree ha$^{-1}$. However, under the same RCP and time range in the fire-off simulation, the tree density was 2,363 tree ha$^{-1}$. We also compared the tree density between the fire-on and fire-off simulations under all the RCP scenarios and found that the tree density increased in the fire-off simulations compared to that in the fire-on simulations. Under the RCP8.5 scenario, on average, 174 trees ha$^{-1}$ year$^{-1}$ died due to the fire (Figure S25.c).

We also conducted a more detailed assessment of several forest structure variables, such as tree DBH, crown area, and tree height, from 2006 to 2100 under all the RCP scenarios. Under the RCP8.5 scenario, in the fire-on simulation, the results showed that tree DBH values varied from 0 to 4.7 m (average 0.9 m), tree height from 0 to 75.4 m (average 24.2 m), and crown area from 0 to 15.1 m$^2$ (average 5.7 m$^2$). The average tree structure in the fire-off simulation was greater than that in the fire-on simulation, with average tree DBH, tree height, and crown area of 0.97 m, 24.1 m, and 6.5 m$^2$, respectively. The correlations between the tree structure variables under fire-on and fire-off simulation conditions were similar and highly correlated; the overall average correlation among the tree DBH, tree height, and crown area variables was 97 % (Figure 10). Specifically, according to region classification, highest to the lowest value of tree height, tree DBH, and crown area value is in the west region, then centralregion, and east region. On average for 2081-2100 under RCP8.5 in each region, the tree height, tree DBH, and crown area variables show values of 28.43 ± 0.8 m, 1.1 ± 0.004 m, 5.7 ± 0.01 m$^2$ (west region), 28.3 ± 0.9 m, 1.2 ± 0.04 m, 7.8 ± 0.08 m$^2$ (central region), and 30.2 ± 1.0 m, 1.2 ± 0.06 m, 8.5 ± 0.2 m$^2$ (east region) (Figure S37, Figure S38, and Figure S39). Furthermore, we found an interesting pattern, the simulated tree allometry variables (tree height, tree DBH, and crown area) in eastern Siberia exhibit a greater range of values compared to those in central and western Siberia (Figure S37, Figure S38, and Figure S39). Overall, all tree allometry variables in Siberia exhibit an increasing trend, and the differences between fire-on and fire-off simulations for all tree allometry variables are most pronounced in eastern Siberia.

In addition, the relationship between the three variables (tree height, tree DBH, and crown area) in the west region and central region shows a linear trend where the higher the tree height, the greater the tree DBH and the wider the crown area (Figure 11). The east region shows an interesting pattern, different from other regions, where there is low tree (Figure 11.c). The western and central Siberia exhibit a greater range of tree height values compared to eastern Siberia (Figure 11.a and b). An interesting pattern was observed in western Siberia, where trees with high tree height and large DBH but low crown area were detected in some locations (Figure 11.a).

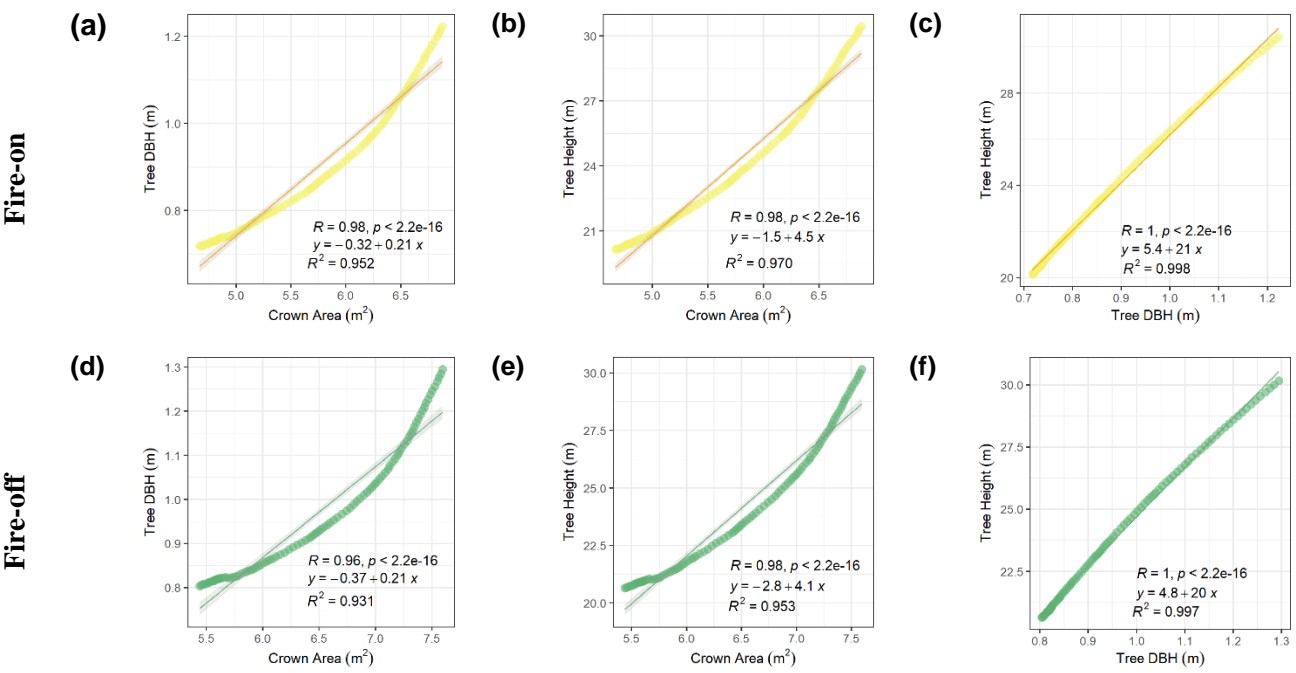

**Figure 10.** Annual average comparison of tree DBH, tree height and crown area variables in **(a-c)** fire-on and **(d-f)** fire-off simulations (2006-2100). Each point represents one grid latitude average of each variable.

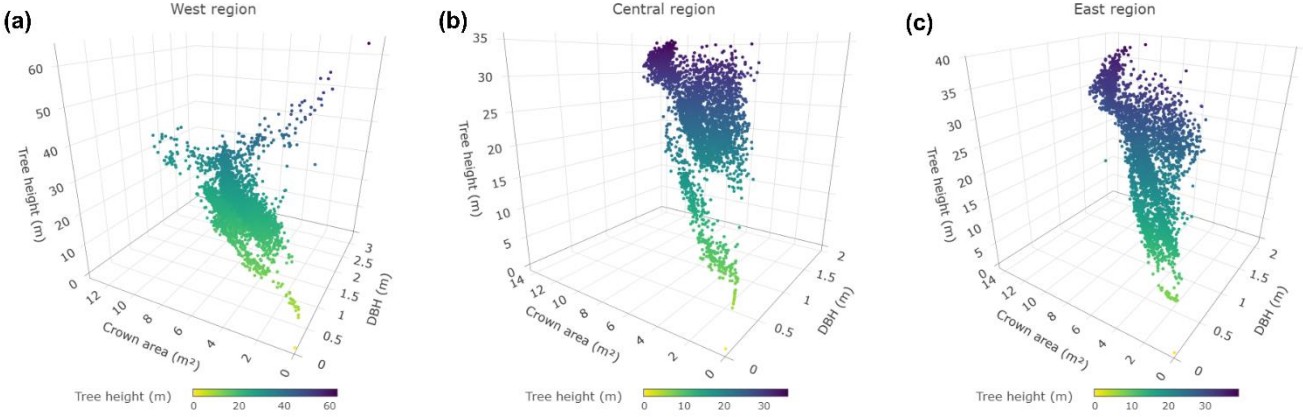

**Figure 11.** Relationships between simulated tree height, tree DBH, and crown area under fire-on simulation and RCP8.5 scenario from 2000 to 2100 in: **(a)** West region, **(b)** Central region, and **(c)** East region of Siberia.

### 3.7 Fire and AGB variable comparisons

We performed internal comparisons of fire and AGB variables within the improved model to ensure that the model worked properly and that the variable calculation processes were interrelated. Eastern Siberia had low fire patterns (Figure S23), and when compared with the AGB, this area also had very low AGB. We extracted the AGB data in the marked area with coordinates of 130-142°E and 65-80°N and discovered that the average simulated aboveground biomass in the area was 65.59 g C m$^{-2}$ from 1997 to 2023, compared to 416.4 g C m$^{-2}$ in the one-grid high-AGB areas. Furthermore, we assessed the fire danger index (FDI) variable in these low AGB areas and found that the mentioned region had a value of 0, indicating that it had a very low fire potential (Figure S24.a ).

We also compared the fire variables (burned fraction, burned biomass) and AGB variables between the improved model and the default model. According to the default model, the correlation between the burned fraction and burned biomass was 0.83, the correlation between burned fraction and AGB was 0.82, and the correlation between burned biomass and the AGB was 0.88 (Figure S8.a-c). According to the improved model, the correlation between the burned fraction and burned biomass was 0.93, the correlation between burned fraction and AGB was 0.96, and the correlation between burned biomass and the AGB was 0.9 (Figure S8.d-f ). Overall, both the default and improved models are well integrated, with the improved model demonstrating superior integration compared to the default model.

### 3.8 Future projection of burned biomass emissions

Our model projects that from 2000 to 2100, Siberia will produce $CO_2$ emissions of 10 to 11,000 x10$^8$ g $CO_2$ year$^{-1}$ (Figure 12). The distribution patterns of $CO_2$ and other emissions are similar because all emissions are calculated based on the same

variable dry matter emissions. Over the twenty-year period, we projected an increasing trend in $CO_2$ emissions across the various RCP scenarios, which aligns with the projected increase in forest fires through 2100.

The average from 2000 to 2100 shows that $CO_2$ emissions are highest under the RCP6.0, RCP2.6, RCP4.5, and RCP8.5 scenarios, with values of $885.8 \pm 75.4$, $877.82 \pm 82.6$, $871.4 \pm 80.6$, and $865.5 \pm 69.6$ Tg $CO_2$, respectively. Specifically under the RCP6.0 scenario, the highest projected emissions are expected in the periods 2021-2040, 2041-2060, 2061-2080, and 2081-2100, with Siberia producing $CO_2$ emissions of $769.24 \pm 14.37$, $830.52 \pm 15.61$, $877.93 \pm 16.34$, $940.46 \pm 20.59$, and $981.73 \pm 12.61$ Tg $CO_2$, respectively (Figure 13).

The highest gaseous species emissions were $CO_2$, CO, $PM_{2.5}$, TPM, and TPC, and all of them exhibited similar increasing trends from 2000 to 2100 under all RCP scenarios. Under the RCP6.0 scenario, these emissions are expected to increase by $2.58 \pm 0.75$, $0.21 \pm 0.06$, $0.03 \pm 0.01$, $0.02 \pm 0.01$, and $0.014 \pm 0.006$ Tg species year$^{-1}$, respectively. The increasing trend of emissions production until 2100 is also in line with the FDI variable, which shows the same increasing trend (Figure S24.b). Overall, by 2100, under the RCP6.0 scenario, the production of $CO_2$, CO, $PM_{2.5}$, TPM, and TPC emissions from forest

biomass burning combustion are projected to reach $1,009.00 \pm 75.44$, $80.74 \pm 6.04$, $12.60 \pm 0.91$, $9.54 \pm 0.69$, and $6.61 \pm 0.48$ Tg, respectively. The twenty-year averages of the $CO_2$, CO, $PM_{2.5}$, TPM, and TPC emission data under all the RCP scenarios are provided in Table 4, and the other twenty-eight emissions are provided in Table S6.

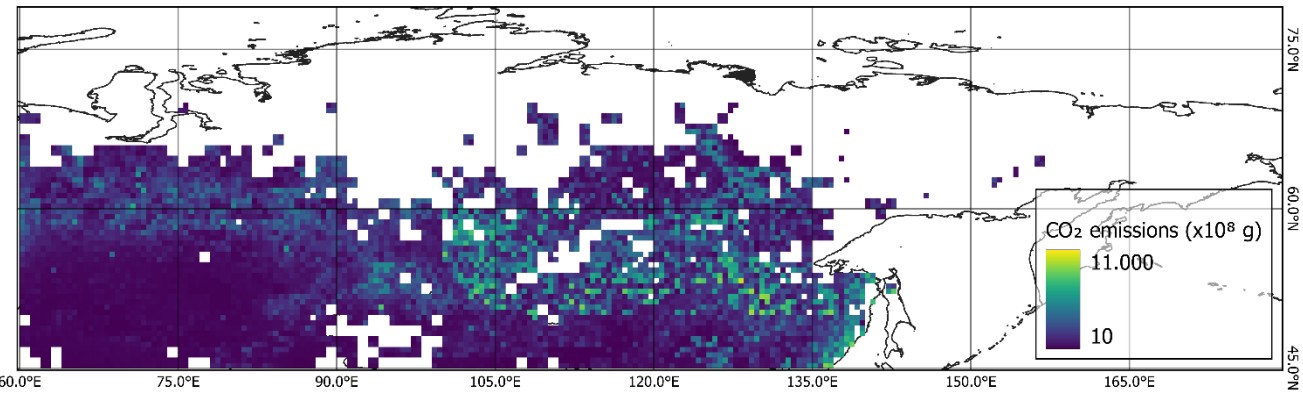

**Figure 12.** Spatial distribution of annual average projected $CO_2$ emissions (1996-2100) under RCP8.5 scenario.

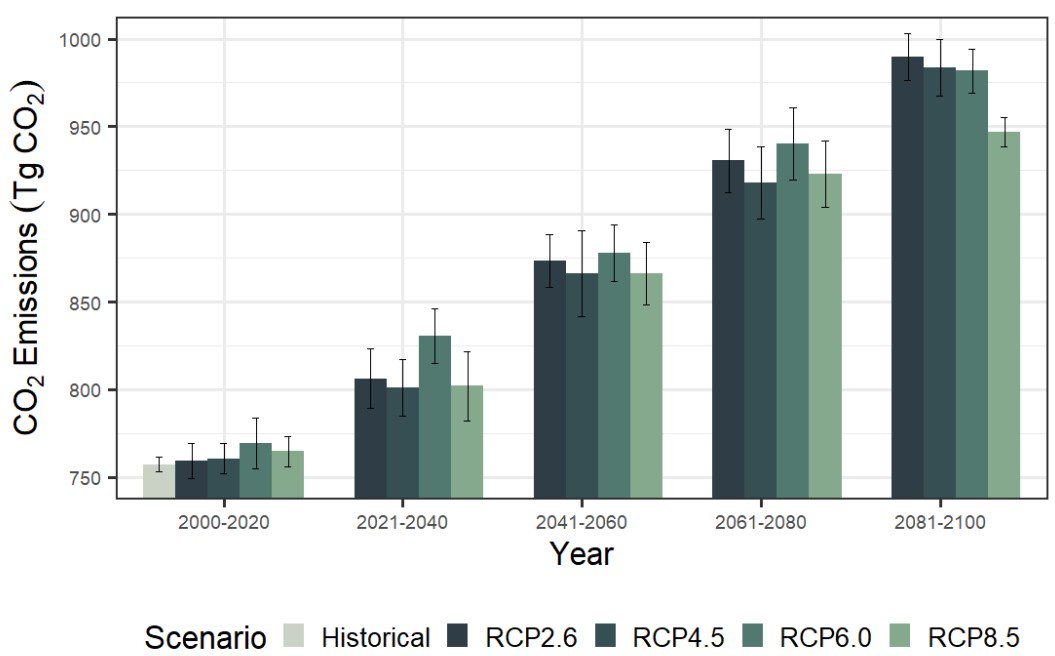

**Figure 13.** Temporal variation of projected $CO_2$ emissions from 2000 to 2100 under different RCPs scenarios.

Table 4. Twenty-year average (± 2 standard deviation) of projected emissions of $CO_2$, CO, $PM_{2.5}$, TPM, and TPC species from forest fires in Siberia (2023-2100). The emissions of the remaining 28 species are listed in Table S6 in the Supplement.

| Emissions | Year | 2000-2020 | 2021-2040 | 2041-2060 | 2061-2080 | 2081-2100 |
|---|---|---|---|---|---|---|
| Tg $CO_2$ year$^{-1}$ | Historical | 757.33 ± 4.64 | n.a | n.a | n.a | n.a |
| | RCP8.5 | 764.70 ± 9.04 | 801.94 ± 20.50 | 866.21 ± 18.29 | 922.85 ± 19.59 | 946.99 ± 8.47 |
| | RCP6.0 | 769.24 ± 14.87 | 830.52 ± 16.01 | 877.93 ± 16.77 | 940.46 ± 21.12 | 981.73 ± 12.93 |
| | RCP4.5 | 760.56 ± 8.82 | 800.86 ± 16.51 | 866.33 ± 25.06 | 918.01 ± 20.92 | 983.71 ± 16.74 |
| | RCP2.6 | 759.38 ± 10.19 | 806.32 ± 17.42 | 873.30 ± 15.29 | 930.61 ± 18.45 | 989.93 ± 13.72 |
| Tg CO year$^{-1}$ | Historical | 60.63 ± 0.37 | n.a | n.a | n.a | n.a |
| | RCP8.5 | 61.22 ± 0.72 | 64.20 ± 1.64 | 69.34 ± 1.46 | 73.88 ± 1.57 | 75.81 ± 0.68 |
| | RCP6.0 | 61.58 ± 1.19 | 66.48 ± 1.28 | 70.28 ± 1.34 | 75.28 ± 1.69 | 78.59 ± 1.04 |
| | RCP4.5 | 60.88 ± 0.71 | 64.11 ± 1.32 | 69.35 ± 2.01 | 73.49 ± 1.68 | 78.75 ± 1.34 |
| | RCP2.6 | 60.79 ± 0.82 | 64.55 ± 1.39 | 69.91 ± 1.22 | 74.50 ± 1.48 | 79.24 ± 1.10 |
| Tg $PM_{2.5}$ year$^{-1}$ | Historical | 9.88 ± 0.06 | n.a | n.a | n.a | n.a |
| | RCP8.5 | 9.97 ± 0.12 | 10.46 ± 0.27 | 11.30 ± 0.24 | 12.03 ± 0.26 | 12.35 ± 0.11 |
| | RCP6.0 | 10.03 ± 0.19 | 10.83 ± 0.21 | 11.45 ± 0.22 | 12.26 ± 0.28 | 12.80 ± 0.17 |
| | RCP4.5 | 9.92 ± 0.11 | 10.44 ± 0.22 | 11.30 ± 0.33 | 11.97 ± 0.27 | 12.83 ± 0.22 |
| | RCP2.6 | 9.90 ± 0.13 | 10.51 ± 0.23 | 11.39 ± 0.20 | 12.14 ± 0.24 | 12.91 ± 0.18 |
| Tg TPM year$^{-1}$ | Historical | 7.48 ± 0.05 | n.a | n.a | n.a | n.a |
| | RCP8.5 | 7.55 ± 0.09 | 7.92 ± 0.20 | 8.56 ± 0.18 | 9.12 ± 0.19 | 9.35 ± 0.08 |
| | RCP6.0 | 7.60 ± 0.15 | 8.20 ± 0.16 | 8.67 ± 0.17 | 9.29 ± 0.21 | 9.70 ± 0.13 |

| Emissions | Year | 2000-2020 | 2021-2040 | 2041-2060 | 2061-2080 | 2081-2100 |
|---|---|---|---|---|---|---|
| | RCP4.5 | $7.51 \pm 0.09$ | $7.91 \pm 0.16$ | $8.56 \pm 0.25$ | $9.07 \pm 0.21$ | $9.72 \pm 0.17$ |
| | RCP2.6 | $7.50 \pm 0.10$ | $7.96 \pm 0.17$ | $8.63 \pm 0.15$ | $9.19 \pm 0.18$ | $9.78 \pm 0.14$ |
| Tg TPC year$^{-1}$ | Historical | $5.18 \pm 0.03$ | n.a | n.a | n.a | n.a |
| | RCP8.5 | $5.23 \pm 0.06$ | $5.49 \pm 0.14$ | $5.93 \pm 0.13$ | $6.32 \pm 0.13$ | $6.48 \pm 0.06$ |
| | RCP6.0 | $5.26 \pm 0.10$ | $5.68 \pm 0.11$ | $6.01 \pm 0.11$ | $6.44 \pm 0.14$ | $6.72 \pm 0.09$ |
| | RCP4.5 | $5.20 \pm 0.06$ | $5.48 \pm 0.11$ | $5.93 \pm 0.17$ | $6.28 \pm 0.14$ | $6.73 \pm 0.11$ |
| | RCP2.6 | $5.20 \pm 0.07$ | $5.52 \pm 0.12$ | $5.98 \pm 0.10$ | $6.37 \pm 0.13$ | $6.77 \pm 0.09$ |

## 4 Discussion

### 4.1 Feasibility of fire simulation

According to the default module, the fires spread throughout almost all of Siberia (S4.a-d, 4.a-d) because the module considered only the fuel amount and fuel moisture content. Thus, if the fuel load met the threshold requirement in any random grid, a fire appeared and could spread to other areas. Furthermore, the spatial distribution and trend of burned biomass under all of the RCP scenarios in the default fire module were not consistent with the burned fraction data. Areas with high burned fraction values should also have high burned biomass, and vice versa.

However, in the improved module, the fires ignited only in areas that were covered in the lightning ignition and population ignition datasets based on the calculation of each ignition factor. This is confirmed by the comparison of the fire variable with the ignition factor variables, the comparison of the burned fraction variable with the lightning flash strikes variable shows a strong correlation of 0.68 ($R^2$=0.45), and the comparison of the burned fraction variable with the population density variable shows a correlation of 0.24 ($R^2$=0.06) (Figure S7). These relatively low correlation values are due to the fact that the presence of an ignition factor does not guarantee that a fire will start; the area needs to have sufficient dry litter to feed the fire. Apart from these variables in the improved model, other factors also influence fire occurrence and spread in real life, such as slope and solar aspect (Rothermell, 1972), but their inclusion at this point was not possible due to the limitations of the model. In addition, when comparing the fire and AGB distributions, the SEIB-DGVM SPITFIRE showed greater agreement than did the default fire module.

However, differences remained between the spatial distribution patterns of the simulated fires and the GFED4s data in eastern Siberia. We believe that the main reason for the lack of simulated fires in eastern Siberia was the scarcity of available fuel and biomass for the ignition and spread of fires. We found that the AGB in these areas (130-142°E and 65-80°N; Figure S23) was very low, averaging 65.59 g C m$^{-2}$. This value was far below the model minimum fuel load threshold requirement of 200 g C m$^{-2}$ (Sato et al., 2007) for fire ignition or spread. All three benchmark datasets, the ESA Biomass CCI (aboveground biomass), GFED4 (burned area), and GFED4s (burned fraction), indicate that fire is present in this area, with ESA Biomass CCI showing an AGB of 2,309.67 g DM m$^{-2}$. It is challenging to produce a model product that precisely predicts observations, as the simulations are highly dependent on the input data and dynamics, while the benchmark datasets were obtained from satellite image estimations that are able to capture natural conditions and events in real time. Even predictions based on satellite observations can differ significantly from field-based observations. For example, the International Forest Fire News (IFFN) Russian Federation reported that 2003 had extremely severe fires in Siberia based both on ground and aerial observation. However, the burnt area was determined to be 2,654,000 ha based on field observations and 17,406,900 ha based on satellite-derived observations (NOAA AVHRR) (IFFN, 2003; Siegert and Huang, 2005). The difference between ground observation data and satellite-derived data is due to differences in the data collection time and continuity. Ground-based observations are carried out only for a short time due to technical difficulties, while

observations based on satellite data are carried out without any significant difficulties (IFFN, 2003). In this case, the SEIB-DGVM SPITFIRE model reported a burned area of 7,969,785 ha, an estimation centered between the observational and satellite data.

Overall, based on the fire variable outputs (burned fraction and burned area) from the improved model generated and validated with benchmark data, we project that Siberia will have an increasing trend until 2100 (Figure S4.d, Figure S6.b, and Figure 8.d). Yasunari et al., (2024) in a comprehensive assessment of the impacts of the Siberian wildfire using MIROC5 stated that there is high probability of increased Siberian wildfires in the future, and this estimate implies that worse air quality due to wildfires is predicted in the future, with frequent exceedances of air quality environmental standards (ES).

Kasischke and Bruhwiler (2003) stated that the level of uncertainty in the burned area parameter for estimating fire emissions in the Russian boreal forest is ± 30% for satellite imagery, while the uncertainty of the parameter is -300% according to official government statistics, resulting in fires being largely underestimated. This difference in uncertainty was caused by the diverse parameters/equations used for estimation, the varying levels of detail of the analysis, and other factors, such as forest type, location, fuel load, fire type, and aboveground biomass density. Differences are also extrapolated when estimations for large areas are based on individual fires (Kasischke and Bruhwiler, 2003; Kukavskaya et al., 2013). Therefore, uncertainties will inevitably persist in model- or simulation-based research when comparing model- or simulation-based data with direct observations.

## 4.2 Forest resilience under fire and climate change

Terrestrial NPP is an essential element of the carbon cycle and global climate dynamics, as it directly affects the $CO_2$ content of the atmosphere, resulting in delayed climatic changes (Running, 2022). If NPP decreases, the land's ability to absorb $CO_2$ will decrease, causing atmospheric $CO_2$ to increase faster and thereby contributing climate change (Running, 2022). Based on the comparison between fire-on and fire-off simulation, under RCP8.5 scenario, from 2000 to 2100 the NPP will decrease by $385.11 \pm 40.4$ g C m$^{-2}$ ($5.03 \pm 1.5$ g C m$^{-2}$ year$^{-1}$) due to wildfires until 2100. Satellite observations one year after boreal forest fires in Alaska and Canada recorded a 60–260 g C m$^{-2}$ loss of NPP (Hicke et al., 2003). In the coniferous forests of the western United States, postfire NPP loss was also recorded and ranged from 67 to 312 g C m$^{-2}$ year$^{-1}$ (Sparks et al., 2018). These data indicate that the NPP simulation results of the SEIB-DGVM SPITFIRE model are also consistent with some observational data in different areas.

NPP and NBP, both are significant elements of the global C cycle and are used as indicators of ecosystem function and are linked to biodiversity, biogeochemical cycling, ecosystem resilience, and other aspects of ecosystem services (Canadell et al., 2007; Richmond et al., 2007; Ito, 2011). However specifically, the mitigation ability of ecosystems is determined by net biome productivity (NBP) (Chapin et al., 2006; Fisher et al., 2014), and climate-driven large anomalies in NBP could impact the structure, composition, and function of terrestrial ecosystems (Frank et al., 2015). The twenty-year average NBP from 2000-2100 shows a carbon sink in Siberia with a increasing trend (Figure S34). Overall, from 2000 to 2100, RCP8.5

produces the highest value, then RCP6.0, RCP4.5 and RCP2.6 with values of 304.61 ± 11.77, 286.78 ± 10.64, 286.17 ± 10.99, and 274.95 ± 9.36 Tg C year$^{-1}$, respectively (Table S4). The historical annual mean value of NBP in Siberia for 2000-2021 of 136.39 ± 83.4 Tg C year$^{-1}$ is also similar to the CLM4CN simulations (annual average 1981-2006) in Eurasia and Boreal and Arctic of 204 and 284 Tg C year$^{-1}$, respectively (Kantzas et al., 2013).

Under all climate scenarios from 2000 to 2100, we estimate that the net biome productivity (NBP) will continue to increase, indicating a continued flux of $CO_2$ from the atmosphere to the land (Figure S34). The classification of NBP variables based on climate input data also shows the correct order, from the smallest under RCP2.6 to the largest under RCP8.5 (Figure S35). This is because climate factors, such as temperature and precipitation, have a positive impact on vegetation (Yuan et al., 2021). On average, from 2000 to 2100, under the RCP8.5 climate scenario, the NBP in Siberia is estimated at 301.3 ± 49.1

670 Tg C (equivalent to 3.01 ± 0.5 Tg C year$^{-1}$). Other studies have similar estimation that the NBP across northern peatlands, including the Russian Far East (RFE) and West Siberian Lowlands (WSL), ranges from 10 to 220 Tg C year$^{-1}$ (Qiu et al., 2022). Additionally, we estimate that the heterotrophic respiration (HTR) in Siberia will continue to increase until 2100. On average under RCP8.5, from 2000 to 2100, the net biome productivity (NBP) value in Siberia is estimated at 4,002.7 ± 967.7 Tg C (equivalent to an increase of 40 ± 9.7 Tg C year$^{-1}$) (Figure S36). We suggest that the high HTR values during those

675 years were attributable to an elevated fuel load (Figure S40.a) followed by significant precipitation (Figure S40.b), which increased litter moisture content (Figure S41.a) and consequently accelerated the decomposition rates of litter and soil organic carbon. Increased of heterotrophic respiration, tree mortality and increased disturbance (drought and fire) contribute significantly to negative carbon fluxes from the ecosystem due to increased temperature and atmospheric $CO_2$ (Sharma et al., 2023). Overall, SEIB-DGVM SPITFIRE simulates that until the end of the 21st century, there will continue to be a

680 strengthening of the land carbon sink in Siberia under all RCP scenarios. Boreal forests (1135 Mha) consistently acted as an average carbon sink of 0.5 ± 0.1 Pg C year$^{-1}$ over the two decades from 1990 to 2010. Furthermore, Asian Russia had the largest boreal carbon sink, which showed no overall change despite increased emissions from wildfire disturbances (Pan et al., 2007).

Boreal forest vegetation is naturally influenced by a variety of periodic disturbances, such as wildfires (Kasischke et al.,

1995), insect outbreaks, and windthrow. Wildfires and insect outbreaks are not necessarily independent, there is a likelihood of wildfires often increasing or decreasing after insect outbreaks (Meigs et al., 2015, 2016). However, wildfires are among the main disturbances that drive forest dynamics, shape forest composition and structure, and affect biomass and productivity (Burns and Honkala, 1990; Greene and Johnson, 1999). Circumpolar northern boreal forests and tundra are likely to continue to warm more than most other terrestrial biomes according to available data from models and observations (Chapin et al.,

2005; Foley, 2005; Meehl et al., 2007; Trenberth et al., 2007; Lee et al., 2021). Based on the observations and changes in regional attributes from 1950 to the present, it is projected that during 2071-2100, the WSB (West Siberia), ESB (East Siberia), and RFE (Russian Far East) will experience an increase in extreme temperatures with high confidence of more than 7 °C for all seasons under the RCP8.5 scenario. Projected warming is most evident on the large continental Siberian Plateau, which has boreal and subboreal climates and biomes (i.e., taiga forests and tundra), during the winter season (Ozturk et al.,

2017; IPCC, 2021). Such changes in climatic extreme scenarios and seasonality are also likely to have multiple effects, including extended but drier growing seasons, the occurrence of more intense convective storms leading to more lightning-caused fires (Hessilt et al., 2021; Kharuk et al., 2022), and decreased forest productivity (Orangeville et al., 2018); additionally, longer, warmer, and drier summers may cause an increase in fire frequency and size in some areas of boreal forests (Krawchuk et al., 2009; Flannigan et al., 2016; Wotton et al., 2017). This finding is in line with our results, which

show that the assessment of forest ecology variables indicates tree mortality due to fire and succession as well as postfire vegetation (Figure S25.c) and affects NPP dynamics in Siberia (Figure S25.a).

Under the RCP2.6 scenario, the SEIB-DGVM estimated the average tree density to be 2,166 tree ha⁻¹ between 200 and 2023 in Siberia. The tree density is greater in northeastern Siberia (1,197 tree ha⁻¹) than in southern Siberia (Miesner et al., 2022). Our simulation resulted in higher tree densities than did the observations in northeastern Siberia, as we covered a larger area

of forest at 60°-180°E and 45°-80°N. The number of trees is affected by the frequency of fires at a certain location. Additionally, the number of trees destroyed by wildfires depended upon the climate scenario used in the simulations but naturally increased with fire frequency and size. In all the RCP scenarios, the number of destroyed trees was greater than that in the historical simulation, and the number of destroyed trees increased annually, indicating that changes in climatic factors affected the surviving tree density. The projected increase in the number of trees destroyed annually is consistent with the

modeled fire product data, which exhibit an increasing trend until 2100. The difference in tree mortality data between climate scenarios is because each climate scenario has a different projected temperature increase. In Siberia, under the RCP8.5 scenario, we simulate that the 2-meter surface temperature will increase by 4.67°C by 2100 (Figure S42). This estimate aligns with the IPCC projections, which predict air temperature increases by 2100 ranging from 0.3-1.7°C (average 1.0°C) under the RCP2.6 scenario, 1.1-2.6°C (average 1.8°C) under the RCP4.5 scenario, 1.4-3.1°C (average 2.2°C) under

the RCP6.0 scenario, and 2.6-4.8°C (average 3.7°C) under the RCP8.5 scenario (IPCC 2021).

The DBH ranges of the trees in the fire-on and fire-off simulations were comparable to those in northeastern Siberia, where the DBH ranged up to 71,6 cm, the tree height up to 28,5 m, and the crown area averaged 4,77 m² (Miesner et al., 2022). As the average DBH variable was similar in the fire-on and fire-off simulations, trees with large DBHs are resistant to fire. This was also confirmed based on observational research in Yenisei Siberia, where trees with a DBH greater than 18.1 cm were

720 the most resistant to further postfire succession (Bryukhanov et al., 2018). Specifically, based on the division of regions, we found an interesting pattern, that in Siberia the eastern region has the highest value of allometry variables (tree height, tree DBH and crown area), then the central region and lowest is the west region (Figure S37, Figure S38, and Figure S39). An interesting pattern was observed in western Siberia, where trees with high height and large DBH but low crown area were detected in some locations (Figure 11.a). We suggest that this happens because of the wildfire, the Siberian central region

has the highest wildfire frequency followed by the west region, then the east region. The major Siberian Forest types are formed by larch (*Larix sibirica, L. gmelinii, and L. cajanderi*) and majority distributed in western and central Siberia (Figure 1, and Figure 1: (Kharuk et al., 2021)). Furthermore, larch is classified as pyrophytic species, that have adapted or evolved under conditions of periodic forest fires, they have adapted and gaining a competitive edge over non-fire adapted species in

regenerating and growing in burned areas (Kharuk et al., 2021). The abundance of species and high frequency of wildfires in the Siberian central and western regions led to excellent larch succession and regeneration as evidenced by the high tree allometry variables and on the other hand the projected continuity of wildfires led to a downward trend. On the other hand, in the eastern region, very few wildfires are simulated, partly due to the low aboveground biomass available in some areas, which affects ignition and fire spread (Figure S23). However, due to the low frequency of wildfires, allometric variables are projected to have an increasing trend until 2100 in the Siberian east region. The unique relationship between allometric variables, which are naturally distributed without a wide gap between grid plots (Figure 11.c), in the eastern region is also due to the area's low wildfire frequency. The forest in the Siberian east region appears to grow and spread naturally without having high impact from the wildfire. While the majority tree species in Siberia: larch, regenerates extremely well on post-fire-mineralized soil, however on contrast, they regenerate very slowly over a ground floor covered in lichen and moss (where the soil's surface is tough for sapling roots to reach) (Kharuk et al., 2016).

## 4.3 Spatial distribution and temporal variation in biomass burning emissions under climate change scenarios

The spatiotemporal dynamics of the biomass burning emissions under all RCP scenarios had similar patterns and trends, but they had slightly different variations in dynamics because climate affects the frequency and distribution of fires. This is evidenced by all fire variables produced by the model, from burned fraction to burned biomass emissions. In the last 20 years of the projection (2080-2100), the highest values were obtained from simulations using climate inputs RCP2.6, RCP4.5, RCP6.0, and RCP8.5. This occurs because each RCP scenario exhibits varying radiative forcing, with RCP8.5 notably experiencing the highest temperature increase (Figure S42) and also projecting the highest precipitation levels (Figure S40.b). The fuel load variable follows a corresponding order reflective of RCP forcing levels, with RCP8.5 showing the highest and RCP2.6 the lowest (Figure S40.a). However, due to increased precipitation and temperature-induced snowmelt, the moisture content of litter fractions in RCP8.5 simulations attains the highest values, contrasting with the lowest values in RCP2.6. Consequently, available fuel loads may not ignite in areas with high moisture content, leading to projections of the highest burned biomass emissions in the last 20 years of RCP climate projections (2080-2100) for RCP2.6, RCP4.5, RCP6.0, and RCP8.5, respectively. The difference in emission values between climate scenarios in the same year shows that temperature has an impact on vegetation succession and climate-sensitive emission production from wildfires (Gutierrez et al., 2021; Stocker et al., 2021). Thus, the model is able to simulate and integrate fire disturbance, forest dynamics or vegetation succession, and burned biomass emissions well.

Over a 20-year average from 2080 to 2100, under RCP6.0 in Siberia, our simulation predicts that forest fires will emit $CO_2$, CO, PM2.5, TPM, and TPC in amounts of 989.93 ± 13.72, 79.24 ± 1.10, 12.91 ± 0.18, 9.78 ± 0.14, and 6.77 ± 0.09 Tg, respectively (Table 4). Spatially, the projections depict heterogeneous patterns of burned biomass emissions, with regions of high emissions intensity concentrated in areas of larch forest (*Larix spp.*), consistent with Figure 1 and our simulation results, where the fire and emission variables show high values in central to southern Siberia (Figure S4.b, Figure 8.b, and Figure S6.b). This is reinforced by field-based estimation data, that fires in this region result in high tree mortality 76%,

Siberian larch forests experience greater aboveground carbon loss after fire than do North American forests, both in absolute and relative levels (Webb et al., 2024). We also visualized all the 33 graphs depicting projected burned biomass emissions, offering valuable insights into the future dynamics of the burned biomass emissions in Siberia. Across these graphs, we observe distinct temporal patterns, revealing trends in burned biomass emissions over time. Under the RCP8.5 to RCP2.6 scenarios, the twenty-year average comparison of overall burned biomass emissions data from 2080-2100, compared to data from 2000-2020, shows projected increases of 23.87%, 27.63%, 29.34%, and 30.36%, respectively (Figure S43). The twenty-year dynamics are summarized in Table 4 and Table S6. Furthermore, each year, various climate scenarios predicted differing emissions based on the respective radiative forcing values (from lowest to highest). The RCP8.5, RCP6.0, RCP4.0, and RCP2.6 scenarios exhibited average annual increases of 0.295%, 0.354%, 0.358%, and 0.361% year$^{-1}$, respectively, from 2000 to 2100.

Under the RCP4.5 scenario, radiative forcing stabilized until 2100 (Thomson et al., 2011), which is consistent with our results, as emissions under the RCP4.5 scenario were more stable than those under the other RCP scenarios. Therefore, its indicated that the trend in fire emissions is consistent with the different scenario-dependent trends in radiative forcings.

Overall, based on the RCPs climate scenario data used (MirocAR5), the emission scenario projected an increase in global mean surface temperature in the range of 1.0-3.7 (0.3~4.8) °C (IPCC, 2014), and currently ranges between 1.5 and 6.0 °C by 2100 compared to 1850-1900 mean value (Lee et al., 2021). One of the impacts of rising global temperatures is the increased occurrence and severity of forest fires, which lead to a greater prevalence of wildfire (Schoennagel et al., 2017; Haider et al., 2019). The global land area burned by wildfires is expected to increase by 35% if the global temperature increases by 2 °C and precipitation patterns change (Pörtner et al., 2022). Extremely high temperature increase the frequency of severe droughts and proliferate wildfires in several regions, such as southern Europe, northern Eurasia, the USA, and Australia (IPCC, 2021). These frequent and severe wildfires will inevitably lead to an increase in the atmospheric concentration of biomass burning products (Marlon et al., 2008; Amiro et al., 2009; Tian et al., 2023).

Forests in Siberia are very important to monitor and assess continuously because they have a significant impact on regional (short-term) and global (long-term) air quality and human health due to the large amounts of carbon emissions, smoke aerosols, and trace gases in the atmosphere. In addition to the observed amount of emissions, OC/EC emissions exceeded 3 times, and emissions of inorganic ions ($SO_4^{2-}$ and $NH_4^+$) were found to be 5 times greater than the annual average wildfire emissions from August 2010 to August 2011 (Popovicheva et al., 2014). Increased Siberian wildfire aerosols would significantly degrade air quality, particularly in the surrounding and downwind regions of Siberia (Yasunari et al., 2024).The emitted substances can be transported over long distances and affect air quality in other regions, including North America and Northeast China (Teakles et al., 2017; Johnson et al., 2021; Sun et al., 2023).

Estimating future fire emissions and their impact on air quality is challenging due to model limitations and uncertainties in estimation methods, potential mixing of emissions in the atmosphere, climate radiative forcing factors, and emission transport (Winiger et al., 2017; Schacht et al., 2019). The SEIB-DGVM SPITFIRE was not able to reproduce the events in the validation data for the same year or month but simulated similar dynamic patterns and values. This difference occurs

because the benchmark data obtained from satellite image data closely follow natural conditions, while the model accumulates uncertainties due to its long simulation period. The emission estimation method used in the model refers to the dry matter variable and the emission factor from Andreae and Merlet (2001) and Andreae (2019), where the emission factors are obtained from laboratory and small field experiments. Each species has specific characteristics that require different assessment methods, and the combustion characteristics can be very different from those of large-scale open biomass burning and wildfires. Kasischke and Bruhwiler (2003) reported that the level of uncertainty in the emission factor parameters for estimating emissions from fires in Russian boreal forests was ±20-50%, which agrees with the ±50% uncertainty level for major emissions presented by Andreae and Merlet (2001). However, in the SEIB-DGVM SPITFIRE, we also used the latest emission factor from Andreae (2019), which was developed for oxygenated volatile organic compounds and for HCN; this approach improved all assessment compound emissions significantly with more accurate measurements and has been widely used by various dynamic global vegetation models to estimate biomass burning emissions globally. Overall, the comparison between different climate RCP scenarios provides further insight into uncertainties and variability in the projections, offering valuable information for understanding the potential impacts of future burned biomass emissions on air quality, climate dynamics, and ecosystem health. Through this analysis, our study contributes to a better understanding of the drivers and implications of burned biomass emissions, informing policy decisions and management strategies aimed at mitigating their environmental and societal impacts.

## 4.4 Model uncertainty

Our study is a process of combining and improving the SEIB-DGVM model with the SPITFIRE fire module, each of which has different characteristics and some default variables, parameters, and inputs. The implementation of any complex model improvement inherently introduces uncertainty, stemming from various sources such as parameterization choices, model structure, new model input and the representation of complex biophysical processes. Specifically, in the context of the fire module enhancement, uncertainties may arise from the characterization of ignition sources, fire behavior, fuel dynamics, and fire spread mechanisms. These uncertainties can significantly influence the accuracy and reliability of model projections, particularly in simulating the spatial and temporal patterns of fire occurrence, intensity, and impacts on vegetation dynamics and carbon cycling.

Kasischke and Bruhwiler (2003), mentioned that in some cases, the data needed to generate input parameters for those equations are very well defined, whereas in others, they are based on a very limited set of the observations data. Thus, the input data selection and the parameter setting for those equations calculation is the source of the uncertainty, provided emission range (Table 5 in Kasischke and Bruhwiler, 2003). In more detail Kasischke and Bruhwiler (2003) stated that uncertainty can be classified into two groups. First, environmental characteristics (direct or indirect observations) including location, fire type, and aboveground/ground-layer biomass. Second, uncertainties from parameters that can be measured in

individual biomass combustion processes, while the application on a large scale, time differences and climatic influences are very challenging, and the combustion process consists of several stages that emit different emissions for each stage.

To mitigate model uncertainty, we have employed rigorous model verification, calibration, and evaluation procedures, comparing model outputs with several benchmark datasets (Table 3). The verification helps to ensure the new inputs (ignition factors) can be read, processed, and output properly. The process of calibrating all major variables with benchmark datasets is carried out sequentially and with several iterations, ensuring that the output of individual variables matches the benchmark dataset that is the target of validation. This validation process helps assess the model's ability to reproduce historical fire patterns and dynamics accurately.

However, this does not close the uncertainty in the model we have developed, we still have limitations where the emission variable distribution pattern of emissions is strongly influenced by the pattern of the resulting fire, because the emission variable is calculated with the same dry matter emissions. This spatial distribution also affected other vegetation variables, due to the relationship calculation between fire and vegetation variables. This is a potential further study to adjust the factors that affect the distribution of fire to be similar to the benchmark data. In addition, natural factors that affect the dynamics of significant fire disturbances at specific times are still not well simulated by our model. Inversely, our model also has the advantage of being able to simulate numerically averaged data in the long term very accurately, based on the results of numerical comparisons with benchmark data, the model is able to simulate with a value of 99%.

## 5 Conclusions and recommendations

We introduced the SPITFIRE fire module into the SEIB-DGVM and achieved a better representation of fire dynamics in Siberia between 1996 and 2100 by creating monthly outputs and producing several new outputs related to fires at a 0.5° spatial resolution, such as vegetation and burned biomass emission variables. Our modifications have led to a more realistic depiction of fire frequency, intensity, and extent, aligning the model outputs more closely with benchmark datasets. The major variables related to fire (vegetation, $CO_2$ and $PM_{2.5}$ emissions, burned area, burned fraction, aboveground biomass, and dry matter) all reached an agreement of 70.7 % or greater with the observations. Additionally, the improved model accurately simulated forest structure, increasing the agreement between the simulated and observed dataset patterns and further emphasizing the reliability of our model and its emission projections. Under the RCP2.6 scenario, we estimated that the $CO_2$, CO, $PM_{2.5}$, total particulate matter (TPM), and total particulate carbon (TPC) emissions in Siberia will continue to increase annually until 2100 by an average of $2.71 \pm 0.87$, $0.22 \pm 0.07$, $0.04 \pm 0.01$, $0.03 \pm 0.01$, and $0.02 \pm 0.01$ Tg species year$^{-1}$, respectively. Moreover, forest fires in Siberia in 2100 are projected to emit all five of these compounds under the RCP8.5 scenario, amounting to $1010.00 \pm 82.64$, $80.84 \pm 6.61$, $13.17 \pm 1.08$, $9.97 \pm 0.82$, and $6.91 \pm 0.57$ Tg, respectively.

Although our research has made significant steps, there are several limitations that require further research. Future studies should minimize the uncertainty of the simulations and achieve better fits with benchmark datasets on fire, vegetation, and emission products. Specific parameter settings need to also be developed to emphasize regional and seasonal differences.

Continued improvement in the fire module and consideration of feedback loops will be crucial to continuously enhancing the accuracy of our models. Our work contributes to a more comprehensive understanding of the intricate interactions between fire dynamics, ecosystems, and climate, creating a new path for informed decision-making and broadening the field of biogeochemistry, global elemental cycles, and the importance of accurate vegetation dynamic modeling.

## Appendix A

### A.1 Input and outputs of the SEIB-DGVM SPITFIRE

A.1.1. Input

1) Location: Latitude and altitude.
2) Soil (fixed in time): Soil moisture at the saturation point, field capacity, matrix potential, wilting point, and albedo.
3) Climatic data (daily): Air temperature, soil temperature, fraction of cloud cover, precipitation, humidity, and wind velocity.
4) Atmospheric carbon dioxide ($CO_2$) concentrations
5) Fire ignition factors: population density (GPWv4) and lightning flash rate (LIS/OTD HRFC)

A.1.2. Output

1) Carbon dynamics (daily–yearly): Terrestrial carbon pool (woody biomass, grass biomass, litter, and soil organic matter), $CO_2$ absorption and emission rates.
2) Water dynamics (daily): Soil moisture content (in three layers), interception rate, evaporation rate, transpiration rate, interception rate, and runoff rate.
3) Radiation (daily): Albedo from the terrestrial surface.
4) Properties of vegetation (daily–yearly): Vegetation type, dominant plant functional type, leaf area index, tree density, size distribution of trees, age distribution of trees, woody biomass for each tree, and grass biomass per unit area.
5) Disturbances (monthly–yearly): fire fraction, burned area, burned biomass, FDI, complete SPITFIRE variables, and 33 types of burned biomass emissions.

### A.2 Processes in the SEIB-DGVM SPITFIRE and the approaches used to represent each process

| Process | Approach | References |
|---|---|---|
| **Disturbance** | Fire as an empirical function of fuel (litter and aboveground biomass), fuel moisture, and ignition factor (human- and lightning-caused) | (Thonicke et al., 2001, 2010) |
| **Biogeochemical** | Trace gas emissions as an empirical function of the total amount of biomass burning and emission factor of each trace gas species | (Andreae and Merlet, 2001) |

## A.3 PFTs, variables, parameters, and constants in the model's equations

| Abbreviation | Description | Unit |
|---|---|---|
| TrBE | Tropical broad-leaved evergreen | - |
| TrBR | Tropical broad-leaved raingreen | - |
| TeNE | Temperate needle-leaved evergreen | - |
| TeBE | Temperate broad-leaved evergreen | - |
| TeBS | Temperate broad-leaved summergreen | - |
| BoNE | Boreal needle-leaved evergreen | - |
| BoNS | Boreal needle-leaved summergreen | - |
| BoBS | Boreal broad-leaved summergreen | - |
| TeH | Temperate herbaceous ($C_3$ grass) | - |
| TrH | Tropical herbaceous ($C_4$ grass) | - |
| M3 | Probability of each PFTs survival after fire (varying 0.0–1.0) | - |
| $pool\ w$ | The soil water content of each soil layer | mm/day |
| $Depth$ | Depth of each soil layer | meter |
| $Wfi$ | Field capacity | $m^3\ m^{-3}$ |
| Ab | Area burnt | ha/time unit |
| A | Grid cell area | ha |
| $\rho_b$ | Fuel bulk density | $kg\ m^{-3}$ |
| FDI | Fire Danger Index (0.0–1.0) | - |
| $L_B$ | Length to breadth ratio for woody and grass PFTs | - |
| $U_{forward}$ | Forward wind speed | m/s |
| $E(N_{ig})$ | Expected number of fire ignition event (sum of population and lightning ignitions) | $km^2$/time unit |
| $E(N_{ih})$ | Expected number of human-caused fire ignition | $km^2$/time unit |
| $E(N_{il})$ | Expected number of lightning-caused fire ignition | $km^2$/time unit |
| Ip | Ignition parameter: Define the power of lightning caused ignition (0.0–1.0) | - |

| Abbreviation | Description | Unit |
|---|---|---|
| $\omega_o$ | Relative moisture content | - |
| NI | Nesterov Index | $^oC^2$ |
| NBP | Net Biome Production | g C year$^{-1}$ |
| Tmax | Maximum temperature | $^oC$ |
| Tmin | Minimum temperature | $^oC$ |
| Tdew | Dew-point temperature | $^oC$ |
| $m_e$ | Moisture extinction | - |
| $\alpha_{av}$ | Drying parameters for 1-, 10- and 100-h fuel classes | $^oC^{-2}$ |
| $ROS_{f, surface}$ | Forward rate of spread of surface fire | m min$^{-1}$ |
| $ROS_{b, surface}$ | Backward rate of spread of a surface fire | m min$^{-1}$ |
| $I_R$ | Reaction intensity | kJ m$^{-2}$ min$^{-1}$ |
| $\xi$ | Propagating flux ratio | - |
| $\phi_w$ | Wind factor | - |
| $P_b$ | Probability of fire per unit time | Time unit$^{-1}$ |
| $\varepsilon$ | Effective heating number | - |
| $Q_{ig}$ | Heat of preignition | kJ kg$^{-1}$ |
| $t_{fire}$ | Fire duration | min |
| $I_{surface}$ | Surface fire intensity | kW m$^{-1}$ |
| SH | Scorch Height | m |
| F | PFT-parameter in crown scorch equation | - |
| CK | Fraction of crown scorch | - |
| $T_H$ | Tree height | m |
| CL | Crown length of woody PFT | m |
| $P_m$ | Probability of postfire mortality | - |
| $P_m(CK)$ | Probability of mortality as a result of crown scorching | - |
| $P_m(\tau)$ | Probability of mortality by cambial damage | - |
| p | Parameter for woody PFTs used in $P_m(CK)$ equation | - |
| $\tau_l$ | Residence time of the fire | min |
| $\tau_c$ | Critical time for cambial damage | min |
| BT | Bark thickness | cm |

| Abbreviation | Description | Unit |
|---|---|---|
| par1, par2 | Parameters for woody PFTs used in bark thickness calculation | - |
| DBH | Diameter at breast height | m |
| $E_{i,j}$ | Fire emissions of trace gas and aerosol species i and the PFT j | g species $m^{-2}$ $s^{-1}$ |
| $EF_{i,j}$ | PFT-specific emission factor | g species (kg dry matter (DM))$^{-1}$ |
| $CE_j$ | Combusted biomass of *PFT $j$* due to the fire | g C $m^{-2}$ |
| C | Unit conversion factor from carbon to dry matter | g C (kg DM)$^{-1}$ |
| $D_T$ | Distance traveled | m |
| $U_{forward}$ | Forward wind speed | m $min^{-1}$ |
|  |  |  |

Additional equations and variables of the implemented SPITFIRE module are referred to with adjustments to Thonicke et al. (2010) Table A1 and Appendix A-B, respectively.

**Code and data availability**

The spatially explicit individual-based dynamic global vegetation model (SEIB-DGVM) SPITFIRE code and data generated from this study (fire, vegetation, and 33 emission variables in Siberia) are available at
895 https://doi.org/10.5281/zenodo.8299732

**Supplement**

The supplement related to this article is available online at: https://egusphere.copernicus.org/preprints/2024/egusphere-2024-105/egusphere-2024-105-supplement.pdf

**Author contributions**

TK conceptualized of the project and experimental design with help from ND and HS. HN, LV, and TM contributed model cording and writing. developed the model code, and RN performed the model simulations and analyzed model output with validation data from TS and RH. RN prepared the paper with contributions from all co-authors.

**Competing interest**

**Disclaimer**

**Acknowledgments**

The authors extend their gratitude to Tomohiro Hajima and Junko Mori from the Japan Agency for Marine-Earth Science and Technology (JAMSTEC) for providing the climate datasets for this study.

**Financial support**

This research was supported by the Japan Society for the Promotion of Science (JSPS) KAKENHI Grant-in-Aid for Scientific Research (B) 20H04317.

**Review statement**

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
