# Peer review of "Future projections of Siberian wildfire and aerosol emissions"

_EGUsphere, 2024_

## Referee Comment (RC2)

MS Title: Future prediction of Siberian wildfire and aerosol emissions via the improved fire module of the spatially explicit individual-based dynamic global vegetation model

Authors: Reza Kusuma Nurrohman, Tomomichi Kato, Hideki Ninomiya, Lea Vegh, Nicolas Delbart, Tatsuya Miyauchi, Hisashi Sato, Tomohiro Shiraishi and Ryuichi Hirata

**General comment**

The manuscript titled "Future prediction of Siberian wildfire and aerosol emissions via the improved fire module of the spatially explicit individual-based dynamic global vegetation model" by Reza Kusuma Nurrohman and colleagues evaluates the implementation of the SPITFIRE fire module into the SEIB dynamic global vegetation model for Siberia. The model is forced with reanalysis data provided by CRU and with global climate model data produced by MirocAR5 for the historical period as well as four emission scenarios until 2100. The manuscript compares simulated and observed fire-related variables and evaluates the impact of wildfires on emissions and tree mortality. The text would benefit from restructuring and enhanced focus on the aspects that really matter. Large parts of the results section are very descriptive, and it is not always self-evident why these results matter and what they imply. It is worthwhile for the authors to revisit the entire text and carefully assess what information is really worth communicating. Additionally, restructuring the results section such that it first evaluates the model during the historical period and then describes future projections afterwards is recommended. Furthermore, adding an overview figure or table that systematically compares and summarizes model performance for the old and new versions of the model is suggested. This is important to convince the reader that the new version is indeed superior. Finally, many of the literature references are outdated and should be replaced with more recent ones. I recommend that the manuscript may be considered for publication in Biogeosciences after major revision. Please find my detailed comments below.

**Detailed Comments**

L1 Note that when you are assessing the impact of emission scenarios, you should refer to *projections* rather than to *predictions*. Also, but this is completely up to the authors, my personal recommendation is to choose a more catchy title. For instance, you could simply shorten the tile to: "Future projections of Siberian wildfire and aerosol emissions"

L17 In statistics, we distinguish between accuracy and precision. You don't want precise values, rather you want accurate values.

L21 Before writing about the future changes, please add a sentence that describes how well the model reproduces historic values.

L23 The absolute numbers are not very intuitive. Please provide the relative changes in emissions and trends, instead.

L38 Please find more recent references here. A good place to start would be the 6th Assessment Report, Working Group I, Chapter 5.

L43 Currently this sentence reads as if the fact that small fires matter is consistent with a positive trend in fires. This does not make sense to me. Please replace "This finding" with something more explicit.

L61 Please use more recent references if you want to provide numbers that represent "current estimates", e.g. Friedlingstein, Pierre, et al. "Global carbon budget 2023." Earth System Science Data 15.12 (2023): 5301-5369.

L122 Please find more recent reference. The study is 20 years old and much has happened in the last two decades.

L124 Please don't cite the entire report. Specify the working group and chapter. Also, replace this reference with the most recent assessment report.

L146 The notation suggests that $E$ is a function of $n_{ig}$. What is $n_{ig}$?

L148 Please consider using a bar rather than an underscore to indicate the mean area.

L149 A grid is composed of gridcells. Please replace "grid" with "gridcell".

L264 Please include an analysis that shows how the inclusion of SPITFIRE affects the model's ability to reproduce the observed patterns during the historical period. So far you only write that the pattern of the new model version is different.

L275 Do lowest and highest values refer to spatially averaged annual mean values? Please explain.

L284 Does "uniform spatial distribution" mean that the amount of burned biomass is equal across gridcells? Please clarify.

L290 Do you have any thoughts on what causes the box-like pattern? Also, why does the default model version produce values across the entire domain, while the SPITFIRE model version has many gridcells with no values?

L292 You write that "from 2006 to 2100, the value decreased from 5.09 to 5.05 kg DM m$^{-2}$". Do the values refer to 2006 and 2100? Since there may be considerable variability, it may be better to compare 20 year averages instead, or calculate the trend.

L301 What are "estimation parameters"?

L305 What do you mean by common roots? Please rewrite this sentence.

L307 I am not sure what you are trying to communicate here. You find that a high fire frequency coincides with a low AGB. Do you think this makes sense because fires burn trees and therefore biomass is low? Or is it counter-intuitive because the more biomass, the more fuel load, the higher the probability of fires? How do you interpret your results?

L314 See comment as for L292.

L321 What does "full" refer to in this context?

L322 Why is NPP "the most measurable element of the carbon cycle"? NPP is actually difficult to measure at a canopy scale. The eddy covariance method for instance, measures NEE, from which GPP and ecosystem respiration are then derived. You could also make the case that carbon pools are easier to measure than carbon fluxes.

L325 Please add relative measures here as well, as absolute values alone are not very intuitive.

L337 Why does this matter and what do you conclude from this?

L349 Why does this matter and what do you conclude from this?

L360 Does the "annual average spatial distribution" expressed in percentage refer to the correlation coefficient $R$ of annual mean values of individual gridcells? If that is the case, please express your results in terms of fraction ($R^2 = 0.79$). Please do that consistently throughout the text.

L369 Please quantify how similar the modeled and the observed values are using a statistical metric, such as a correlation coefficient, bias, or RMSE.

L371 Since both values are identical the word "while" does not fit here. Same applies to the next sentence. Please rephrase.

L402 Is this total *annual* rainfall? Is this really rainfall or precipitation, which is the sum of rainfall and snowfall?

L403 This sentence seems to contradict the previous sentence, with two different rainfall values for the same region and time period. Please rewrite.

L417 The sentence does not make any sense to me. Why would CO2 emissions be representative for other GHG emissions, simply because most of the mass emitted is $CO_2$? The next sentence on the emission factor gives a better rational. I suggest you delete the first sentence and rewrite this part. Also, I suggest that you remind the reader what the other emissions are.

L425 What is the difference between the CO2 emission values given here and the ones quoted in the previous sentence? Both are described as annual average values.

L434 Equation 29 is located in the supplementary material. If you want to use this equation in your argument, I suggest you describe it in the text. Otherwise it is difficult to follow your argument, unless one reads the corresponding text in the supplementary material.

L454 The low-emission area only contains a few gridcells, so I don't think these values are very meaningful. To my understanding, Figure 11 shows CO2 emissions associated with fires. Why then do some grid cells have negative values? Also, the chosen color legend makes it difficult to determine whether the emissions are low but positive or negative. Finally, the legend says that the figure shows annual average differences 1996-2100. Differences with respect to what?

L501 You arite that the "lighting flash rate affected fire ignition in Siberia by 46%". This sounds as if 46% of the fire ignition was caused by lightning. However, the next sentence suggests that the 46% value refers to a correlation coefficient, which does not translate into the proportion of fires caused by lightning. Please rewrite.

L505 Replace "Rothermell (1972)" with "(Rothermell, 1972)"

L525 What is the "burned area parameter"? Do you know how this parameter uncertainty affects the estimated emissions?

L536 Replace "and eventually causing" with "and thereby contributing".

L537 The variable that determines the strength of the carbon sink is NBP, which includes disturbances such as wildfires, rather than NPP. Assessing the impact of wildfires on NPP is valid, but, given the focus of this paper on wildfires, I suggest you also write about NBP.

L544 Replace "circumpolar" with "Circumpolar"

L541 You may want to mention that wildfires and insect outbreaks are not necessarily independent, e.g. the likelihood of wildfire often increases after insect outbreaks.

L545 Cite more recent literature, e.g. the IPCC Sixth Assessment Report, Working Group I, Chapter 4.

L567 For RCP8.5 it should be far more than 2 degrees C. Please provide more accurate numbers here.

L571 Many DGVMs prescribe tree allometry. Is this also the case for SEIB? If yes, then that is why fires have no impact on tree allometry.

L580 A decrease in fire emissions in subsequent years could also be caused by the fact that there is simply less biomass available for burning, since some of the biomass has been burnt during previous years.

L586 It is not clear to me what point you are trying to make. Do you simply want to say that the model is able to account for the effect of fire history on future fires? This is to be expected, and I don't think it deserves a discussion. But I may be missing the point, in which case I suggest you rewrite this paragraph.

L596 You cannot draw the conclusion that a model is accurate, simply because its response is consistent with the change in the perturbation. I would just write that the trend in fire emissions is consistent with the different scenario-dependent trends in radiative forcings.

L599 The projected increase depends on your emission sceanrio, and currently ranges between 1.5 and 6.0 degrees C by 2100 compared to 1850-1900 mean value (IPCC, AR6, WGI, Chapter 4).

**Tables**

Table 2: Replace RCP80 with RCP8.5 and use a consistent notation in table in text (e.g. RCP8.5 rather than RCP85)

Table 4: Replace "Baseline" with "Historical". Note the meaning of +/- (1 or 2 standard deviations?)

**Figures**

Figure 3 Move the text in the Figure caption into the main text as you are describing the figure.

Figure 4 and 5 All panels look very similar. If you want to work out the differences, then I suggest you provide a different type of plot.

Figure 6 Replace "Principle component" with "Principal component". Why do you need to do a PCA here? This would deserve description in the main text, if it is really worthwhile. As I mention above, please verify whether SEIB prescribes allometric relationships, which would explain why the impact of fires is small on allometry. I am not sure whether this analysis is worth including.

Figure S16 Does this figure show that the model tends to overestimate AGB between 25 and 75 Mg ha$^{-1}$? I would also add a diagonal 1:1 line so the reader sees where the dots should ideally be located. What do the colors of the dots represent? Please replace the horizontal axis label *Ha* with *ha*.

Figure 8 Why does SEIB-DGVM has uncertainty bars but GFED4s does not? Also, explain the meaining of the uncertainty bar in the Figure caption. Is this +/- SD across multiple years?

Figure 12 Increase axis value fontsize

---

## Author Comment (AC1)

**Reviewer #1**:

Reza et al. improved the fire module in the SEIB-DGVM model and used the model to project Siberia wildfire dynamics, and emissions under various climate scenarios. I appreciate the valuable model development and analysis, and it provide us important implications on how future climate change affects Siberia boreal forest. The paper is informative and has a lot of important results to discuss, but it is not well-written in terms of smoothness and readability. There are confusions on methodology and data usage. Most importantly, this manuscript over-emphasizes future RCP projections. However, historical validation is not clear and convincing. A more reasonable approach is to first thoroughly demonstrate the model performance during the historical period by comparing it with observations. Then discuss future projections and caveats. Below are my major comments.

**Response:**

Thank you for your valuable feedback on our manuscript. We appreciate your understanding of importance in Siberia's boreal forests under future climate scenarios. We understand your concerns regarding the clarity and readability of the manuscript, as well as the need for a stronger emphasis on historical validation. We will revise the manuscript to improve its smoothness and address any confusions regarding methodology and data usage. Additionally, we will prioritize demonstrating the model's performance during the historical period before discussing future projections. Once again, thank you for your constructive comments and we have replied to your comments and also adjusted our manuscript accordingly.

**1.** There is no quantitative evidence of the model's performance. During the historical period, how did the model simulate burned area, and fire emissions, compared with existing datasets (e.g., GFED) at gridcell level? Plot scatter plots of modeled versus observed variables (burned area, emissions, biomass) will be super helpful. Showing R2 and RMSE of gridcell level comparison in the abstract is highly encouraged. As it is stated in the abstract the motivation for integrating SPITFIRE into DGVM is to improve the accuracy of fire modeling. Therefore, it is also important to show how much improvement has been achieved by SPITFIRE compared with DGVM's default fire model.

**Response:**

Thank you for your valuable feedback on our manuscript. We have validated the model using some benchmark data at longitude average grid cell level. GFED4s: burned area 1996-2016 (Figure S14.b in supplement). GFED4s: burned fraction 1997-2016 (Figure S14.a in supplement), dry matter 1997-2016 (Figure S19 in supplement), $CO_2$ emissions 1997-2016 (Figure 9.a: L445). GBEI: $CO_2$ emissions 2001-2020 (Figure 9.b: L445). ESA Biomass CCI: Aboveground biomass 2010, 2017-2018 (Figure S16 in supplement).

We have added the $R^2$ and RMSE value on the abstract.

In the model, the historical simulation starts from 1850-2005, and the future simulation starts from 2006-2100. We extracted the relevant years from both simulation periods when needed to compare with the benchmark datasets.

We updated the comparison of AGB model and ESA Biomass CCI data: previous figures are based on the longitude average, while the new figures are based on the latitude average. We added double error marks to identify the model uncertainty and the discrepancy with the benchmark datasets. We will replace all of the previous plot with the new plot below.

[Figure]

| Figure S16.a (adjusted) AGB 2010 | Figure S16.b (adjusted) AGB 2017 |

[Figure]

| Figure S16.c (adjusted) AGB 2018 | Figure S14.a Burned Fraction (1997-2016) |

[Figure]

| Figure S14.b Burned Area (1996-2016) | Figure S19 Dry Matter (1997-2016) |

[Figure]

| Figure 9.a GFED4s $CO_2$ (1997-2016) | Figure 9.b GBEI $CO_2$ (2001-2020) |

we added a comparison of the main variables generated in the default model and the improved model in sections 3.1 to 3.4.

Related to the fire estimation improvements of the improved model with the default model, we have added this explanation in sub section 3.2. and figure S5. in the supplement.

[Figure]

[Figure]

We replaced the previous Figure 2 with new Figure 2 above to describe in more detail about the difference and the integration between the default SEIB-DGVM and the SEIB-DGVM SPITFIRE.

**2.** Improvement from annual time step estimate to monthly is an important change. However, most of the fires last less than one month. How does the monthly time step fire module resolve a process that lasts much shorter than a month?

**Response:**

Estimating fire occurrence at a daily timestep would likely decrease the model accuracy (there are just too many factors to expect the model fit perfectly with reality). However, rather than knowing the exact day of the fire, knowing in which month(s) to expect higher fire activity would help to adjust prevention measures. Furthermore, we have calibrated our variable with the benchmark datasets (related to the fire: we used GFED4s and GFED4 datasets) which have the daily time-step data, thus the monthly output is an accumulation of the daily process.

**3.**     Human ignition is considered in E term (equation 2), how about human suppression on fire spread?

**Response:**

Human suppression on the fire ignition is considered in the equation 8 (in the supplement). The human ignition factor (Nih) value will increase as the population increases, but if at a certain population density, the ignition factor will decrease steadily to a value of 0, indicating that there is human suppression on fire. Equation (8) has a maximum value at a population density of 16 $km^{-2}$.

PD is the population density (individuals $km^{-2}$), and aND (ignitions $individual^{-1}$ $d^{-1}$) is a parameter expressing the propensity of people to produce ignition events. aND is and user-definable parameter, with a scale of 0.0 – 1.0, and in this study we adjust the value to 0.7 or 70%  (total of human and unknown caused fire), as the recent research in eastern Siberia Xu *et al.*, (2022), shown that fires over Yakutia, $31.4 \pm 6.8\%$ caused by lightning ignitions, $51.0 \pm 6.9\%$ caused by anthropogenic ignitions, and the last 14.4% unknown cause.

Human suppression on fire is considered/ calculated in the model at the ignition stage only. While at the fire spread, it is only influenced by fuel availability, moisture, and climatic factors (wind). However, the model performs calculations on each individual (0.5 x 0.5) grid cell. Therefore, if observed more broadly, humans have suppression of ignition and fire spread.

**4.**     Methodology, between 2.2 model description and 2.3 model application, there seems missing a section about model calibration. How was the SPITFIRE module calibrated for burned areas or emissions? How was the DGVM model calibrated to capture observed AGB?

**Response:**

Before validating with benchmarks and observational data, we calibrated all major output variables (Burned Fraction, Burned Area, Dry Matter, Aboveground Biomass, $CO_2$ Emissions, $PM_{2.5}$ Emissions, and some forest ecology variables). The emission calibration process is only carried out on $CO_2$ and $PM_{2.5}$ emissions as representative major emissions.

We have added the explanation of the model calibration process in 2.3 Model Calibration, the other subchapters will be renumbered accordingly, 2.4. Model application, and so on.

"We calibrate the improved model by using all of the benchmark datasets (Table 3). The calibration process is done sequentially, from burned fraction, burned area, dry matter, aboveground biomass, burned biomass emissions, and the forest ecology variables (Figure 3). The process is sequential because one variable is used for the calculation of another variable (such as burned fraction and burned area affecting aboveground biomass, forest structure, dry matter, and emissions).  One calibration process is performed with multiple iterations until the output variable has similar numerical values and spatial distribution to

the benchmark data, and the process is repeated for other variables once the previous variable has been calibrated." (L211)

We have adjusted the "Figure 3. Workflow of improving the SEIB-DGVM fire module" by adding a model calibration scheme with benchmark data. In the next manuscript, we will replace Figure 3 with the latest one.

[Figure]

**Figure 3.** Workflow of improving the SEIB-DGVM fire module.

**5.** For future projection, are population density changes and lightning flash changes considered under future scenarios?

**Response:**

In future simulations, the variables of population density and lightning flash remain the same as in historical simulations (constant). This is because the temporal coverage of the datasets are not long enough (LIS/OTD High-Resolution Full Climatology (HRFC) V2.3.2015 (2000-2020) and Gridded Population of the World (GPWv4) (2015) (Table 2).

**6.** Section 2.4, the model is validated by GFED4 burned area and GFED4s burned fraction. However, GFED4s and GFED4 are different because GFED4S include smaller fires. How to reconcile GFED4s and GFED4 during validation?

**Response:**

Yes, GFED4s burned fraction and GFED4 burned area are different because GFED4s includes small fires. We have mentioned this in section 3.1.1 Fire products (L 305: revised manuscript). The validation process is carried out separately between variables (burned fraction and burned area), so there is no reconciliation process needed between benchmark datasets (GFED4s and GFED4).

Based on the validation results, in the comparison of distribution patterns, the results of the burned fraction variable comparison model with GFED4s have a higher value than the burned area variable model with GFED4. Because GFED4s data has a slightly wider distribution pattern because it including small fires.

**7.** Section 3.1-3.4. Historical validation is an important component of this study. I would like to suggest before discussing RCP results thoroughly compare the simulated burned area, emission, and biomass with observations. Discuss model performance and biases at gridcell level, and the regional level.

**Response:**

We have validated the model in section 3.7. We will move this section to the first section.

Regarding the historical validation, there is a difference in that the historical time in the model starts from 1850-2005 (according to the climate input data) and the 2006-2100 data is included in the future simulation, while the benchmark data varies from 1996-2020. (This topic has been mentioned in the 1ˢᵗ comment response). Thus, we extracted the relevant years from both simulation periods when needed to compare with the benchmark datasets.

Instead of grid level validation, we used latitude average comparison to determine the average value, pattern and dynamics of variables at longitude point of view. Furthermore, in accordance of previous studies, model projections are not validated spatially at grid cell level but they validated the model output with observational data in a numerical comparison (specific area, temporal average, and only few variables). e.g. Fig. 3 Verification of the simulated length of fire season using LPJ-DGVM against observations in the sample regions (Thonicke et al., 2001), Fig. 4 Fire return intervals for the period 1987–96 derived from the national fire statistics of forest services and simulated for the same period by the LPJ-DGVM (Thonicke et al., 2001), Fig. 6. Observed (MODIS) versus simulated fire season lengths for biomes (Thonicke et al., 2010), and Fig. 8. (a) Comparison of observed (Ni, 2004) and simulated net primary production in northern China. (b) Comparison of observed (Sukhinin et al., 2004) and simulated area burnt for 1997–2002 (Thonicke et al., 2010).

In this study we compare both (numerical comparison and spatial comparison: average latitude at Siberian level and regional level) for all major variables. Therefore, our validation process is better because it covers all major variables and uses two types of validation. (section 4.1. Paragraph 3).

Below are the validation results for all major variables at the regional level. We divide Siberia into three areas: west region (60°-90°E and 45°-80°N), central region (90°-120°E and 45°-80°N), and east region (120°-180°E and 45°-80°N).

We added this explanation in the Section 2.5. (L258) and all of the results section related to each variables.

[Figure]

[Figure]

| **West region** | **Central Region** | **East Region** |
|---|---|---|

[Figure]

[Figure]

**8.** Section 3.5 Fire-off simulation is not mentioned in the methodology section. When was the fire turned off? Does the model run an extended spinup with fire-off? One would expect much larger aboveground biomass when the fire is turned off.

**Response:**

We have explained the fire-off simulation in the section 2.4. Model application (L244). We ran the fire-off simulation separately from the fire-on simulation (starting from spinup until future simulation using the same protocols: model settings and climate input dataset). The aboveground biomass was indeed larger when the fire was off.

We explained the comparison result in the section 3.5 Forest ecological variables under fire-on and fire-off simulation.

**9.** Figure 7 showing spatial coverage is confusing because GFED4s provide burned fraction, which should be directly compared with SEIB-DGVM at gridcell level. Figure 8, a and b are duplicated, both showed long-term monthly average dry matter emission. Figure 9, why dry matter emission is perfectly simulated by SEIB-DGVM (shown in Figure 8), but the performance of $CO_2$ emission is much worse. It seems inconsistent. What does it look like, if plot a spatial average comparison of DM emissions between SEIB-DGVM with GFED4s?

Thank you for the the comments. We will address each point separately:

a. Our purpose of showing Figure 7 is only to determine the agreement of burned fraction model variables with GFED4s data (ignoring the value of each grid), we did as in Fig. (b) Comparison of observed (Sukhinin et al., 2004) and simulated burnt area for 1997-2002 (Thonicke et al., 2010).
   We have also compared the burned fraction variable (and all variables) with benchmark data at the grid level (mean latitude: Figure S14.a and updated plots are in the response to first comment).
b. Yes, figure 8 a. and b. show the same data, we will remove figure 8.b in the adjusted version of the manuscript.

   Figure 8 is a comparison of annual average extracted values of modeled dry matter emissions with GFED4s data. Numerically and on a large scale (entire study site: Siberia) and long-time horizon (monthly average), the model is able to produce values that are very close to the benchmark data. The model is able to simulate well because each variable has been calibrated with benchmark data with several iterations to produce good data.

   In contrast, Figure 9 is a spatial comparison, where the spatial distribution pattern of emissions (all fire-related variables), has a similar distribution pattern because it comes from fire variables (burned fraction and burned area) which are calculated with specific equations to produce emission variables.

   The above is also explained as a limitation of the model in section 4.1 (L 574), that the model is able to simulate numerically well, but the distribution pattern is highly dependent on the initial variable (fire), so adjusting ignition and related factors can produce a better distribution pattern (closer to benchmark data), which will affect other variables (dry matter emissions, agb, forest ecology, and all burning biomass emissions).

   As to the last point, we have compared the modeled dry matter emission variables with GFED4s (Figure S19 in the supplement, and the new plot is in the response to the first comment).

---

## Author Comment (AC2)

**Reviewer #2**:

**General comment**

The manuscript titled "Future prediction of Siberian wildfire and aerosol emissions via the improved fire module of the spatially explicit individual-based dynamic global vegetation model" by Reza Kusuma Nurrohman and colleagues evaluates the implementation of the SPITFIRE fire module into the SEIB dynamic global vegetation model for Siberia. The model is forced with reanalysis data provided by CRU and with global climate model data produced by MirocAR5 for the historical period as well as four emission scenarios until 2100. The manuscript compares simulated and observed fire-related variables and evaluates the impact of wildfires on emissions and tree mortality. The text would benefit from restructuring and enhanced focus on the aspects that really matter. Large parts of the results section are very descriptive, and it is not always self-evident why these results matter and what they imply. It is worthwhile for the authors to revisit the entire text and carefully assess what information is really worth communicating. Additionally, restructuring the results section such that it first evaluates the model during the historical period and then describes future projections afterwards is recommended. Furthermore, adding an overview figure or table that systematically compares and summarizes model performance for the old and new versions of the model is suggested. This is important to convince the reader that the new version is indeed superior. Finally, many of the literature references are outdated and should be replaced with more recent ones. I recommend that the manuscript may be considered for publication in Biogeosciences after major revision. Please find my detailed comments below.

**Response:**

Thank you for your thorough review of our manuscript, and we appreciate your constructive feedback and insightful suggestions for improvement.

We understand your suggestion regarding the restructuring of the manuscript to enhance its clarity and focus on key aspects. We will carefully reassess the content of the results section to ensure that it effectively communicates the significance of our findings and their implications. Additionally, we acknowledge the importance of prioritizing the evaluation of the model's performance during the historical period before discussing future projections. We will revise the manuscript accordingly to improve its coherence and readability.

Furthermore, we appreciate your recommendation to include an overview figure or table that systematically compares and summarizes the model performance for the old and new versions. This will indeed provide a clearer demonstration of the improvements made in the new version of the model.

Regarding the outdated references, we will thoroughly review the literature and update the references with more recent ones to ensure the accuracy and relevance of our citations.

We are grateful for your suggestion consideration of our manuscript for publication in Biogeosciences after major revision. Your feedback will undoubtedly contribute to enhancing the quality of our work, and we are committed to addressing all your suggestions in our revised manuscript.

Once again, thank you for your time and valuable input.

**Detailed Comments**

L1 Note that when you are assessing the impact of emission scenarios, you should refer to projections rather than to predictions. Also, but this is completely up to the authors, my personal recommendation is to choose a more catchy title. For instance, you could simply shorten the tile to: "Future projections of Siberian wildfire and aerosol emissions"

**Response:**

We appreciate your suggestion to use the term "projections" instead of "predictions" when referring to the impact of emission scenarios. We will make sure to adjust the terminology accordingly to accurately reflect the nature of our analyses. Additionally, we are grateful for your recommendation to consider a more catchy title for our manuscript. We have adjusted our manuscript title according to your suggestion.

L17 In statistics, we distinguish between accuracy and precision. You don't want precise values, rather you want accurate values.

**Response:**

Thank you for your insightful comment regarding the distinction between accuracy and precision in statistics. We have adjusted this word.

L21 Before writing about the future changes, please add a sentence that describes how well the model reproduces historic values.

**Response:**

We have implemented your recommendation and have incorporated a sentence to address this aspect in our manuscript.

The adjusted sentence now reads as follows:

"The model is able to reproduce historical data well compared to the benchmark datasets, based on the validation results spatially, has the following values: AGB ($R^2$=0.43, RMSE=21.9 Mg ha$^{-1}$), burned fraction ($R^2$=0.75, RMSE=0.01), burned area ($R^2$=0.609, RMSE=690 ha), dry matter emission ($R^2$=0.63, RMSE=0.01 Kg DM m$^{-2}$), GFED4s $CO_2$ emission ($R^2$=0.608, RMSE=52.4 Tg), GBEI $CO_2$ emission ($R^2$=0.67, RMSE=69.5 Tg). Spatially, the model is able to produce data with the same distribution pattern with a

value of 61.8%. Numerically and on a long-term average, the model is able to produce values with very high accuracy of around 99% compared with the benchmark datasets."

L23 The absolute numbers are not very intuitive. Please provide the relative changes in emissions and trends, instead.

**Response:**

We have adjusted the text accordingly to provide relative changes in emissions and trends. We have added a percentage increase trend each year of $0.0264 \pm 0.01$ % year$^{-1}$ (under the same RCP scenarios all emissions have the same increase trend). Under the RCP 8.5, has the highest percentage of increasing trend.

The adjusted sentence (L24) as follows: " Under the Representative Concentration Pathways 8.5 climate scenario, we estimated that the $CO_2$, CO, $PM_{2.5}$, total particulate matter (TPM), and total particulate carbon (TPC) emissions in Siberia will continue to increase annually until 2100 by $0.0264 \pm 0.01$ % year$^{-1}$ or individually by $214.4 \pm 79.4$, $17.16 \pm 6.35$, $2.8 \pm 1.03$, $2.1 \pm 0.78$, and $1.47 \pm 0.54$ Gg species year$^{-1}$, respectively."

L38 Please find more recent references here. A good place to start would be the 6th Assessment Report, Working Group I, Chapter 5.

**Response:**

Thank you for your suggestion. We have read the 6th Assessment Report, Working Group I, Chapter 5, and other recent literature to identify appropriate references for inclusion in this section.

We have adjusted the reference in L41 (revised manuscript) into: "which will cause more emissions from biomass burning (Flannigan et al., 2009; Gauthier et al., 2015) and human activity (Schmoldt et al., 1999; Hantson et al., 2016)."

L43 Currently this sentence reads as if the fact that small fires matter is consistent with a positive trend in fires. This does not make sense to me. Please replace "This finding" with something more explicit.

**Response:**

We have corrected the word "This finding" to the following: "Increased incidence of wildfire in some regions around the world"

We hope this clarified the meaning of the sentence.

L61 Please use more recent references if you want to provide numbers that represent "current estimates", e.g. Friedlingstein, Pierre, et al. "Global carbon budget 2023." Earth System Science Data 15.12 (2023): 5301-5369.

**Response:**

Thank you very much for the suggestion, we have checked the publication of Friedlingstein, Pierre, et al. "Global carbon budget 2023." Earth System Science Data 15.12 (2023): 5301-5369.

However, we found it difficult to extract by region (Siberia) to compare with global data. So we have replaced the use of the words "current estimates" and replaced reference [1] with [2 and 3].

[1] Tchebakova, N.M., Monserud, R.A. and Nazimova, D.I. (1994) 'A Siberian vegetation model based on climatic parameters', Canadian Journal of Forest Research, Volume 24,. Available at: https://doi.org/10.1139/x94-208

[2] Kasischke, E.S. (2000) 'Boreal Ecosystems in the Global Carbon Cycle', (Houghton), pp. 19–30. Available at: https://doi.org/10.1007/978-0-387-21629-4_2.

[3] Hantemirov, R. M., Corona, C., Guillet, S., Shiyatov, S. G., Stoffel, M., Osborn, T. J., Melvin, T. M., Gorlanova, L. A., Kukarskih, V. V., Surkov, A. Y., von Arx, G., and Fonti, P.: Current Siberian heating is unprecedented during the past seven millennia, Nat. Commun., 13, 1–8, https://doi.org/10.1038/s41467-022-32629-x, 2022.

So the latest adjusted sentence is as follows (L64):

Boreal vegetation store between 17% of the world's carbon, yet encompasses almost 30% of all terrestrial carbon stocks (Kasischke, 2000; Gauthier et al., 2015), and two-thirds are located in Siberia, Russia (Shvidenko and Nilsson, 2003).

L122 Please find more recent reference. The study is 20 years old and much has happened in the last two decades.

**Response:**

We have adjusted the sentence into the following sentence (L125):

In the Arctic, a rapid warming trend has been observed, and the increase in temperature over the last 20 years of the 20th century was 2 to 3 times higher than the global average, while in the first 20 years of the 21st century, it exceeded four times (Chylek et al., 2022). This enormous increase in temperature in Siberia, affecting the duration and speed of snowmelt and accelerates thawing of carbon-rich permafrost (Natali et al., 2019; Schuur et al., 2015; Nitzbon et al., 2020), which results in drier ground cover, an increased frequency of wildfires, longer fire seasons, and increased ignition sources (Kharuk et al., 2022).

L124 Please don't cite the entire report. Specify the working group and chapter. Also, replace this reference with the most recent assessment report.

> **Response:**
>
> We tried to find a similar statement in the recent assessment report, but we couldn't find it. So, we decided to use another recent reference.
>
> The adjusted sentence is as follows (L130):
>
> These changes may result in a new climate state in which heatwaves as well as the associated the occurrence of wildfires may become routine and more severe (Hantemirov et al., 2022; Landrum and Holland, 2020). Produced emissions from thawing permafrost and from wildfire are likely to feed into the global carbon cycle's feedback on climate change (Schuur et al., 2015), and triggering further warming trends globally (Schimel et al., 2001; Kharuk et al., 2011; Krylov et al., 2014).

L146 The notation suggests that E is a function of nig. What is nig?

> **Response:**
>
> Yes, $E(n_{ig})$ is the expected number of fire ignition event. Ignition event $E(nig)$ is the sum of independent calculations of lightning-caused ($n_{il}$) and human-caused ($n_{ih}$) fire ignition events, ignoring stochastic variations (Thonicke et al., 2010).
>
> $$Enig = nih + nil \qquad (6)$$
>
> This is explained in section 2.2.1. Burned area calculation input variables in supplement (L8).

L148 Please consider using a bar rather than an underscore to indicate the mean area.

> **Response:**
>
> Thank you, now it's has been adjusted.

L149 A grid is composed of gridcells. Please replace "grid" with "gridcell".

> **Response:**
>
> it has been adjusted.

L264 Please include an analysis that shows how the inclusion of SPITFIRE affects the model's ability to reproduce the observed patterns during the historical period. So far you only write that the pattern of the new model version is different.

**Response:**

Related to the comparison with the benchmark dataset of all of the major variables has been explained in the section 3.1. Improved model validation

Now. we have compared the burned fraction variable between the benchmark data (GFED4s), improved SEIB-DGVM (SPITFIRE), and default SEIB-DGVM. We will add the explanation in the beginning of the section 3.2 Burned Fraction.

[Figure]

Figure Rev1.

Improved model is able to produce burned fraction variables better than the default model. Based on the spatial comparison of the 1997-2016 average burned fraction variables GFED4s, SEIB-DGVM SPITFIRE, and Default SEIB-DGVM, it is known that SEIB-DGVM SPITFIRE is able to produce data with a similarity of 75% with GFED4s data, while the default model is at 68% (Figure Rev1). *[... continue]*

L275 Do lowest and highest values refer to spatially averaged annual mean values? Please explain.

**Response:**

The lowest and highest values in this sentence refer to figure S7. Annual (total) burned area projection (2006-2100). We have adjusted the first sentence to give more explanation about the burned area (L434).

"Overall, under all the scenarios, the **annual total burned area** exhibited the same increasing trend, with the RCP4.5 scenario reaching the **highest value**. Under the RCP4.5 scenario from 2006 to 2100, the burned area has an average value of 1457.6 Ha grid$^{-1}$ year$^{-1}$ and is projected to increase with values of 80.4 to 83.7 x 10$^5$ hectares (Figure S6.b)."

L284 Does "uniform spatial distribution" mean that the amount of burned biomass is equal across gridcells? Please clarify.

**Response:**

Uniform spatial distribution -> our intention is to explain that the fire variables (burned fraction, burned area, and burned biomass) have the same distribution pattern, which means that the three variables are confirmed to be mutually integrated because the calculation process comes from the first fire variable, burned fraction. This is consistent with the GFED4s benchmark data, where the burned fraction and dry matter emission variables have the same distribution pattern.

Whereas in the default model, the distribution pattern between the burned fraction (Figure S4.a-d) and burned biomass (Figure 4.a-d) variables are different.

We have adjusted the sentence (highlighted sentences is for this comment adjustment, whole paragraph is also adjusted with **comment L292**) as follows:

"The improved model confirmed uniform spatial distribution patterns for the fire variables: burned fraction (Figure S4.b), burned area (Figure S6.a), and burned biomass (Figure 4.b). All of the improved module fire variables confirmed to be mutually integrated because the calculation process comes from the first fire variable (burned fraction). Compared to the improved model, the spatial distribution pattern of the burned biomass variable from the default model was wider and spread across the entire Siberia region (Figure 4.a). The spatial distribution pattern of burned fraction (S4.a-d) and burned biomass (Figure 4.a) in the default model is different and exhibited a box-like pattern in the center of the map.

Under all the RCP scenarios from 2006 to 2100, the burned biomass variable from the default model exhibited a decreasing trend (Figure 4.c). Under the RCP8.5 scenario, from 2006 to 2100, the value decreased from 5.09 to 5.05 kg DM m$^{-2}$, with an overall mean value of 5.07 kg DM m$^{-2}$. Meanwhile, the burned fraction product from default model shows an increasing trend (Figure S4.c), this indicates a misalignment in the default model between the burned fraction and burned area variables. On the other hand, the burned fraction (Figure S4.d), burned area (Figure S6.b), and burned biomass (Figure 4.d) variables in the improved model all have an increasing trend, indicating harmony between the three variables. Burned biomass variable under the RCP8.5 scenario from 2006 to 2100, the lowest and highest values were 8.0 and 8.71 kg DM m$^{-2}$, respectively, with an overall mean value of 8 .52 kg DM m$^{-2}$. The twenty-year variations and their trends of dry matter emissions up to 2100 are 8.37 ± 0.07 (3.3%), 8.46 ± 0.04 (0.7%), 8.53 ± 0.06 (1.3%), 8.52 ± 0.05 (-0.68%), 8.57 ± 0.08 (1.07%) (Figure S32)."

L290 Do you have any thoughts on what causes the box-like pattern? Also, why does the default model version produce values across the entire domain, while the SPITFIRE model version has many gridcells with no values?

**Response:**

We suggest the box-like pattern in the default model is caused by the spatial distribution pattern of the aboveground biomass variable (Figure 5.a). Because of the high availability of AGB in that area (box-like area) and the spread of fire (burned fraction) covering that area, the distribution pattern of burned biomass variable also has the same pattern (Figure 4a.a-d).

In the default model, the model produces fire variables that cover the entire domain because the default fire module only considers fuel availability, moisture content, and minimum threshold of fuel availability: 200 g C m$^{-2}$ (Sato et al., 2007). Therefore, if a particular grid has dry fuel above the minimum threshold, a fire will occur, and it can ignite/ spread to neighboring grids.

While in the improved model, we add the fire ignition factors of anthropogenic ignition (population density) and natural ignition (lightning flash) calculation. So that even though the surrounding grid has dry fuel availability, if there is no ignition, the fire does not occur in that area.

Regarding the SPITFIRE module output on some variables that have areas that have no value, originally, they had a value of 0, but I removed it to n.a. because it is to match the data pattern of the two fire benchmark data (GFED4 and GFED4s).

L292 You write that "from 2006 to 2100, the value decreased from 5.09 to 5.05 kg DM m−2 ". Do the values refer to 2006 and 2100? Since there may be considerable variability, it may be better to compare 20 year averages instead, or calculate the trend.

**Response:**

Yes, the values are referred to 2006 and 2100. We have adjusted this by adding a comparison of the 20-year average and the trend. However, we only discussed the data in the improved model, whereas in the original manuscript the last paragraph discussed the data in the default model.

We have adjusted the wording of this paragraph in accordance with this comment and the previous comment.

The adjusted paragraph and supporting figures are as follows (L446):

"Under all the RCP scenarios from 2006 to 2100, the burned biomass variable from the default model exhibited a decreasing trend (Figure 4.c). Under the RCP8.5 scenario, from 2006 to 2100, the value decreased from 5.09 to 5.05 kg DM m$^{-2}$, with an overall mean value of 5.07 kg DM m$^{-2}$. Meanwhile, the burned fraction product from default model shows an increasing trend (Figure S4.c), this indicates a misalignment in the

default model between the burned fraction and burned area variables. On the other hand, the burned fraction (Figure S4.d), burned area (Figure S6.b), and burned biomass (Figure 4.d) variables in the improved model all have an increasing trend, indicating harmony between the three variables. Burned biomass variable under the RCP8.5 scenario from 2006 to 2100, the lowest and highest values were 8.0 and 8.71 kg DM m$^{-2}$, respectively, with an overall mean value of 8.52 kg DM m$^{-2}$. ==The twenty-year variations and their trends of dry matter emissions up to 2100 are 8.37 ± 0.07 (3.3%), 8.46 ± 0.04 (0.7%), 8.53 ± 0.06 (1.3%), 8.52 ± 0.05 (-0.68%), 8.57 ± 0.08 (1.07%) (Figure S32).==”

[Figure]

Figure Rev2.

L301 What are "estimation parameters"?
   **Response:**

   in this case -estimation parameters- means that the Aboveground biomass estimation process in the default model and improved model uses the same calculation method.

L305 What do you mean by common roots? Please rewrite this sentence.
   **Response:**
   In this sentence "common roots" means that the default model and improved model use the same basic aboveground calculation method, but the improved model is adjusted for the impact of fire distribution patterns (based on natural and anthropogenic ignition factors). So even though they use the same aboveground biomass calculation process, they have differences in distribution patterns (Figure 5) and trends in annual average dynamics (Figure S10).

We have adjusted the paragraph in section 3.5 (revised manuscript). as follows (L 459):

The aboveground biomass calculations in the default model and improved model used the same ==estimation process== because the trunk biomass in the SEIB-DGVM included coarse root biomass; therefore, only approximately 2/3 of the trunk biomass was classified as aboveground biomass (Sato et al., 2007). ==However, during the calibration of the aboveground biomass variable with the ESA Biomass CCI benchmark dataset, we made adjustments of the calculation impact of fire and its distribution pattern (based on natural and anthropogenic ignition factors) on the availability of aboveground biomass.==

L307 I am not sure what you are trying to communicate here. You find that a high fire frequency coincides with a low AGB. Do you think this makes sense because fires burn trees and therefore biomass is low? Or is it counter-intuitive because the more biomass, the more fuel load, the higher the probability of fires? How do you interpret your results?

**Response:**
My point was to convey your first explanation, that the AGB values (in the improved model: especially in some parts of central Siberia) are lower than in the default model because of the high frequency of fires burns biomass.

I have adjusted the whole paragraph in section 3.4 just like the answer in **comment L305 (L464: revised manuscript).**

L314 See comment as for L292.

**Response:**
We have adjusted the paragraph combined with the previous adjustment as follow (L476):

==The twenty-year variations and their trends of aboveground biomass up to 2100 are $69.09 \pm 0.168$ (0.13%), $69.04 \pm 0.151$ (-0.54%), $68.88 \pm 0.128$ (-0.07%), $68.37 \pm 0.226$ (-0.76%), $68.36 \pm 0.189$ (0.63%) (Figure Rev3).==

[Figure]

Figure Rev3.

L321 What does "full" refer to in this context?

**Response:**

"Full" simulation in this context means a complete simulation covering all phases (spinup, historical and future with all four RCPs scenarios) with fire on and fire off simulation, and with 5 repetitions of 5 times to get the average of the random runs values. Random runs: 1) random distribution of different vegetation seeds per gridcel, 2) Tree establishment, and 3) Individual trees that died from fire.

We have adjusted the sentence as follow (L484):

"We conducted complete simulations under fire-on and fire-off modes to compare and assess vegetation dynamics during forest fires. …."

L322 Why is NPP "the most measurable element of the carbon cycle"? NPP is actually difficult to measure at a canopy scale. The eddy covariance method for instance, measures NEE, from which GPP and ecosystem respiration are then derived. You could also make the case that carbon pools are easier to measure than carbon fluxes.

**Response:**

1. Running (2022), stated that NPP has been the most easily measured intermediate step because it represents tangible, visible plant biomass, while NEE is the final net exchange of $CO_2$. NPP has been measured by ecologists and agronomists for centuries, but measurement of NEE was only routinely possible with the development of

automated eddy covariance flux towers in the 1980s, of which more than 500 now operate around the world.

Refference
Running, S.W. (2022) 'GLOBAL ARIDIFICATION AND THE DECLINE OF NPP: A COMMENTARY on Projected increases in global terrestrial net primary productivity loss caused by drought under climate change by Dan Cao, Jiahua Zhang, Jiaqi Han, Tian Zhang, Shanshan Yang, Jingwen Wang, Foyez', Earth's Future, 10(11), pp. 1–3. Available at: https://doi.org/10.1029/2022EF003113.

2. However, we have adjusted the sentences with your suggestion and the reference as follow:

We conducted complete simulations under fire-on and fire-off modes to compare and assess vegetation dynamics during forest fires. Assessing vegetation dynamics can be done by understanding the carbon pools in the certain region or globally, where carbon pools are easier to measure than carbon fluxes. In this study, the net primary production (NPP) is used as a reference variable because is an important metric of the global carbon cycle (Running, 2022) and measures the rate of global plant growth. Based on the comparison of results between the fire-on and fire-off simulations, the NPP variable under all of the RCP scenarios shows a downward trend with some small fluctuations. Under the RCP8.5 scenario, an average loss of $319.3 \pm 28.2$ g C m$^{-2}$ year$^{-1}$ occurred during 1997–2100 (Figure S23.a).

L325 Please add relative measures here as well, as absolute values alone are not very intuitive.

**Response:**
We have adjusted the sentence, as follows (L489):

"Under the RCP8.5 scenario, an average loss of $319.3 \pm 28.2$ g C m$^{-2}$ year$^{-1}$ occurred during 1997–2100 (Figure S23.a). …"

L337 Why does this matter and what do you conclude from this?

**Response:**
To (internally verify the model) show that both simulations (fire-on and fire-off) individually run well. The comparison between the three tree structure variables (crown area, tree DBH, and tree height) shows a good relationship with an average value of 98%.

(Comparison of tree structure simulation results with observational data is discussed in section 4.2 (L655).)

L349 Why does this matter and what do you conclude from this?

**Response:**

L349: "We extracted the AGB data in the marked area with coordinates of 130-142°E and 65-80°N (Figure S23) and discovered that the average simulated aboveground biomass in the area was 65.59 g C m⁻² from 1997 to 2023, compared to 416.4 g C m⁻² in the one-grid high-AGB areas."

This point is important because it will be discussed in relation to the distribution pattern of the fire variable (and its affected the spatial distribution of other variables, up to the projection of burning biomass emissions).

Further discussion is provided in section 4.1. paragraph 3 (L569), that the low AGB value in the area caused the fire can not to be ignited (due to the AGB value below the fuel load threshold: 200 g C m⁻² (Sato et al., 2007). Compared to the benchmark data, fire was detected in these areas (GFED4 and GFED4s), and these areas had an average AGB of 2,309.67 g DM m⁻².

L360 Does the "annual average spatial distribution" expressed in percentage refer to the correlation coefficient R of annual mean values of individual gridcells? If that is the case, please express your results in terms of fraction (R 2 = 0.79). Please do that consistently throughout the text.

**Response:**
Yes, it does. We have adjusted this sentence (L360) and the whole paragraph according to your suggestion.

The entire adjusted paragraph in section 3.1.1. Fire products (revised manuscript) is as follows (L304):

"We compared the annual average distribution patterns of burned fraction variable (1997-2016) in the SEIB-DGVM SPITFIRE and GFED4s data, and most patterns differed only in eastern Siberia (Figure 7, Figure S10). Compared to the burned fraction variable, burned area GFED4 has a smaller distribution pattern because it does not consider small fires (Figure S9.a). Comparison analysis of burned fraction variables between SEIB-DGVM SPITFIRE and GFED4s showed a linear relationship with a correlation coefficient of R=0.87 (R²=0.75) (Figure S11.a). Similar to the comparison with GFED4s, the comparison of SEIB-DGVM SPITFIRE output of burned area variables with GFED4 data (1996-2016) shows a linear relationship with a correlation coefficient of R=0.78 (R²=0.61) (Figure S11.b). Furthermore, in the three regions (west, central and east), the partial comparison of the burned fraction variable with GFED4s showed values of R²=0.68, R²=0.51, and R²=0.58 (Figure S13), while for the burned area variable showed values of R²=0.51, R²=0.54, and R²=0.506 (Figure S14), respectively. The burned fraction correlated better because both the GFED4s and the

model's fire module considered small fires; many scattered fire data with values less than 0.1 and approximately 0.1 were found in both the model's output and the GFED4s data.

The fire products (burned fraction and burned area) in the improved model have the same spatial distribution because they are calculated based on one core variable (Eq. 1). However, the spatial distributions of GFED4s (burned fraction) and GFED4 (burned area) differ for two reasons: first, because GFED4 does not consider small fires (Giglio et al., 2013) while GFED4s does, and second, because GFED4s use the modified burned fraction equation, which is able to calculate the exact fire fraction and fuel load (not uniformized) in a grid cell (Van Der Werf et al., 2017)."

L369 Please quantify how similar the modeled and the observed values are using a statistical metric, such as a correlation coefficient, bias, or RMSE.

**Response:**
We have adjusted this sentence by adding RMSE (L322):

"The mean average burned fraction during 1997-2016 was 0.0137 in the simulations, compared to the GFED4s, which recorded the same value of 0.0137 with an RMSE value of $7.2 \times 10^{-4}$. Furthermore, the mean average burned area of the model in 1996-2016 was 1428.5 ha grid$^{-1}$ year$^{-1}$, compared to the GFED4 burned area data, which closely recorded value of 1425.1 ha grid$^{-1}$ year$^{-1}$ by an RMSE value of 70.2 ha grid$^{-1}$ year$^{-1}$. "

L371 Since both values are identical the word "while" does not fit here. Same applies to the next sentence. Please rephrase.

**Response:**
We have rephrased the sentences (L322).

"The mean average burned fraction during 1997-2016 was 0.0137 in the simulations, compared to the GFED4s, which recorded the same value of 0.0137 with an RMSE value of $7.2 \times 10^{-4}$. Furthermore, the mean average burned area of the model in 1996-2016 was 1428.5 ha grid$^{-1}$ year$^{-1}$, compared to the GFED4 burned area data, which closely recorded value of 1425.1 ha grid$^{-1}$ year$^{-1}$ by an RMSE value of 70.2 ha grid$^{-1}$ year$^{-1}$. "

L402 Is this total annual rainfall? Is this really rainfall or precipitation, which is the sum of rainfall and snowfall?

**Response:**
It's based on the reference, in the previous sentence, they mentioned that its precipitation. I suggested it was mixed (sum) of rainfall and snowfall. Thank you for the detailed review.

I have adjusted the sentence as follow (L352):

Severe wildfires in 2003 were due to low precipitation, as total precipitation reached only 36.0 mm in the Buryatia Republic and 45.7 mm in the Chita Oblast between August 2002 and May 2003 (IFFN, 2003).

L403 This sentence seems to contradict the previous sentence, with two different rainfall values for the same region and time period. Please rewrite.

> **Response:**
> It looks like some words were changed during the editing process and we did not notice the discrepancy. Our intention is to compare with the normal condition/ annual average (how much the precipitation in those regions)
>
> we have adjusted as follows (L354):
>
> "While the 41-year average precipitation between August and May (1981-2022), in the Buryatia Republic was approximately 332.23 mm, and in the Chita Oblast was approximately 119.45 mm. Thus, the low precipitation in 2003 was an anomaly outside of the annual average range."

L417 The sentence does not make any sense to me. Why would $CO_2$ emissions be representative for other GHG emissions, simply because most of the mass emitted is $CO_2$? The next sentence on the emission factor gives a better rational. I suggest you delete the first sentence and rewrite this part. Also, I suggest that you remind the reader what the other emissions are.

> **Response:**
> Thank you very much for the suggestions,
>
> We have adjusted the initial paragraph to be as follows (L368: revised manuscript):
>
> 'Emissions from biomass burning contribute significantly to the global budget for residual gases and aerosols that affect the climate. It's estimated that biomass burning contributed up to 50% of global CO and $NO_x$ emissions in the troposphere (Galanter et al., 2000), and the most emitted gas during biomass burning is $CO_2$ (Ritchie et al., 2020). Since $CO_2$ emissions are the primary emissions that contribute to climate change, it is critical to assess and monitor them continuously.
>
> In this study, out of 33 projected emissions (Table 4 and Table S4), we validated the $CO_2$ variable that able to represent all projected emissions because all estimated emissions are derived from the same burned dry matter variable, which differs only in the emission factor value of each gaseous emission. "

L425 What is the difference between the $CO_2$ emission values given here and the ones quoted in the previous sentence? Both are described as annual average values.

> **Response:**
> Technically, the first sentence is obtained based on the annual mean data (each source: 1997-2020) ± standard deviation of the three annual extracted data (Table S3). Meanwhile, the second sentence is the mean ± standard deviation of the spatial data (whole data from the 1997-2020) of those three data.
>
> The word "mean" was missing in the second sentence, we corrected it now (L374)..
>
> "The highest annual average value of $CO_2$ emissions from 1997 to 2020 is from GFED4 data, followed by SEIB-DGVM SPITFIRE and then the GBEI product, with values of $105.64 \pm 50.69 \times 10^{13}$ g $CO_2$, $76.41 \pm 0.87 \times 10^{13}$ g $CO_2$, and $62.4 \pm 26.09 \times 10^{13}$ g $CO_2$,

respectively (Table S3). …… However, when comparing the mean of averaged $CO_2$ emissions all the years from 1997 to 2020 of SEIB-DGVM SPITFIRE, GFED4s, and GBEI, we found that they were similar: $141.1 \pm 11.5$ Gg $CO_2$ year$^{-1}$, $157.2 \pm 14.8$ Gg $CO_2$ year$^{-1}$, and $148.7 \pm 7.12$ Gg $CO_2$ year$^{-1}$, respectively."

L434 Equation 29 is located in the supplementary material. If you want to use this equation in your argument, I suggest you describe it in the text. Otherwise it is difficult to follow your argument, unless one reads the corresponding text in the supplementary material.

> **Response:**
> Thank you very much for your suggestion, we will merge sub section **"2.2.8. Trace gas and aerosol emissions"** in the supplement into section **2.2. Improved fire module principles** in the main manuscript.

L454 The low-emission area only contains a few gridcells, so I don't think these values are very meaningful. To my understanding, Figure 11 shows CO2 emissions associated with fires. Why then do some grid cells have negative values? Also, the chosen color legend makes it difficult to determine whether the emissions are low but positive or negative. Finally, the legend says that the figure shows annual average differences 1996-2100. Differences with respect to what?

> **Response:**
> Negative value mean being a carbon sink. However, in the revised manuscript we didn't use this low-high emission area classification anymore. We decided to explain and discuss the emission from the total area (there will be no classification for high and low emission area in the next manuscript version).

L501 You arite that the "lighting flash rate affected fire ignition in Siberia by 46%". This sounds as if 46% of the fire ignition was caused by lightning. However, the next sentence suggests that the 46% value refers to a correlation coefficient, which does not translate into the proportion of fires caused by lightning. Please rewrite.

> **Response:**
>
> We have rewritten this paragraph as follows (L558):
>
> ." This is confirmed by the comparison of the fire variable with the ignition factor variables: the comparison of the burned fraction variable with the lightning flash strikes variable shows a strong correlation of 0.68 ($R^2$=0.45), and the comparison of the burned fraction variable with the population density variable shows a correlation of 0.24 ($R^2$=0.06) (Figure S8). "

L505 Replace "Rothermell (1972)" with "(Rothermell, 1972)"

> **Response:**
> it has been adjusted.

L525 What is the "burned area parameter"? Do you know how this parameter uncertainty affects the estimated emissions?

**Response:**

The parameters are mentioned in chapter 2. Methods (Kasischke and Bruhwiler, 2003). The first parameter is (in?) the parameters for calculating the variable is **biomass burning carbon release (Ct)**, namely total burned area (A), average aboveground biomass density (Ba), aboveground vegetation carbon fraction (assumed to be 0.45) (fca), fraction of aboveground vegetation consumed during fires (βa), carbon density (Cg) and fraction of organic mat carbon consumed during fires (βg) (Kasischke and Bruhwiler, 2003).

We have added the section 4.4. Model uncertainty, to discuss more about this matter

We experienced exactly the same thing as described by Kasischke and Bruhwiler, 2003, the input data and parameter adjustments greatly affect the final output, which is the projected emissions from burned biomass. First, the distribution pattern of the fire variable is very influential, because the distribution pattern of emissions for all gases estimated will be the same. Secondly, we also explained in L569 that the reason why the fire could not be ignited in the eastern Siberia area was due to the low AGB level generated by the model in that area and above the minimum threshold fuel load setting. Thus, this range uncertainty is still an opportunity for further model improvement in future studies.
* * *
We have added the section 4.4. Model uncertainty, to discuss more about this matter

L536 Replace "and eventually causing" with "and thereby contributing".

**Response:**
we have adjusted the sentence according to your suggestion.

L537 The variable that determines the strength of the carbon sink is NBP, which includes disturbances such as wildfires, rather than NPP. Assessing the impact of wildfires on NPP is valid, but, given the focus of this paper on wildfires, I suggest you also write about NBP.

**Response:**

Thank you very much for the constructive suggestion, we have adjusted by adding the NBP variable in L601:

"By default, both SEIB-DGVM and SEIB-DGVM SPITFIRE defaults do not have NBP output, but the NBP input variable is available. We have added the NBP output and have re-run the simulation using this equation to estimate the NBP:

NBP = NPP - Rh - fire flux (Chapin et al., 2006; Fisher et al., 2014) ''

--

Chapin, F. S., et al. (2006), Reconciling carbon-cycle concepts, terminology and methods, Ecosystems, 9, 1041–1050, doi:10.1007/s10021-005-0105-7.

Fisher, J. B., D. N. Huntzinger, C. R. Schwalm, and S. Sitch (2014), Modeling the terrestrial biosphere, Annu. Rev. Environ. Resour., 39,91–123, doi:10.1146/annurev-environ-012913-093456.

--

We have also obtained the simulation results of NBP under all RCP scenarios, which we will discuss in the new version of the manuscript.

L544 Replace "circumpolar" with "Circumpolar"
> **Response:**
> Thank you, now has been adjusted.

L541 You may want to mention that wildfires and insect outbreaks are not necessarily independent, e.g. the likelihood of wildfire often increases after insect outbreaks.
> **Response:**
> Thank you very much for your suggestion, we have adjusted the sentence as follows (L627):
>
> --
>
> ”Wildfires and insect outbreaks are not necessarily independent, there is a likelihood of wildfires often increasing or decreasing after insect outbreaks (Meigs et al., 2015, 2016).”
>
> Meigs, G.W., Campbell, J.L., Zald, H.S.J., Bailey, J.D., Shaw, D.C., Kennedy, R.E., 2015. Does wildfire likelihood increase following insect outbreaks in conifer forests? Ecosphere 6, 1–24. https://doi.org/10.1890/ES15-00037.1
>
> Meigs, G.W., Zald, H.S.J., Campbell, J.L., Keeton, W.S., Kennedy, R.E., 2016. Do insect outbreaks reduce the severity of subsequent forest fires? Environ. Res. Lett. 11. https://doi.org/10.1088/1748-9326/11/4/045008
>
> --

L545 Cite more recent literature, e.g. the IPCC Sixth Assessment Report, Working Group I, Chapter 4.
> **Response:**
>
> we have adjusted (added) more recent literature (IPCC Sixth Assessment Report, Working Group I, Chapter 4) for this sentence (L XX):

"Circumpolar northern boreal forests and tundra are likely to continue to warm more than most other terrestrial biomes according to available data from models and observations (Chapin et al., 2005; Foley, 2005; Meehl et al., 2007; Trenberth et al., 2007; Lee et al., 2021)."

L567 For RCP8.5 it should be far more than 2 degrees C. Please provide more accurate numbers here.

**Response:**

Thank you for your suggestion, now has been adjusted as follows:

By 2100, the air temperature will increase by range 0.3-1.7 (1.0) °C under the RCP2.6 scenario, 1.1-2.6 (1.8)°C under the RCP4.5 scenario, 1.4-3.1 (2.2) °C under the RCP6.0, and 2.6-4.8 (3.7) °C under the RCP8.5 scenarios (IPCC 2021).

L571 Many DGVMs prescribe tree allometry. Is this also the case for SEIB? If yes, then that is why fires have no impact on tree allometry.

**Response:**

In SEIB-DGVM, the rules of allometry are not entirely prescribed.

For herbaceous PFTs, both below-ground and storage biomass are preserved after a wildfire and used for the recovery of above-ground biomass. During this recovery period, herbaceous PFTs work on producing above-ground biomass while reducing their storage biomass, thus increasing the allocation ratio to above-ground biomass in the post-fire phase.

For woody PFTs, fire only gives the option for individual trees to either die or survive. The surviving trees only lose their foliage biomass. As the foliage is lost, fine root biomass becomes unnecessary, leading to its rapid loss due to its fast turnover rate. In the spring following a fire, surviving trees convert storage resources into foliage and fine root biomass. The new net primary production (NPP) from the newly formed foliage first prioritizes the recovery of leaves and fine roots. Therefore, fires increase the allocation ratio to the foliage and fine roots in surviving woody plants.

L580 A decrease in fire emissions in subsequent years could also be caused by the fact that there is simply less biomass available for burning, since some of the biomass has been burnt during previous years.

**Response:**

Thank you very much for the suggestions, it has been adjusted (L664).

A decrease in fire emissions in subsequent years could be caused by the fact that there is less biomass available for burning since some of the biomass has been burnt during previous

years. Also, it could indicate the occurrence of postfire vegetation succession in these areas (Figure 11).

L586 It is not clear to me what point you are trying to make. Do you simply want to say that the model is able to account for the effect of fire history on future fires? This is to be expected, and I don't think it deserves a discussion. But I may be missing the point, in which case I suggest you rewrite this paragraph.

**Response:**

We apologize for the unclear writing structure and thank you very much for the detailed review.

Previously, in this sentence we would like to mention that the improved model (SEIB-DGVM SPITFIRE), has proven to be able to integrate well the new ignition factors (lightning strikes, and population density), which affect the distribution pattern of variable fire and burned biomass emissions. Furthermore, the improved model is also able to simulate well the dynamics of vegetation with variable fire under RCPs scenarios.

Due to the change of use of Figure 11 (classification of low emission and high emission areas), we will not use it in the next version of the manuscript (related to comment L454). We will rewrite this paragraph.

Thank you very much for your suggestions.

L596 You cannot draw the conclusion that a model is accurate, simply because its response is consistent with the change in the perturbation. I would just write that the trend in fire emissions is consistent with the different scenario-dependent trends in radiative forcings.

**Response:**

Apologize for the previous incorrect sentence and thank you very much for the given suggestion. Now the sentence has been adjusted (L691):

Therefore, its indicated that the trend in fire emissions is consistent with the different scenario-dependent trends in radiative forcings.

L599 The projected increase depends on your emission sceanrio, and currently ranges between 1.5 and 6.0 degrees C by 2100 compared to 1850-1900 mean value (IPCC, AR6, WGI, Chapter 4)

**Response:**

Thank you very much for you suggestions, we have adjusted the mentioned sentence (L693).

Overall, based on the RCPs climate scenario data used (MirocAR5), the emission scenario projected an increase in global mean surface temperature in the range of 1.0-3.7 (0.3~4.8) ºC (IPCC, 2014), and currently ranges between 1.5 and 6.0 ºC by 2100 compared to 1850-1900 mean value (Lee et al., 2021)

**Tables**

Table 2: Replace RCP80 with RCP8.5 and use a consistent notation in table in text (e.g. RCP8.5rather than RCP85)

**Response:**

Thank you very much for the suggestions, all of the related notation has been adjusted accordingly.

Table 4: Replace "Baseline" with "Historical". Note the meaning of +/- (1 or 2 standard deviations?)

**Response:**

Thank you very much for the suggestions, all of the related notation has been adjusted accordingly. It was 2 standard deviations.

"Table 4. Twenty-year average (± 2 standard deviation) of projected emissions of $CO_2$, CO, $PM_{2.5}$, TPM, and TPC species from forest fires in Siberia (2023-2100). The emissions of the remaining 28 species are listed in Table S4 in the Supplement."

**Figures**

Figure 3 Move the text in the Figure caption into the main text as you are describing the figure.

**Response:**

Thank you for the suggestions, now has been adjusted.

"SEIB-DGVM code modification are described in the section 2.2., and annual average ignition factor variables (population density and lightning flash rate) are used constantly throughout all phases to compare the improved fire regime module product with the default, previous fire module, and we run both models with the same protocols (Figure S3 in the Supplement)[*]. Simulations were run in three phases (spin-up, historical and future) and the simulation was run with the fire mode on and fire mode off to compare and assess the vegetation products during fire, and also each phase was replicated 5 times to minimize bias due to random variables in the tree morality[1]. Verification stage is to ensure the new input data can be read, produced, and processed properly[3]. After verifying that the new module

was incorporated seamlessly, we validated the model outputs (fire, vegetation and emissions variables) by using GFED4, GFED4s, ESA Biomass CCI and GBEI benchmark datasets[4]. To determine the impact of fire and climate on forest structure and their interactions, we ran the simulations with 5 climate scenarios (baseline, RCP8.5, RCP6.0, RCP4.5, and RCP2.6) using the same configuration as the fire-on simulation (Figure 3)."

Figure 4 and 5 All panels look very similar. If you want to work out the differences, then I suggest you provide a different type of plot.

**Response:**

We combined the distribution pattern map image and the graph (figure S9 and S10) into the main text. We still display one distribution pattern map to be able to see the spatial distribution pattern of the data. The adjusted figure 4 and 5 as follows:

[Figure]

**Figure 4.** Distribution pattern map and graph of annual average burned biomass (2006-2100). Default SEIB-DGVM (a and c), SEIB-DGVM SPITFIRE (b and d).

[Figure]

**Figure 5.** Distribution pattern map and graph of annual average aboveground biomass (2006-2100). Default SEIB-DGVM (a and c), SEIB-DGVM SPITFIRE (b and d).

Figure 6 Replace "Principle component" with "Principal component". Why do you need to do a PCA here? This would deserve description in the main text, if it is really worthwhile. As I mention above, please verify whether SEIB prescribes allometric relationships, which would explain why the impact of fires is small on allometry. I am not sure whether this analysis is worth including.

**Response:**

This PCA was not discussed in the main manuscript because in that figure, PCA was used only for color grading. In the next manuscript, we did not use color grading again for the plots.

Regarding the tree allometry in SEIB-DGVM, we have explained in the answer to comment L571.

*"In SEIB-DGVM, the rules of allometry are not entirely prescribed.*

*For herbaceous PFTs, both below-ground and storage biomass are preserved after a wildfire and used for the recovery of above-ground biomass. During this recovery period, herbaceous PFTs work on producing above-ground biomass while reducing their storage biomass, thus increasing the allocation ratio to above-ground biomass in the post-fire phase.*

*For woody PFTs, fire only gives the option for individual trees to either die or survive. The surviving trees only lose their foliage biomass. As the foliage is lost, fine root biomass becomes unnecessary, leading to its rapid loss due to its fast turnover rate. In the spring following a fire, surviving trees convert storage resources into foliage and fine root biomass. The new net primary production (NPP) from the newly formed foliage first prioritizes the recovery of leaves and fine roots. Therefore, fires increase the allocation ratio to the foliage and fine roots in surviving woody plants."*

Figure S16 Does this figure show that the model tends to overestimate AGB between 25 and 75 Mg ha−1 ? I would also add a diagonal 1:1 line so the reader sees where the dots should ideally be located. What do the colors of the dots represent? Please replace the horizontal axis label Ha with ha.

**Response:**

We have updated the plot for comparison of AGB data as follows:

[Figure]

Based on the previous plot and this plot (latitude averaged) shows that the model tends to overestimate around 21 Mg ha⁻¹. However, based on the grid-wide annual average, the model and benchmark data show similar values **(L330)**: In 2010, 2017, and 2018, the simulations predicted 63.714 ± 64.89 Mg DM ha⁻¹ year⁻¹, 64.141 ± 65.54 Mg DM ha⁻¹ year⁻¹, and 64.313 ± 65.61 Mg DM ha⁻¹ year⁻¹, respectively, while the ESA Biomass CCI data showed 64.027 ± 56.95 Mg DM ha⁻¹ year⁻¹, 64.548 ± 54.69 Mg DM ha⁻¹ year⁻¹, and 65.05 ± 55.78 Mg DM ha⁻¹ year⁻¹, respectively, for the same years.

Figure 8 Why does SEIB-DGVM has uncertainty bars but GFED4s does not? Also, explain the meaining of the uncertainty bar in the Figure caption. Is this +/- SD across multiple years?

**Response:**

Apologies for the previous plot we didn't include the uncertainty for the GFED4s data. Now we have added as your suggestions for both plots.
Yes, the SD across multiple years (1996-2016).

[Figure]

**Figure 8.** Monthly Average Dry Matter Emission data comparison of GFED4s and SEIB-DGVM SPITFIRE (1997-2016): a) Monthly average seasonality b) Monthly average comparison. The standard deviation obtained from each monthly data from 1997-2016 for each data.

Figure 12 Increase axis value fontsize

**Response:**

Has been adjusted as follow:

---

## Author Comment (AC3)

**Reviewer #3**:

This study projected future wildfire activities in Siberia by implementing a process-based fire module into a dynamic vegetation model. The main conclusion was that fire emissions continued to increase due to climate change. The authors also quantified the negative impacts of fire activity on tree mortality and NPP. While the topic of the study well fit the scope of BG, I found the presentation was very poor and many results were contradicting. I suggested a major revision but with strong doubts about the credibility of the model and the projections.

**Response:**

Thank you for taking the time to review our manuscript. We appreciate your acknowledgment that the topic of our study aligns well with the scope of Biogeosciences (BG). Your feedback regarding the presentation and clarity of our results is noted, and we apologize for any confusion or inconsistencies encountered.

We understand the importance of ensuring the credibility of our model and projections. Your comments have raised valid concerns in this regard, and we are committed to addressing them through a major revision of the manuscript. We will carefully reevaluate our methodology, results, and interpretations to ensure accuracy and coherence throughout the manuscript. Moreover, we will provide additional clarification and justification for our findings to mitigate any perceived contradictions. We recognize the significance of transparently communicating our research outcomes to facilitate a better understanding among readers.

Your feedback is invaluable to us, and we are grateful for the opportunity to improve our manuscript with your guidance. We assure you of our willingness to make necessary adjustments to enhance the credibility and quality of our work.

Thank you once again for your thorough review and constructive comments.

1. First, the model descriptions were unclear. The authors put most of details, including equations and descriptions, to the supplementary material. However, the key processes should be listed in the main text for clarity. Most important, the links between the fire module and vegetation model should be explicitly explained. For example, what parameters did the SEIB-DGVM provide for SPITFIRE, and how the fire activity affect the vegetation distribution and ecosystem productivity. Such information determines which variables to be validated against observations.

   **Response:**

   We briefly describe the details of the model in section "2.2. Improved model principles" that include basic fire disturbance equation and emission estimation. We wrote the

details of other variable calculations in the supplementary file because our model development only combines two existing models (SEIB-DGVM with SPITFIRE module), which have been described in detail by their respective authors (Sato et al., 2007; Thonicke et al., 2010).

The explanations we wrote in the equations are adjustments made in the integration process, and additional development processes we carried out, such as increasing the annual timestep to monthly timestep, and adding calculations of 33 burned biomass emissions.

a) We understand that we have not explained about fire-vegetation relationships in this section, and we appreciate for your suggestions. We have added the information about fire-vegetation relationships in this section (2.2), as follows:

"The fraction of individual trees killed by a fire depends on PFT fire resistance (M3, Table 1). All grass leaf biomass, all dead and living tree leaf biomass, half of the dead tree trunk biomass, and half of the litter pool are released into the atmosphere as $CO_2$ during a fire, while the dead tree's residual biomass is converted into litter. In reaction to fire, all deciduous PFTs convert their phenology phase to dormancy, and if the stock resource of grass PFTs (gmassstock) does not meet the minimal value (50 g DM m$^{-2}$) following fire, the deficit is supplemented from litter (Sato et al., 2007). Furthermore, related to the fire-vegetation relationships, for herbaceous PFTs, both below-ground and storage biomass are preserved after a wildfire and used for the recovery of above-ground biomass. During this recovery period, herbaceous PFTs work on producing above-ground biomass while reducing their storage biomass, thus increasing the allocation ratio to above-ground biomass in the post-fire phase. For woody PFTs, fire only gives the option for individual trees to either die or survive. The surviving trees only lose their foliage biomass. As the foliage is lost, fine root biomass becomes unnecessary, leading to its rapid loss due to its fast turnover rate. In the spring following a fire, surviving trees convert storage resources into foliage and fine root biomass. The new net primary production (NPP) from the newly formed foliage first prioritizes the recovery of leaves and fine roots. Therefore, fires increase the allocation ratio to the foliage and fine roots in surviving woody plants."

b) Regarding the integration of parameters and variables in SEIB-DGVM for the SPITFIRE module, we determined three main variables that must be validated, namely fire variables (burned fraction, burned area, dry matter emission: GFED4s and GFED4), Aboveground biomass (ESA Biomass CCI), and $CO_2$ emissions (GFED4s, and GBEI). The determination is based on the relationship between fire, vegetation and emissions resulting from the interaction of these two variables.

We considered to replace the current figure 2 with the complete improvement scheme, to give more information about the improvement and the integration between SEIB-DGVM and SPITFIRE module. The new Figure 2, as follow:

[Figure]

**Population**

**Lightning**

**Location:**
Latitude
Longitude

**Climate:**
T max,
T min
T soil, P,
Rh, cloud,
wind

**Atmospheric CO$_2$**

**Soil Properties:**
W sat, W fi,
W mat, W wilt,
Albedo, Depth

| Frac$_{moisture\ litter}$ | Fuel$_{min}$ |
| Min$_{gmass\ stock}$ | m$_{extinc}$ |
| Frac$_{trunk\ ag}$ | M$_3$ |
| Frac$_{trunk\ lost\ at\ fire}$ | |
| Fuel$_{load}$ | Mass$_{overall}$ |
| Fuel$_{moist}$ | Moist$_{factor}$ |
| Fire$_{factor}$ | Fire$_{prob}$ |
| Mass$_{combust}$ | Fire$_{number}$ |
| Mass$_{stock}$ | |

**Vegetation Dynamics**
Physics, Physiology, and Ecological
Dynamics

**SEIB-DGVM**
Default Fire Module

| $Ip$ | Nil | NID | $\rho_p$ | $\rho_b$ | $\xi$ | $SH$ |
| Nih | | Firein | CL | RCK | $\varepsilon$ | $CK$ |
| FDI | | $CE$ | par1 | par2 | Q$_{ig}$ | $BT$ |
| $P_{sp}$ | Enig | C | $Pcd$ | F | $\Phi_W$ | T$_C$ |
| Ab$_{fract}$ | $NI$ | $\Gamma_{max}$ | $\alpha_V$ | s$_T$ | ROS$_f$ | P$_m(\tau)$ |
| Ab | $\omega_o$ | $\Gamma$ | $h$ | | ROS$_b$ | P$_m$(CK) |
| Mass$_{comb\ (C)}$ | | W$_n$ | S$_E$ | | L$_{b\ (tree,\ grass)}$ | P$_m$ |
| Mass$_{comb\ (DM)}$ | | $\eta_m$ | $B$ | | $\bar{a}_f$ | EF$_{1-33}$ |
| $\sigma$ | $\beta$ | $\eta_S$ | $C$ | | t$_{fire}$ | $E_x$ |
| A | $\beta_{op}$ | $I_R$ | $E$ | | I$_S$ | |

**SEIB-DGVM SPITFIRE**
Improved Fire Module

input          adjustment          feedback

Model input   PFT parameter   Local parameter   Global parameter   Variable   Model output

**2.** Second, the model validations were questionable. Normally, the validations come before the projections. However, this study put the validations of fire models in the section 3.7, far behind the future projections. Some of the validations were not necessary. For example, Figure S2 compared the input and output of lightning flash rate and population density, which were actually the input of fire models instead of the prediction. Some validations were not consistent between different presentations. For example, Figure S17a compared the simulated and observed dry matter. The model predictions showed poor performance by missing almost all the observed fire episodes. However, in Figure 8, the monthly validations of simulated dry matter showed perfect performance with almost the same magnitude (Figure 8a) and R2=1 (Figure 8b) against observations. These results were too good to be true and seemed not consistent with Figure S17. Furthermore, the perfect simulation of dry matter (Figure 8b) resulted in a poor prediction of CO2 emissions (Figure 9). Was such bias attributed only to the emission factors? Then why making such a great effort to develop the sophisticated fire module when the simplest parameters caused the largest biases? BTW, I do not believe the R2 >0.6 in Figure 9 given such a wide range of scattering points.

**Response:**

Thank you very much for your detailed comments. We answer this comment in the following points:

a) Apologize for the incorrect writing structure, in the previous version of the manuscript, subsections 3.1-3.4 discussed the performance of the default model with the improved model. We will move section 3.7 Improved model validation to the first section (3.1) in the adjusted manuscript.

b) Our opinion, Figure S2 is quite important for us, because in our model development process (integration of SEIB-DGVM with SPITFIRE module), the ignition factor is the first variable that we integrate and differentiate with the previous fire module (Globfirm). Figure S2 shows one of the verification processes, that the model is able to read the input (after modification) and is able to process and produce input with exactly the same data as the given input (population density and lightning flash rate variables).
However, it's just one of the improvement processes, that's why we decided to just put those images in the supplementary.

c) Figure S17 is the Monthly Dry Matter Emission data comparison between GFED4s and SEIB-DGVM SPITFIRE, describing the dynamics of GFED4s data and the simulation results produced by SEIB-DGVM SPITFIRE. We explain this in L397, and the paragraphs that follow. This has been briefly explained at L412: the model is not yet able to reproduce the exact value at a specific time of year or month because the model is run in a longterm phase and is not yet able to predict sudden natural and anthropogenic conditions/ factors (model limitation).
Meanwhile Figure 8 is Monthly Average Dry Matter Emission seasonality data comparison of GFED4s and SEIB-DGVM SPITFIRE (1997-2016). The results of this comparison correctly show the value of R2 = 0.99 (L415). This is an advantage of the model, where the model is able to learn data patterns over a long period of time with a high level of accuracy, because there is a calibration process carried out between the

model variables and the variables from the benchmark data used, as well as the integration between these variables in the model (Figure S11).

d) We apologize for not making it clearer in sub-section 2.4. related to the validation of the model output with some benchmark data that we did two types of comparison: numerical comparison and spatial comparison, and we did not explain the comparison in the manuscript (only in the figure label).

Figure 8b shows the Monthly Dry Matter Emission data comparison between GFED4s and SEIB-DGVM SPITFIRE (1997-2016) in plot form. This comparison is done by comparing the monthly extracted data of each dataset in a period of 20 years from 1997-2016.

Figure 9 is the $CO_2$ emissions spatial average comparison of SEIB-DGVM SPITFIRE module product with GFED4s and GBEI.

To clarify the information, we have corrected **L432** (This section only describes the internal model calculation process between dry matter emissions and gaseous emissions produced by SEIB-DGVM SPITFIRE.) and the sentence after it as follows:

"When comparing the annual average dry matter emission data and $CO_2$ emissions generated by the model, the results correlated perfectly (100%, Figure S27), indicating that the model runs well according to Equation (3) and the projected $CO_2$ and other emissions have the same distribution patterns as the dry matter variable, because all of the emissions calculation are based on the dry emission variable. However, they differ in their values because each emission species has a different emission factor."

And we adjusted the sentence in **L436** to clarify the validation process, as follows:

"Spatially, the annual average $CO_2$ emission model data were 63% (Figure 9.a) and 67% (Figure 9.b) correlated with the GFED4s and GBEI data, respectively."

The difference in spatial distribution of model-generated variables with GFED4s data have been explained in **L508.** The comparison results are different due to differences in the data being compared, numerically extracted data and spatial data. We will write about this in a separate subsection "Model uncertainty", which explains that the model is able to simulate very well numerically with all benchmarks data because we have a calibration process for each variable with benchmark data over a long period of time. The limitations of the current model are that it is not able to simulate sudden fluctuations due to natural factors that have not been considered in the simulation process, and the distribution pattern of emission data depends on the distribution pattern of the simulated fire variables. Thus, the fire ignition calculation and its process are very important that will affect the following variables: emissions.

Related to the R2 score of the Figure 9, we have shared the dataset in the Google Spreadsheet, please access through the following URL:

https://docs.google.com/spreadsheets/d/1RkgB8qGQaLAKI4EkeUHZTS0S8L3bq4y8/edit?usp=sharing&ouid=114559960745563618149&rtpof=true&sd=true

**3.** Third, the future projections were doubtful. This study used daily meteorology from four RCP scenarios output by MirocAR5. How to reduce the uncertainties from a single climate model projection? Normally, these scenarios showed very different tendencies of warming, indicating different fire probability. However, the projections of fire activities showed very similar results among these scenarios (e.g., Figure 4, Figure S5, Figure 12 …). Does it mean that the future wildfire activity in Siberia will increase at the similar rate no matter how warm the climate becomes? Furthermore, the updated fire module showed good correlations between burned fraction and burned biomass (Figure S11d), suggesting tight connections between these two parameters. The burned fraction is projected to increase continuously after the year 2040 (Figure S5b). Then why burned biomass showed such a large fluctuation over the same period?

**Response:**

Thank you very much for your detailed comments. We answer this comment in the following points:

a) We use the MirocAR5 dataset as is, regardless of its uncertainty. The model data bias has been corrected using "EWEMBI" observational dataset. (https://www.isimip.org/gettingstarted/input-data-bias-adjustment/details/21/)

b) The fire projection results that do not have much difference between under all different RCP scenarios (e.g., Figure 4, Figure S5, Figure 12...), we suspect based on the calculation process flow, because the fire calculation does not directly use/consider the temperature variable. Instead, the fire variable is estimated based on the calculation of a chain of variables ranging from fuel availability fuel load (litter + aboveground biomass), moisture content, ignition factors (lightning and population), parameters accompanying the calculation of these variables (such as the minimum fuel load threshold of 200 g C m$^{-2}$: Sato et al., 2007), and also this is affected by the type of input dataset used and the limitations of the model.

However, the estimation results produced by the model have been validated with several benchmark data which include the fire itself, vegetation and emissions variables (Table 3).

c) Yes, the correlation of burned fraction and burned biomass variables in the improved model (Figure S11.d) shows the correct relationship, where the increase of burned fraction, the burned biomass also increases.

The burned biomass variable has an increasing trend as well (same as burned fraction) from year to year as shown in the following figure:

[Figure]

Related to the fluctuation is normal, because there is a process of vegetation dynamics (tree death due to fire, and vegetation succession process after forest fires, and phenological factors simulated in this SEIB-DGVM). Thus, there is no contradiction from these variables.

**4.** Finally, the quality of result presentations was low. For example, the figures were very similar among the subplots of Figure 4e-4h, Figure, Figure S4e-S4h, Figure S6. Such information is useless as the readers could not tell their differences. Figure S28 showed the projected fire emissions of 33 trace gases by putting all the subplots together without any summary. It's difficult to tell their differences and the main conclusions. The authors also spent great efforts in comparing results from the default and updated fire modules (e.g., Figures S4, S5, S9, S10, S11). While it's important to understand how different the fire predictions before and after the improvement of fire modules, the authors could show some key results (e.g., Figure S11) and put more efforts in the validations of the updated model against observations, not only for fire activities but also some ecosystem parameters (e.g., tree height, biomass, NPP).

**Response:**

Thank you very much for your detailed comments. We answer this comment in the following points:

a) We have combined the images of each visualized variable. We present one map to show the distribution pattern (representing all estimates from other RCP scenarios as they are spatially similar), as follows:

[Figure]

**Figure S4.** Distribution pattern map and graph of annual average burned fraction (2006-2100). Default SEIB-DGVM (a and c), SEIB-DGVM SPITFIRE (b and d).

[Figure]

**Figure 4.** Distribution pattern map and graph of annual average burned biomass (2006-2100). Default SEIB-DGVM (a and c), SEIB-DGVM SPITFIRE (b and d).

[Figure]

**Figure 5.** Distribution pattern map and graph of annual average aboveground biomass (2006-2100). Default SEIB-DGVM (a and c), SEIB-DGVM SPITFIRE (b and d).

b) Thank you very much for the suggestion, we have added these few sentences in the **L590** to explain Figure S28.
"Spatially, the projections depict heterogeneous patterns of burned biomass emissions, with regions of high emissions intensity concentrated in areas of larch forest (Larix spp.), consistent with Figure 1 and our simulation results, where the fire and emission variables show high values in central to southern Siberia (Figure S4.b, Figure 4.b, and Figure S6.b). This is reinforced by field-based estimation data, that fires in this region result in high tree mortality 76%, Siberian larch forests experience greater aboveground carbon loss after fire than do North American forests, both in absolute and relative levels (Webb et al., 2024). We also visualized all the 33 graphs depicting projected burned biomass emissions, offering valuable insights into the future dynamics of the burned biomass emissions in Siberia. Across these graphs, we observe distinct temporal patterns, revealing trends in burned biomass emissions over time. Under the RCP8.5 to RCP2.6 scenarios, overall emissions by 2100 are projected to increase by 2.6 %, 1.9 %, 1.05 % and 1.04 % compared to 2000 emissions (Figure S28), and the twenty-year dynamics are summarized in Table 4 and Table S4."

c) We present Figures S4, S5, S9, S10, S11 because they are needed to show the differences in simulated fire variables, vegetation, and the relationship between the improved model and the default model. This is not our main purpose, but only to strengthen the explanation given, so we only include it in the Supplement. Regarding the explanation of the figure, it has been explained in detail in the main text, respectively.

Regarding the validation of the model with benchmark data, we have explained in section 2.4, and the validation results are in section 3.7. Improved model validation.

We agree with your suggestion that we should not only discuss fire activity, but also ecosystem parameters. In the manuscript, we have explained about the NPP projection and comparison with some other observational data **(L548).** We have also added the NBP variable (in section 4.2) to the revised manuscript in accordance with other reviewers' suggestions to strengthen the discussion related to the strength of carbon flux caused by disturbance: fire.

**Figures**

Figure 10: What's the meaning of the shadings in (a)? What's the meaning of the points in (b)?

**Response:**

a) The shading in Figure 10.a is the ± standard deviation of $PM_{2.5}$ emissions projected by each RCP scenario.

b)  The points in Figure 10.b are comparisons between the model's PM$_{2.5}$ emission projections with Copernicus Atmosphere Monitoring Service (CAMS) (Romanov et al., 2022) data in seven Russian territories during 2004-2021.

Figure S5: What's the reason for the sharp decline of burned fraction around 2035 in (b)?

**Response:**

We suggest this is due to the high fire fraction in 2038 (which has a similar value in the 2060 projection), resulting in high carbon emissions in that year and beyond. This is evidenced by the low NBP values in 2021-2040 compared to the average of the other years in Figure rev1, and on a point scale, the NBP variables projected by all RCP scenarios have negative values, indicating a carbon source (Figure S35).

Discussion related to NBP (rev1 and rev2 drawings) has been added in sub section 4.2. adjusted according to other co-author's suggestions as well.

[Figure]

Figure S34.

[Figure]

Figure S35.

---

## Referee Report (RR1)

MS Title: Future projections of Siberian wildfire and aerosol emissions by process-based ecosystem model

Authors: Reza Kusuma Nurrohman et al

Review of first revision

**General Comments**

The author's revisions have improved the overall quality of the manuscript. In particular, I appreciate that the authors first evaluate model performance before assessing the future projections. The incorporation of a fire module into a DGVM is an important and non-trivial task and I would like to congratulate the authors on having achieved this. However, the manuscript still reads more like a report than a scientific paper. This is because the authors tend to document all of their findings, rather than focusing on the more interesting ones. While the approach is thorough, it makes the manuscript hard to read. The authors have addressed most of my previous comments to my satisfaction. However, I still have a number of comments below for the revised version of the manuscript. I hope you find my comments useful.

**Detailed Comments**

L1 The title "Future projections of Siberian wildfire and aerosol emissions by process-based ecosystem model" is grammatically incorrect. It would be "produced by" or "made by", or "conducted with a". Personally, I would reduce the title to "Future Projections of Siberian Wildfire and Aerosol Emissions".

L5 Replace "Fires" with "Wildfires"

L18 You don't want your abstract to be wordy. Delete non-essential info, such as "spatially explicit individual-based"

L21 Revise grammar

L21 Avoid acronyms in abstract (other than $R^2$ and RMSE). If you need them, you need to spell them out first, e.g. AGB. In this context I would omit the names of the reference data sets, such as GFED4s.

L24 What "data"? Also, does "numerically" refer to spatial mean values? Rewrite to omit ambiguity.

L26 Replace "climate scenario" with "climate change scenario"

L27 Rather than the increase rate, can you please provide the relative change between 20-year mean historical period against 20-year future period? A relative change between two periods is more intuitive than an increase rate.

L29 What fraction of trees is this? How does this fraction compare to the historical run? I am asking these questions because from the absolute numbers it is not clear how large the impact is. It is easier to grasp when you write for instance, that the amount of trees burnt increases by a factor of three.

L43 What does "human activity" refer to in this context?

L49 What is "global mean $CO_2$ emission intensity" referring to? Are you referring to the annual rate of global total $CO_2$ emissions? Does "NS" refer to Northern Hemispheric Summer?

L51 Why "as well"?

L61 You write that "Prolonged exposure to high $CO_2$ concentrations has negative impacts on health and agriculture". Concerning health, that is indeed true but for very high concentrations of $CO_2$, not atmospheric concentrations. For agriculture, high levels of CO2 can be beneficial. I think the relevant negative health impacts are more related to the emission of particular matter, rather than $CO_2$.

L63 This feedback is important, please elaborate.

L68 Revise grammar

L75 Explain what you mean by a negative impact.

L171 Replace the underline with a bar in Equation (2) to make the notation consistent with the one used in the text.

L172 Grid cells is plural

L182 Start line with "where" and please avoid using 3-nested parenthesis.

L187 LCT is not yet defined

L188 The PFT names are not yet defined, and please don't start a sentence with an acronym

L195 I would delete these two sentences on the verification process. I think it is self-understood that you make sure everything is working properly.

L232 Delete superscript ("inputs$^2$")

L233 Why is the "MirocAR5 Base V2 dataset is generated from CRU TS3.22 climate data"? Miroc is a climate model. Do you mean that the Miroc version you used as bias-corrected with CRU? Please clarify.

L241 The term "saturation" does not fit here.

L243 Write "The SEIB-DGVM code modifications" and revise grammar. I suggest you rewrite the sentence as two sentences.

L249 Revise grammar

L318 Remind the reader what this core variable is

L375 Revise grammar

L498 Write "because it is"

L499 You write that "Based on the comparison of results between the fire-on and fire-off simulations, the NPP variable under all of the RCP scenarios shows a downward trend with some small fluctuations". It sounds as if the difference between simulations with fires on and off reveals a trend in NPP. I am sure this is not what you mean to say, please rewrite.

L565 FDI not defined

L642 Replace "dan" with and"

L649 Revise grammar

L655 Please avoid sentences like these. Instead of copying the raw data from the table, facilitate its interpretation.

L654 Replace "Heterotrophic" with "heterotrophic"

L660 Does this apply to all of the RCPs that you assessed?

L661 Why "saturated"?

**Figures**

Figure 5  Replace "Kg" with "kg" in figure legend (here and elsewhere). Also, increase font of x-axis, and consider replacing the full month names with their first letters (J, F, M, etc.)

Figure 8  The burned biomass should have a time unit as well (mass per unit of area per unit of time). Add GlobFIRM and SPIFIRE to the (c) and (d) panels, respectively, such that the difference between both plots is more obvious.

---

## Author Response (AR2)

MS Title: Future projections of Siberian wildfire and aerosol emissions

Authors: Reza Kusuma Nurrohman et al

Author's response to the Anonymous referee #2

**General Comments**

The author's revisions have improved the overall quality of the manuscript. In particular, I appreciate that the authors first evaluate model performance before assessing the future projections. The incorporation of a fire module into a DGVM is an important and non-trivial task and I would like to congratulate the authors on having achieved this. However, the manuscript still reads more like a report than a scientific paper. This is because the authors tend to document all of their findings, rather than focusing on the more interesting ones. While the approach is thorough, it makes the manuscript hard to read. The authors have addressed most of my previous comments to my satisfaction. However, I still have a number of comments below for the revised version of the manuscript. I hope you find my comments useful.

> **Response:**
>
> Thank you for your thoughtful and encouraging feedback on our revised manuscript. We are pleased to hear that you found our evaluation of the model performance before assessing future projections to be an improvement. We appreciate your recognition of the complexities involved in incorporating a fire module into a DGVM, and we are grateful for your congratulations.
>
> We understand your concern that the manuscript still reads more like a report than a scientific paper due to the comprehensive documentation of our findings. We acknowledge that focusing on the most interesting and relevant results would improve readability and overall presentation. We have discussed this matter internally with other co-authors, we wrote down most of all our findings because we think it is important to prove that what we mentioned in the main manuscript is scientifically validated, which covers from the initial to the end of research stage. However, we have separated the main findings in the main manuscript, while the supporting data and information are separated in the Supplement so that readers can easily find well-documented supporting information online in the Supplement file. in this second revision stage, we slightly reduced the general information (reducing the introduction section and moving some information in the methodology section to the supplement).
>
> We take your comments into careful consideration and make the necessary adjustments to streamline the manuscript. Specifically, we will highlight the key findings and their implications while reducing the emphasis on less critical details to enhance the manuscript's clarity and engagement.
>
> Thank you again for your valuable feedback and for acknowledging the improvements we have made. We are very grateful for the reviews you gave from the first and second stages, it is very constructive and improves the quality of our manuscript. We are committed to further

revising our manuscript in response to your comments and hope that the next version will meet your expectations.
* * *
Update: July 17, 2024
* * *
I wanted to inform you that we identified an incorrect parameter setting in our future simulation, when attempting input data flow tracing last month. Consequently, we have conducted a comprehensive re-simulation following the protocol outlined in the manuscript.

All your suggestions have been implemented accordingly. However, it is possible that there may be some adjustments in the manuscript due to varying trend results observed post re-simulation.

We sincerely appreciate your continued support and guidance throughout the review process. Your feedback has been invaluable to improving our manuscript.

Thank you once again for your assistance.

**Detailed Comments**

**L1** The title "Future projections of Siberian wildfire and aerosol emissions by process-based ecosystem model" is grammatically incorrect. It would be "produced by" or "made by", or "conducted with a". Personally, I would reduce the title to "Future Projections of Siberian Wildfire and Aerosol Emissions".

**Response:**

I apologize for the incorrect grammar in the title. We agree that your suggested title, "Future Projections of Siberian Wildfire and Aerosol Emissions," is clearer and more effective. We will change the title of the manuscript according to your suggestion. Thank you for your detailed review and constructive suggestions.

**L5** Replace "Fires" with "Wildfires"

**Response:**

Has been adjusted: **L14** Wildfires are among…

**L18** You don't want your abstract to be wordy. Delete non-essential info, such as "spatially explicit individual-based"

**Response:**

Has been adjusted: **L18 …** fire module into the dynamic global vegetation model (SEIB-DGVM)…

**L21** Revise grammar

**Response:**

The grammar has been revised: **L21**

The model is able to reproduce historical data well compared to benchmark datasets. Based on the spatial validation, the results are as follows: Aboveground biomass ($R^2$=0.43, RMSE=21.9 Mg ha$^{-1}$), burned fraction ($R^2$=0.75, RMSE=0.01), burned area ($R^2$=0.609, RMSE=690 ha), dry matter emission ($R^2$=0.63, RMSE=0.01 Kg DM m$^{-2}$), $CO_2$ emissions ($R^2$=0.64, RMSE=60.9Tg).

**L21** Avoid acronyms in abstract (other than $R^2$ and RMSE). If you need them, you need to spell them out first, e.g. AGB. In this context I would omit the names of the reference data sets, such as GFED4s.

**Response:**

Has been adjusted: **L21**

Based on the spatial validation, the results are as follows: Aboveground biomass ($R^2$=0.43, RMSE=21.9 Mg ha$^{-1}$), burned fraction ($R^2$=0.75, RMSE=0.01), burned area ($R^2$=0.609, RMSE=690 ha), dry matter emission ($R^2$=0.63, RMSE=0.01 Kg DM m$^{-2}$), $CO_2$ emissions ($R^2$=0.64, RMSE=60.9Tg).

$R^2$ and RMSE value of $CO_2$ emissions validation written in abstract is the average validation value from GFED4s and GBEI.

**L24** What "data"? Also, does "numerically" refer to spatial mean values? Rewrite to omit ambiguity.

**Response:**

Data= model output
Numerically refer to the numerical values derived (extracted) from spatial data. Yes, mostly we compare with the benchmark dataset as mean values.

Thank you very much for the detailed review and suggestion. Now, has been adjusted: **L24**

Overall, the model is able to produce output with spatial distribution patterns similar to the benchmark dataset, with an average similarity of 61.8%. Furthermore, based on the comparison of mean values with the benchmark datasets, the model produces high accuracy, amounting to 99%.

**L26** Replace "climate scenario" with "climate change scenario"

**Response:**

Has been adjusted

**L27** Rather than the increase rate, can you please provide the relative change between 20-year mean historical period against 20-year future period? A relative change between two periods is more intuitive than an increase rate.

**Response:**

Thank you for the suggestion, has been adjusted as follow

We estimated that the $CO_2$, CO, $PM_{2.5}$, total particulate matter (TPM), and total particulate carbon (TPC) emissions in Siberia in 20-year mean historical period (2000-2020) will increase relatively by $43.68 \pm 1.1$, $3.5 \pm 0.09$, $0.56 \pm 0.015$, $0.43 \pm 0.011$, $0.3 \pm 0.008$ Tg species year$^{-1}$, respectively in the 20-year future period (2081-2100) under the Representative Concentration Pathways 8.5.

**L29** What fraction of trees is this? How does this fraction compare to the historical run? I am asking these questions because from the absolute numbers it is not clear how large the impact is. It is easier to grasp when you write for instance, that the amount of trees burnt increases by a factor of three.

**Response:**

This is the "number of tree/ tree density" variable that produced by our model (unit: tree ha$^{-1}$). Tree mortality in SEIB-DGVM consist of background mortality (growth efficiency), heat stress, bioclimatic limit, and due to the wildfire (Sato et al, 2007).

We obtained the mentioned value in **L29** is under fire-on and fire-off simulations to make sure the produced variable is affected (killed) by wildfire only. In addition, we ran using all climate scenarios and repeated 5 times (same protocols and ran in the same simulation-time with other produced variables). This variable is described in section 3.6. We compared the estimated value in the future simulation under RCP8.5 with the historical value (Figure S25.c).

We understand that the suggestion you give will be easier to understand. However, if we state in a factor, the number of tree burnt increases by a factor of 1.08. To make it easier for the reader, we decide to write it as percentage as follows: **L27**

"Under the same climate scenario and period comparison, we estimated that the number of trees burnt increases by 108%, resulting in a 319.3 g C m$^{-2}$ year$^{-1}$ loss of net primary production (NPP)."

**L43** What does "human activity" refer to in this context?

**Response:**

In this context, human activity refers to human activity using fire for land management.

We have adjusted the sentence to be as follows:

.. and human activity by using fire for land management (e.g. use of fire as a tool in the deforestation process) (Hantson et al., 2016; Archibald et al., 2013; Morton et al., 2008).

**L49** What is "global mean $CO_2$ emission intensity" referring to? Are you referring to the annual rate of global total $CO_2$ emissions? Does "NS" refer to Northern Hemispheric Summer?

**Response:**

Global mean $CO_2$ emission intensity refers to global $CO_2$ emissions per unit of area burned.

NS refers to non-significant.

However, in this revision stage we would like to remove this information, we just realized that we misplaced this information. We apologize for our mistake and we're grateful for the detailed review.

Below is the paragraph from the information we cited.

"On the basis of our analysis of the MOPITT CO observations and atmospheric inversions,

we estimate the global fire $CO_2$ emissions to be 1.8 Gt C year$^{-1}$, on average, during 2000–2019, with a non-significant decreasing trend of $-0.5 \pm 0.8\%$ year$^{-1}$ (95% confidence interval; purple curve in Fig. 1A). The quasi-stable emissions combined with a significant decline in global burned areas ($-1.6 \pm 0.4\%$ year$^{-1}$; orange curve in Fig. 1A) suggest that the global mean emission intensity (i.e., $CO_2$ emissions per unit of area burned) has increased by $0.9 \pm 0.9\%$ year$^{-1}$ since 2000 (purple curve in Fig. 1B) (Zheng et al., 2021)."

Zheng, B., Ciais, P., Chevallier, F., Chuvieco, E., Chen, Y. and Yang, H. (2021) 'Increasing forest fire emissions despite the decline in global burned area', Science Advances, 7(39). Available at: https://doi.org/10.1126/sciadv.abh2646.

**L51** Why "as well"?

**Response:**

Previously, we wrote "as well" because at the beginning we stated that global mean $CO_2$ emissions are increasing, then we mentioned in Europe "as well".

We re-checked the sentence's grammar and found that it was incorrect, so we removed "as well".

Thank you for the detailed review.

**L61** You write that "Prolonged exposure to high $CO_2$ concentrations has negative impacts on health and agriculture". Concerning health, that is indeed true but for very high concentrations of $CO_2$, not atmospheric concentrations. For agriculture, high levels of $CO_2$ can be beneficial. I think the relevant negative health impacts are more related to the emission of particular matter, rather than $CO_2$.

**Response:**

Thank you very much for your concern related to these points. Yes, we agree with your statements. We have adjusted the sentences accordingly:

a) We reorder the sentences to keep discussing the impact of increasing atmospheric $CO_2$ emissions at the beginning and adjusted based on the **L63** comment.

   We have adjusted the sentences as follows: **L61**

   "Increasing atmospheric $CO_2$ concentrations alter the global carbon cycle by causing global warming (Van Der Werf et al., 2006, 2010, 2017; Neto et al., 2009; Kaiser et al., 2012; Lin et al., 2013), and the resulting global warming is expected to intensify extreme fire seasons, leading to further surges in carbon emissions that significantly contribute to the global burden of greenhouse gases (fire-climate feedbacks) (Bowman et al., 2009)."

b) Agricultural impact (**L64**)

   "This event also affect the agricultural sector positively and negatively depending on the region, environment, and crop types (Kimball and Idso, 1983)."

c) We stated the impact the $CO_2$ exposure on human health at the ground level after the previous sentence as additional information. **L66**

"Additionally, prolonged exposure to very high $CO_2$ concentrations at ground level has negative impacts on health (Jacobson et al., 2019). Therefore, …"

**L63** This feedback is important, please elaborate.

**Response:**

Has been adjusted (**L61**)

**L68** Revise grammar

**Response:**

Has been adjusted (**L70**)

**L75** Explain what you mean by a negative impact.

**Response:**

Has been adjusted (**L79**)

Furthermore, an increase in atmospheric emissions negatively affects the climate by contributing to global warming and climate change (Randerson et al., 2006; Westerling et al., 2006; Bowman et al., 2009)

**L171** Replace the underline with a bar in Equation (2) to make the notation consistent with the one used in the text.

**Response:**

Thank you for the detailed review, now has been adjusted (**L178**)

**L172** Grid cells is plural

**Response:**

Has been adjusted (**L176**)

**L182** Start line with "where" and please avoid using 3-nested parenthesis.

**Response:**

Has been adjusted (**L191**)

**L187** LCT is not yet defined

**Response:**

Has been adjusted (**L196**)

**L188** The PFT names are not yet defined, and please don't start a sentence with an acronym

**Response:**

The definition of PFTs has been added in Appendix A.3 and the beginning of the sentence

has been adjusted (**L197**). Thank you for your suggestions.

**L195** I would delete these two sentences on the verification process. I think it is self-understood that you make sure everything is working properly.

**Response:**

Yes, now those two sentences have been deleted (**L204**). Thank you for the suggestion.

**L232** Delete superscript ("inputs$^2$")

**Response:**

Has been deleted after "inputs". Now, superscript 2 has been moved to "The model was run in three phases$^2$" **L242**. This is to explain the that the "simulation years$^2$" process setting in Figure 3.

**L233** Why is the "MirocAR5 Base V2 dataset is generated from CRU TS3.22 climate data"? Miroc is a climate model. Do you mean that the Miroc version you used as bias-corrected with CRU? Please clarify.

**Response:**

Thank you very much for the suggestion. Yes, I meant that, because the historical simulation used climate data from CRU TS3.22 and the future simulation from MirocAR5 Base V3 data (correction: we used version 3). So to ensure harmonized climate input data, Miroc climate data has been bias-corrected using CRU TS3.22 data.

Thank you very much for the detailed review and constructive suggestion.

**L246.** The MirocAR5 Base V3 dataset has been bias-corrected with CRU TS3.22 climate data, so using these two datasets consecutively in spin-up, historical, and future simulations ensures the harmony of the input climate data.

**L241** The term "saturation" does not fit here.

**Response:**

The "saturation" has been replaced with "equilibrium".

Adjusted sentence as follows **L256**

Another study by Arakida et al. (2021) also confirmed that a spin-up period of 100 years was sufficient for the equilibrium of the LAI, aboveground biomass, and GPP at all the study sites in Siberia.

**L243** Write "The SEIB-DGVM code modifications" and revise grammar. I suggest you rewrite the sentence as two sentences.

**Response:**

Thank you for the detailed review. Apologize for the incorrect sentence structure, we tidied up section 2.4 by slightly changing the paragraph structure to improve the flow.

We made changes to the placement of the last paragraph sentence as follows (already applied to the revised manuscript).
* * *
SEIB-DGVM code modification are described in the section 2.2. **(deleted).**

The annual average ignition factor variables (population density and lightning flash rate) were used consistently throughout all simulation phases **(L236).**

We ran the improved model (SEIB-DGVM SPITFIRE) and the default model (SEIB-DGVM GlobFIRM) under the same protocols to equally compare and assess their fire products (Figure S3 in the Supplement)[*] **(L238).**

Simulations were run in three phases (spin-up, historical and future) and the simulation was run with the fire mode on and fire mode off to compare and assess the vegetation products during fire, and also each phase was replicated 5 times to minimize bias due to random variables in the tree morality[1]**(L239).**

In addition, we have verification stage[3] to ensures that the new input data can be read, produced, and processed properly (Rabin et al., 2017). Then, we calibrate all of the major emissions individually and sequentially with the benchmark dataset because each variable affects other variables, and we need to ensure the final output is comparable with the benchmark datasets[4]. After verifying that the new module was incorporated seamlessly, we validated the model outputs (fire, vegetation and emissions variables) by using GFED4, GFED4s, ESA Biomass CCI and GBEI benchmark datasets[5]**(L258).**

Five different types of RCP scenario climate data were used to determine the impact of fire and climate on forest structure and their interactions **(L248).**
* * *
**L249** Revise grammar

**Response:**

Grammar has been revised.

**L318** Remind the reader what this core variable is

**Response:**

Has been adjusted (**L326**). The core variable of fire products is "fire probability" which affects all fire variables and their derivatives.

**L375** Revise grammar

**Response:**

The grammar of this paragraph (**Lines 368 to 378**) has been revised. Thank you for the detailed review.

**L498** Write "because it is"

**Response:**

Thank you, has been adjusted.

**L499** You write that "Based on the comparison of results between the fire-on and fire-off simulations, the NPP variable under all of the RCP scenarios shows a downward trend with some small fluctuations". It sounds as if the difference between simulations with fires on and off reveals a trend in NPP. I am sure this is not what you mean to say, please rewrite.

**Response:**

Thank you for the detailed review and we're grateful for your understanding and constructive suggestions.

We have adjusted the sentence as follows

We obtained the NPP lost variable due to wildfire from fire-on and fire-off simulations. The NPP lost variable under all RCP scenarios shows a decreasing trend with some small fluctuations (**L509**).

**L565** FDI not defined

**Response:**

Has been adjusted fire danger index (FDI). **L556**

**L642** Replace "dan" with and"

    **Response:**

  Has been adjusted. **(L656)**

**L649** Revise grammar

    **Response:**

  Has been adjusted. **(L662)**

**L655** Please avoid sentences like these. Instead of copying the raw data from the table, facilitate its interpretation.

    **Response:**

Has been adjusted **(L668)**. Thank you very much for your detailed review and valuable suggestions.

"The 20-year average simulation results of HTR variables for 2000-2020 (historical) and 2021-2040 (RCP8.5) showed the highest values compared to the average of other years, with values of $2572.86 \pm 78.08$ and $2534.29 \pm 96.30$ Tg C year$^{-1}$, respectively (Table S5). We suggest that the high HTR values during those years were due to the high decomposition rates of litter and soil organic carbon, and a decrease in the burned fraction during that period (Figure S4.d)."

**L654** Replace "Heterotrophic" with "heterotrophic"

    **Response:**

Has been adjusted. **(L667)**

**L660** Does this apply to all of the RCPs that you assessed?

    **Response:**

Yes, the simulation results under all RCP scenarios show the same trend. **(L673)**

"Overall, SEIB-DGVM SPITFIRE simulates that until the end of the 21st century, there will continue to be a weakening of the land carbon sink in Siberia under all RCP scenarios. This is also reinforced by the CESM2 model simulation, which indicates the same trend globally."

**L661** Why "saturated"?

**Response:**

Apologies for the improper choice of words; we have adjusted it to "pronounced." **L676**

"Negative NBP extremes become more frequent and pronounced by the end of the 21st century, suggesting that terrestrial ecosystems may lose their potential to absorb anthropogenic carbon and mitigate the effects of climate change (Sharma et al., 2023)."

**Figures**

Figure 5 Replace "Kg" with "kg" in figure legend (here and elsewhere). Also, increase font of x-axis, and consider replacing the full month names with their first letters (J, F, M, etc.)

**Response:**

Figure 5 has been adjusted by adjusting the unit writing "kg" and x-axis months in 3-letter abbreviations.

Writing unit "kg" has been adjusted in all figures in the main text and supplement.

Figure 8 The burned biomass should have a time unit as well (mass per unit of area per unit of time). Add GlobFIRM and SPIFIRE to the (c) and (d) panels, respectively, such that the difference between both plots is more obvious.

**Response:**

Has been adjusted by adding time unit (year$^{-1}$) and adding model name labels to panels c and d. Thank you very much for the constructive suggestions.

**Author's response to referee #3**

The authors have made large efforts to improve the study following most of comments from reviewers. However, some concerns remain unsolved and should be responded carefully to improve the credibility of the future projection.

**Response:**

Thank you for recognizing the efforts we have made to improve our study in response to the reviewers' comments. We appreciate your acknowledgement of our work thus far.

We understand that some concerns remain unsolved, and we are committed to addressing them thoroughly in this review stage. Your feedback is crucial in enhancing the credibility of our future projections, and we take your comments very seriously.

Once again, thank you for your constructive feedback and for your dedication to improving the quality of our work.

1) **First,** the validation is still confusing. Fig. 2 compares the spatial distribution of burned fraction from GFED4 and the model. However, it is only the agreement of coverage instead of the locations of hotspots. As I know, most of fires are located in the eastern Siberia at a narrow band. Please show the map of GFED4, the model, and their difference separately with values indicating the spatial distribution of burned fraction.

**Response:**

Thank you for your detailed feedback and for pointing out the confusion regarding the validation of our model. We understand your concern about the need for a more precise comparison of the spatial distribution of burned fractions.

Apologize, perhaps the intent of Figure 2 is Figure S10 (in the supplement) which relates to the comparison of the spatial distribution of the burning fraction between GFED4s and the model.

Since the first version of our manuscript, we have visualized the spatial distribution map of the burned fraction variable. At this second revision stage, the image is in Figure S10 (in the Supplement).

**Author's response to referee #3**

[Figure]

**Figure S10.** Spatial distribution of annual averaged (1997-2016) burned fraction variable of: **(a)** GFED4s **(b)** SEIB-DGVM SPITFIRE

Indeed, hotspot data and burned fraction data are different, but burned fraction data represents the fraction of fire in the proportion of the area (simulated each grid cell) at a certain of time. The more frequent or intense the fire occurs in the area, the higher the burned fraction value. Thus, it can be said that the comparison we have made represents all hotspots from 1997 to 2016 on a large scale in Siberia.

[Figure]

**Figure rev1.** Spatial distribution differences of burned fraction variables from GFED4s and SEIB-DGVM SPITFIRE (GFED4s - SEIB DGVM SPITFIRE)

Figure rev1 shows that there are hotspot differences between SEIB-DGVM SPITFIRE and GFED4s. The high negative value (-0.186) indicates the hotspot in SEIB-DGVM SPITFIRE is higher than the hotspot in GFED4s, while the high positive value indicates the hotspot in GFED4s (0.248).

The hotspots of SEIB-DGVM SPITFIRE are influenced by the fuel load variables available by default (proved by the same hotspot pattern in SEIB-DGVM default: Figure S4.a).New ignition factors (lightning: Figure S1.b and population density: Figure S1.d) affect the burned fraction spread over Siberia, which is different from the default SEIB-DGVM where no other hotspots are visible besides the green circles.

[Figure]

**Figure S4. (a)** Spatial distribution of annual averaged burned fraction of SEIB-DGVM GlobFIRM from 2006 to 2100. **(b)** Spatial distribution of annual averaged burned fraction of SEIB-DGVM SPITFIRE from 2006 to 2100. *Green circle and orange arrow is for review purposes only, isn't available in Figure S4 in the Supplement.*

[Figure]

**Figure S1.** Spatial distribution of lightning flash rate (LIS/OTD HRFC) and population density (GPWv4) input data: **(a)** LIS/OTD HRFC global, **(b)** LIS/OTD HRFC Siberian, **(c)** GPWv4 global, **(d)** GPWv4 Siberian

We have explained in **(L359)**, that there are dynamic extreme events in GFED4s in section 3.1.3, where the current model is not able to accurately spatially simulate the existing extreme events. Currently the model is able to simulate in the long term and a fairly large area (already explained in section 4.4 model uncertainty). In the burned fraction variable, as evidenced by the very slight difference with the GFED4s data, the model broadly has a slight overestimate value of about -0.0121. In the area of eastern Siberia, as you explained there is no detectable difference, perhaps the very small fraction value in the GFED4s data is due to its location in the narrow band.

Regarding the difference in spatial distribution between the model and benchmark data (GFED4s and GFED4) in eastern Siberia, we have explained in section 4.1 Feasibility of fire simulation (**L611**).

**Author's response to referee #3**

**2)** Fig. 6 (original Fig. 9) compares the monthly fire emissions from model and GFED4. The authors provide the data link for the check of high R2 as I questioned. However, there are only 141 data samples for the period of 1997-2016 (20 years or 240 months). How are these 141 samples composed of? Furthermore, the R=0.78 is mainly due to the 0 values for both observations and models. If these zero values are removed, the R will be only ~0.3, suggesting that the model actually has low prediction of fire magnitude when the fires occur.

**Response:**

Thank you for your detailed feedback and for highlighting the concerns regarding Fig. 6 (original Fig. 9).

Regarding the 141 data samples, we acknowledge the need to clarify their composition. These samples represent latitude average grid-scale comparison from both dataset (SEIB-DGVM SPITFIRE and GFED4s).

We have written the source of the data comparison data in the image caption of the previous revised manuscript.

[Figure]

**Figure 6.** Latitude average spatial comparison of simulated $CO_2$ emissions of SEIB-DGVM SPITFIRE with GFED4s from 1997 to 2016 (**a**) and GBEI from 2001 to 2020 (**b**) dataset. Standard deviation obtained from the annual $CO_2$ emission data of each dataset.

Our opinion, the model (especially the global model: DGVM) will be very difficult to simulate on a large scale but with high accuracy on grid-level. This does not mean that we neglect accuracy, but rather that we strive for a broader level of agreement (multiple grids or regions combined). In our current study, the main limitation is still in the simulation of the distribution pattern of the fire variable, which greatly influences its derivative variables: up to the burned biomass emission variable. However, numerically (as a mean value) the model is able to produce values that are very similar (99%) to the benchmark data. This has been explained in section 4.4 Model uncertainty. The limitations of the current model are our opportunities for further development, in order to be able to produce better distribution patterns.

We used latitude average comparison to determine the average value, pattern and dynamics of variables at longitude point of view. Furthermore, in accordance of previous studies, model projections are not validated spatially at grid cell level but they validated the model output with observational data in a numerical comparison (specific area, temporal average, and only few variables). e.g. Fig. 3 Verification of the simulated length of fire season using LPJ-DGVM against observations in the sample regions (Thonicke et al., 2001), Fig. 4 Fire return intervals for the period 1987–96 derived from the national fire statistics of forest services and simulated for the same period by the LPJ-DGVM (Thonicke et al., 2001), Fig. 6. Observed (MODIS) versus simulated fire season lengths for biomes (Thonicke et al., 2010), and Fig. 8. (a) Comparison of observed (Ni, 2004) and simulated net primary production in northern China. (b) Comparison of observed (Sukhinin et al., 2004) and simulated area burnt for 1997–2002 (Thonicke et al., 2010).

In this study we compare both (numerical comparison and spatial comparison: average latitude at Siberian level and regional level) for all major variables. Therefore, our validation process is better because it covers all major variables and uses two types of validation.

We performed an apples-to-apples comparison of the model simulation results and benchmark data by comparing each grid at the average latitude. A value of 0 represents the absence of fire in the area, and the model is also able to simulate this. We believe that a good fire model is not only able to simulate fire events with high accuracy, but also able to simulate the absence of fire events, which is in complete agreement (fire event and non-fire event) with the benchmark data used.

**3)** Fig. 7 compared the PM2.5 emissions from the model and CAMS for 2004-2021 (18 years). Why there are only 12 data points on the scatter plot?

   **Response:**

   Thank you very much for the detailed review, we apologize for the wrong year in the caption.

   Whole CAMS data is from 2004 to 2021 (Romanov et al., 2022). However, we only use CAMS data from 2010 to 2021 for the comparison of $PM_{2.5}$ emissions, because within that year range, both datasets show the same trend.

   We have adjusted L409 and caption in Figure 7.

**4)** **Second,** the future projection seems unreasonable. Although no one could accurately predict future changes of fires, there are still some principles we could follow based on historical variations of boreal fires. The study shows very limited differences of future fire emissions among the four climate scenarios (Fig. 13). In addition, under the RCP8.5 to RCP2.6 scenarios, overall emissions by 2100 are projected to increase by 2.6%, 1.9%, 1.05% and 1.04% compared to 2000 emissions (Figure S28). These changes are too small to believe as future climate is quite different among the four scenarios and the RCP8.5 scenario projects an extremely warming world. The authors claimed that "the fire calculation does not directly use/consider the temperature variable. Instead, the fire variable is estimated based on the calculation of a chain of variables ranging from fuel availability fuel load (litter + aboveground biomass), moisture content…" I think the moisture content and fuel load should also be different under these climate scenarios. How the authors consider the changes of these driving factors in the projection?

   **Response:**

   Thank you for your detailed feedback on our manuscript, particularly regarding the future projections of fire emissions under different climate scenarios.

   We appreciate your concerns about the small differences in future fire emissions among the four climate scenarios and the seemingly limited changes projected by 2100. We understand that this appears counterintuitive given the expected significant differences in future climate conditions, especially under the RCP8.5 scenario.

   We have rechecked most of the variables described in the manuscript and show the same pattern with very little difference between RCP scenarios.

**Author's response to referee #3**

Since early last month, we have been meticulously tracking the flow of climate input data to ensure accurate reading, processing, and output generation.

During this tracking process, we discovered a slight misconfiguration in the reading of climate input data for our future simulations, resulting in a 30-year loop that affected our model output. We have promptly corrected this configuration and conducted a re-simulation following the protocol outlined in the manuscript.

Subsequently, we have revised all figures, tables, and explanations in both the manuscript and supplementary materials.

Regarding the projected changes, we present the latest findings as follows: "Under the RCP8.5 to RCP2.6 scenarios, the twenty-year average comparison of overall burned biomass emissions data from 2080-2100, compared to data from 2000-2020, shows projected increases of 23.87%, 27.63%, 29.34%, and 30.36%, respectively (Figure S43)."

Furthermore, in this simulation, we observed that burned biomass emissions are projected to increase from the highest to the lowest under the RCP2.6, RCP4.5, RCP6.0, and RCP8.5 scenarios, respectively.

The spatiotemporal dynamics of the biomass burning emissions under all RCP scenarios had similar patterns and trends, but they had slightly different variations in dynamics because climate affects the frequency and distribution of fires. This is evidenced by all fire variables produced by the model, from burned fraction to burned biomass emissions. In the last 20 years of the projection (2080-2100), the highest values were obtained from simulations using climate inputs RCP2.6, RCP4.5, RCP6.0, and RCP8.5. This occurs because each RCP scenario exhibits varying radiative forcing, with RCP8.5 notably experiencing the highest temperature increase (Figure S42) and also projecting the highest precipitation levels (Figure S40.b). The fuel load variable follows a corresponding order reflective of RCP forcing levels, with RCP8.5 showing the highest and RCP2.6 the lowest (Figure S40.a). However, due to increased precipitation and temperature-induced snowmelt, the moisture content of litter fractions in RCP8.5 simulations attains the highest values, contrasting with the lowest values in RCP2.6. Consequently, available fuel loads may not ignite in areas with high moisture content, leading to projections of the highest burned biomass emissions in the last 20 years of RCP climate projections (2080-2100) for RCP2.6, RCP4.5, RCP6.0, and RCP8.5, respectively.

**Author's response to referee #3**

We have observed that the resulting fuel load aligns with the order of RCP radiative forcing scenarios, with the highest load under RCP8.5 and the lowest under RCP2.6. Interestingly, the fuel load under the RCP8.5 simulation also exhibits a higher moisture fraction, which correlates with higher precipitation values.

This relationship is supported by the average values of the fire factor variable across scenarios: RCP8.5 (0.88), RCP6.0 (0.89), RCP4.5 (0.91), and RCP2.6 (0.92), indicating a decreasing trend in fire factor with decreasing radiative forcing.

These findings underscore the complex interactions between climate variables and fire behavior, as outlined in our manuscript. In addition, based on the variables that we have presented, it shows that the calculation flow in our fire module has run well according to the input data used and the variable calculation is well-integrated with each other.

[Figure]

**Figure S40. (a)** Temporal variation of simulated SEIB-DGVM SPITFIRE fuel load in Siberia under different RCPs climate scenarios from 2000 to 2100. **(b)** Temporal variation of precipitation under different RCPs climate scenarios in Siberia from 2000 to 2100

**Author's response to referee #3**

[Figure]

**Figure S41. (a)** Temporal variation of simulated SEIB-DGVM SPITFIRE moisture litter fraction in Siberia under different RCPs climate scenarios from 2000 to 2100. **(b)** Temporal variation of fire factor under different RCPs climate scenarios in Siberia from 2000 to 2100

**Author's response to referee #3**

[Figure]

**Figure S42.** Temporal variation of average 2 m air temperature in Siberia under different RCPs climate scenarios from 2000 to 2100